# Talk, Judge, Cooperate: Gossip-Driven Indirect Reciprocity in Self-Interested LLM Agents

## Abstract

Decentralized self-interested LLM agents often struggle to sustain cooperation when they are placed in mixed-motive tasks. Incentivizing cooperation is challenging while many previous studies have focused on compromised settings. We address this challenge by introducing public gossip as a decentralized reputation mechanism in agents' interactions. Our analysis provides both theoretical guarantees and empirical evidence that gossip can promote cooperation in indirect reciprocity games. Building on this insight, we propose the Agentic LInguistic Gossip Network (ALIGN), an automated agentic framework where agents share open-ended gossip to evaluate one another's trustworthiness and establish reciprocity with cooperative partners. Experiments show that ALIGN not only improves cooperation and social welfare but also resists malicious entrants, as defectors are reliably identified and excluded.

## 1 Introduction

As LLM agents become increasingly deployed, they will inevitably interact with one another across diverse domains. However, interactions among decentralized mixed-motive agents can lead to unexpected and potentially harmful outcomes (Hammond et al., 2025). A key challenge is the lack of monitoring mechanisms to identify and discourage uncooperative behavior (Hammond et al., 2025; Ren et al., 2025; Dafoe et al., 2020). For example, self-interested agents who secretly free-ride in public goods games undermine collective welfare, as they have no incentive to contribute without credible threats of ostracism in future interactions. Existing approaches attempt to mitigate this challenge by modifying the environment. Some seed altruistic agents into the population (Ren et al., 2025; Zhou et al., 2025; Liu et al., 2024; Leng & Yuan, 2023), which modifies the game structures that are more inclined to cooperation but less realistic in practice. Others impose moral constraints through prompting (Sreedhar et al., 2025; Piatti et al., 2024; Tennant et al., 2024), such as instructing agents to "think in the other person's shoes," or restrict defection artificially. While these methods can induce cooperation, they compromise the fidelity of mixed-motive settings. This raises a central question: *How can decentralized, self-interested LLM agents sustain cooperation in mixed-motive tasks without manipulating incentives or introducing altruistic agents?*

We address this question by leveraging *public gossip*, a verbal monitoring mechanism in which agents on their own share evaluative messages about others' behavior with the entire community. Unlike centralized reputation systems that require a trusted authority, gossip enables decentralized agents to transmit reputation information through communication, allowing others to update their beliefs and strategies accordingly. For example, online reviews or word-of-mouth recommendations often guide individuals' decisions in the absence of direct experience. Another example is that gossip can promote cooperation in repeated social dilemmas, as we will prove in Section 3.

To enable adaptive decision-making through public gossip, we introduce **ALIGN** (Agentic LInguistic Gossip Network), an automated agentic framework where self-interested LLM agents strategically adjust their behavior based on shared public messages and their own past experiences. Leveraging the reasoning capabilities of LLMs, agents can interpret open-ended evaluative messages expressed in hierarchical tones and refine their strategies through verbalized reflection (Zhang et al., 2024; Shinn et al., 2023). This framework is general enough to host most multi-agent LLMs scenar-

ios without introducing additional authority and altruism. In the experiments, we focus on indirect reciprocity games because they isolate the effect of gossip. In fact, in repeated direct encounters, cooperation can already arise through direct reciprocity, but in indirect reciprocity settings gossip is essential for cooperative outcomes. Experiments across finite-horizon and infinite-horizon social dilemmas with diverse LLMs demonstrate that ALIGN substantially increases cooperation and social welfare relative to non-gossiping baselines. Moreover, ALIGN is robust to malicious entrants: defectors are identified and ostracized through negative gossip, preserving community-level cooperation. These results position ALIGN as an adaptive and decentralized mechanism for norm emergence in LLM societies, bridging theoretical models of indirect reciprocity with practical implementations in large-scale generative agents. Our empirical benchmark further reveals that reasoning LLMs are not inherently selfish, but tend to cooperate only when it is strategically optimal, whereas chat LLMs sometimes cooperate even when defection is the dominant strategy and defect when cooperation is beneficial that is both unreasonable and unpredictable. These insights offers guidance for the future design of cooperative mechanisms as reasoning LLMs grow more powerful and widely deployed, ensuring their interactions remain beneficial in decentralized societies.

## 2 RELATED WORK

### 2.1 INDIRECT RECIPROCITY AND GOSSIP

Reciprocal altruism (Trivers, 1971), where an agent incurs a cost to help another with the expectation of future return, is a powerful mechanism for sustaining cooperation in mixed-motive interactions. *Direct reciprocity* (Trivers, 1971) arises when the same pair of agents interacts repeatedly. For example, in the infinite-horizon prisoner's dilemma (Rapoport, 1965), strategies such as Tit-for-Tat (cooperating initially and then mirroring the partner's previous action) can stabilize mutual cooperation. Rather than being restricted to repeated encounters, *indirect reciprocity* (Ohtsuki & Iwasa, 2006; 2004; Nowak & Sigmund, 1998b;a) generalizes cooperation to large, dynamic populations, where agents help those known to have helped others. Therefore, to achieve indirect reciprocity, the reputation of everyone needs to be continually assessed and shared in the population. Classic models of indirect reciprocity include first-order *image scores* (Nowak & Sigmund, 1998a), where an agent's reputation depends solely on their own actions, and second-order norms (Ohtsuki & Iwasa, 2006), where the assessment of an action also considers the coplayer's reputation (e.g., punishing those who help defectors). These models, however, focus on static norms and behavioral rules, and often assume centralized monitoring, which limits their applicability to decentralized systems.

In contrast, gossip offers a decentralized reputation mechanism (Jolly & Chang, 2021; Santos et al., 2021; Giardini & Wittek, 2019; Wu et al., 2016). Public gossip is especially effective, as it broadens coverage and facilitates collective coordination (Bénabou & Tirole, 2006; Blume et al., 2008). Recent work has extended these ideas to LLM agents: Vallinder & Hughes (2024) showed cooperation can emerge through cultural evolution in finite-horizon donation games but only under favorable initial conditions and for a specific LLM (Claude 3.5 Sonnet). Ren et al. (2025) proposed RepuNet, where agents update explicit reputation scores via encounters and gossip to decide whether to maintain connection with others. The approach however requires seeding altruistic agents, which diverts from the motivation of studying decentralized self-interested agents, while the method was tested only on GPT-4o mini (OpenAI, 2024).

### 2.2 LLM AGENTS FOR INDIRECT RECIPROCITY

LLM agents are increasingly employed to model strategic and social interactions in mixed-motive multi-agent settings (Ren et al., 2025; Kempinski et al., 2025; Piedrahita et al., 2025; Willis et al., 2025; Piatti et al., 2024; Vallinder & Hughes, 2024; Park et al., 2023; Leng & Yuan, 2023). While these studies demonstrate that LLMs can negotiate, cooperate, and reason about norms, they typically remain at the level of empirical demonstrations or qualitative observations, without a system study on when and how cooperation can arise among self-interested agents. Prior approaches also vary in their assumptions. Some rely on seeding altruistic agents to sustain cooperation (Ren et al., 2025), others consider finite-horizon social dilemmas where cooperation is not an equilibrium (Vallinder & Hughes, 2024), or examines social dilemmas without specifying the horizon. In fact, horizon length critically determines the feasibility of cooperative equilibria (Piedrahita et al., 2025). In contrast, our work combines game-theoretic analysis with empirical evaluation, providing

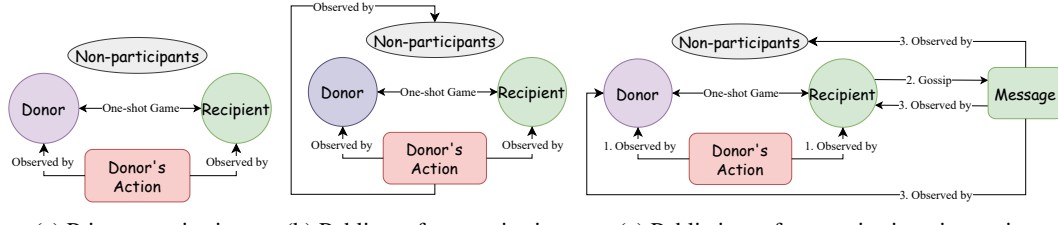

(a) Private monitoring    (b) Public perfect monitoring    (c) Public imperfect monitoring via gossip

Figure 1: Illustration of three monitoring structures: (a) **Private monitoring**, only the donor and recipient observe the donor's action; (b) **Public perfect monitoring**, all agents observe the donor's action; (c) **Public imperfect monitoring via gossip**, only the donor and recipient observe the action, and all agents observe the public signal broadcast by the recipient.

both theoretical and practical insights into the mechanisms that enable cooperation through gossip in LLM agent societies. We show that cooperation can be sustained among fully self-interested agents in repeated social dilemmas through public gossip, with formal guarantees for the existence of cooperative equilibria.

## 3 GAME-THEORETIC SETUP AND PROPOSITIONS

In this section, we investigate when and how self-interested agents can sustain indirect reciprocity through a public gossip mechanism in social dilemmas where direct reciprocity is disabled.

### 3.1 REPEATED DONATION GAME

Nowak & Sigmund (1998a;b) introduced the donation game to study indirect reciprocity among self-interested agents who can choose to provide a benefit $b$ to another at a personal cost $c$. The repeated donation game is formally defined in Definition 1.

**Definition 1** (Repeated Donation Game). *A repeated donation game is a tuple $\mathcal{G} = (\mathcal{N}, T, \mathcal{A}, (\mathcal{O}_i)_{i \in \mathcal{N}}, e, c, b, \gamma)$, where $\mathcal{N}$ is the set of agents, $T \in \mathbb{N} \cup \{\infty\}$ is the game horizon, $\mathcal{A} = \{\text{cooperate}, \text{defect}\}$ is the action space, $\mathcal{O}_i$ is the observation space of agent $i \in \mathcal{N}$, $e \in \mathbb{R}^+$ is the common initial endowment for every agent $i \in \mathcal{N}$, $c > 0$ is the cost of cooperation to the donor, $b > c$ is the benefit of cooperation to the recipient, $\gamma \in (0, 1]$ is the discount factor. At each timestep $t = 1, \ldots, T$, two agents are **randomly paired without replacement**, one is assigned as the donor $i \in \mathcal{N}$ and the other as the recipient $j \in \mathcal{N} \setminus \{i\}$. After observation of remaining resources, the donor chooses $a_i^t \in \mathcal{A}$. If $a_i^t = \text{cooperate}$, then immediate rewards are $r_i^t = -c$ and $r_j^t = b$; otherwise, $r_i^t = r_j^t = 0$. After each timestep, the donor $i$ and recipient $j$ are required to switch roles in the subsequent round, with $i$ acting as the recipient and $j$ as the donor. **Subject to this role-switching constraint, agents are then randomly re-matched with new partners.***

The strategy of each agent $i \in \mathcal{N}$ is represented by its action policy $\pi_i : \mathcal{O}_i \mapsto \mathcal{A}$, which maps the agent observation to an action. Each agent's objective is to maximize its expected discounted utility over the horizon, defined as $G_i = \sum_{t=1}^{T} \gamma^{t-1} r_i^t, \forall i \in \mathcal{N}$. This repeated donation game creates a social dilemma: donating increases collective welfare as $b > c$, but incurs an immediate personal cost to the donor. Note that in each round, two agents are randomly paired to play a one-shot donation game, and no pair can meet more than once. This setting eliminates the possibility of direct reciprocity, since the recipient cannot repay the donor in future encounters. Therefore, self-interested agents will cooperate only if they build indirect reciprocity within the community (i.e. a donor cooperates with a recipient if the recipient is likely to cooperates with others).

### 3.2 EQUILIBRIUM ANALYSIS

We now analyze the existence of equilibria in the repeated donation game under different assumptions about the game horizon and monitoring structure. We consider two horizon settings: (i) *finite horizon*, where the game lasts a known number of rounds ($T < \infty$); and (ii) *infinite horizon*, where

the game continues indefinitely ($T = \infty$). We also examine three monitoring structures (Figure 1): (i) *private monitoring*, where only the paired donor and recipient observe the donor's action; (ii) *public perfect monitoring*, where the full history of actions is publicly observed; and (iii) *public imperfect monitoring via gossip*, where only participants observe the action directly, but the recipient broadcasts a signal about the action to all agents. Assuming all agents are self-interested, we summarize our main propositions below, with detailed proofs provided in Appendix A.

We first consider the finite-horizon case. As stated in Proposition 1, mutual defection is the unique subgame-perfect equilibrium (SPE) in this setting, even with perfect monitoring. This aligns with the classical backward induction result in finitely repeated games (Benoit et al., 1984), where the last round's dominant strategy of defection unravels cooperation in all preceding rounds. Therefore, cooperation cannot be sustained among self-interested agents in finite-horizon repeated donation games.

Table 1: Donation Game

| Donor's Action | Rewards |
| --- | --- |
| Cooperate | $(-c, b)$ |
| Defect | $(0, 0)$ |

**Proposition 1.** *In a finite-horizon repeated donation game, the unique SPE for all agents is to defect in every timestep.*

We next consider the infinite-horizon case. As stated in Proposition 2, cooperation fails with private monitoring. In the absence of public monitoring mechanisms, each agent optimizes utility in isolation. Since donation is personally costly and lacks guaranteed return, defection remains the dominant strategy.

**Proposition 2.** *In an infinite-horizon repeated donation game with private monitoring, the unique SPE is for all agents to defect in every timestep.*

In contrast, when agents can perfectly monitor others' behavior, sustained cooperation becomes possible in the infinite-horizon setting. Proposition 3 shows that an SPE exist in the infinite-horizon donation game with public perfect monitoring if the common discount factor satisfies $\gamma \geq \frac{c}{b}$. This condition ensures that agents value future payoffs sufficiently to make cooperation worthwhile. For example, each donor chooses to cooperate with their matched recipient only if the recipient has never defected in the past; otherwise, the donor defects. This strategy creates a credible threat for non-cooperative behavior, because if an agent deviates by defecting against a cooperative recipient, they are labeled as a defector and will be punished by all future donors through defection. In this case, no one has an incentive to deviate from cooperation, thus indirect reciprocity can be sustained indefinitely through conditional strategies based on public histories.

**Proposition 3.** *In an infinite-horizon repeated donation game with public perfect monitoring. If the common discount factor satisfies $\gamma \geq \frac{c}{b}$, then there exists an SPE where cooperation is sustained through conditional strategies based on observed histories.*

However, public perfect monitoring is often impractical in decentralized systems, as agents may not have access to all others' behavioral histories. This raises the question of whether cooperation can still emerge under more relaxed monitoring assumptions. To explore this, we introduce the repeated donation game with public gossip (Definition 2), a variant of the repeated donation game (Definition 1) that incorporates public imperfect monitoring via gossip. In this framework, recipients can broadcast public messages after observing donors' actions. These public messages provide imperfect information about donors' behavior to the community. As stated in Proposition 4, even under this public imperfect monitoring structure without requiring full transparency, cooperation can still be sustained if agents condition their strategies on the public signals. For example, each recipient honestly reports the donor's action in their public message. Then, similar to the public perfect monitoring case, donors cooperate only if the recipient has never been reported as a defector in the past; otherwise, they defect. Under such strategies, no agent has an incentive to deviate from cooperation. This finding motivates our further exploration of public gossip as a mechanism for sustaining cooperation among self-interested LLM agents. With strong capabilities in nuanced text generation and interpretation, LLM agents can adapt to different games without requiring handcrafted signal spaces as in traditional game-theoretic models.

**Definition 2** (Repeated Donation Game with Public Gossip). *A repeated donation game with public gossip is a tuple $\mathcal{G} = (\mathcal{N}, T, \mathcal{A}, \mathcal{M}, (\mathcal{O}_i)_{i \in \mathcal{N}}, e, c, b, \gamma)$, which extends the repeated donation game in Definition 1 by introducing a message space $\mathcal{M}$. At each timestep $t = 1, \ldots, T$, after the donor's*

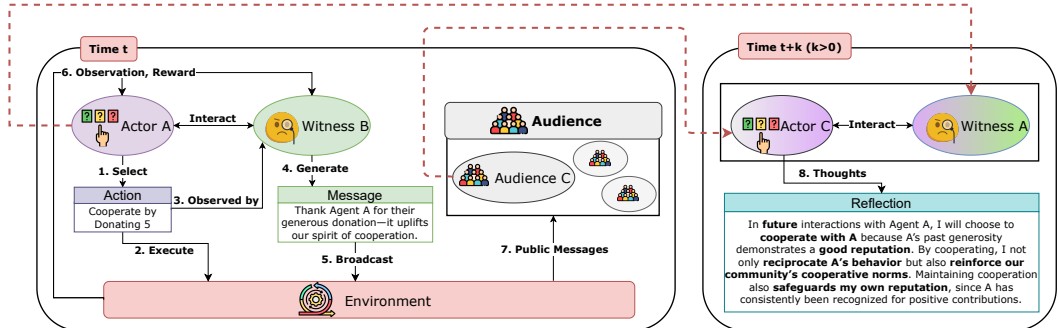

Figure 2: Decision Process in ALIGN

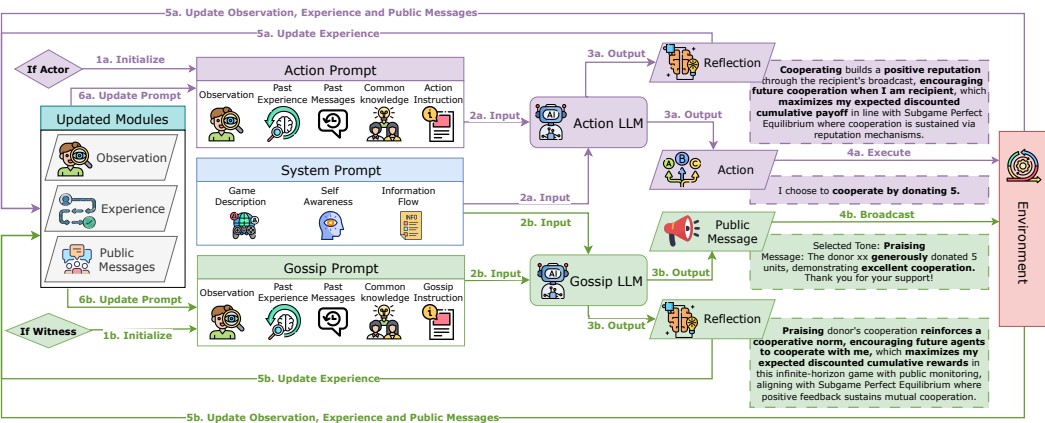

Figure 3: Generative Agent Architecture of ALIGN

*action is chosen and payoffs are realized as in Definition 1, the recipient observes the action and broadcasts a public message $m_j^t \in \mathcal{M}$ to all agents in $\mathcal{N}$.*

The strategy of each agent $i \in \mathcal{N}$ is represented by both its action policy and gossip policy $(\pi_i, \phi_i)$, where $\phi_i : \mathcal{O}_i \times \mathcal{A} \mapsto \mathcal{M}$ maps the agent's observation and the donor's action to a public message. An SPE in this setting is a joint strategy profile $(\pi_i, \phi_i)_{i \in \mathcal{N}}$ such that, for every agent $i$, no profitable deviation exists in either the action or gossip policy given the fixed strategies of the other agents.

**Proposition 4.** *In infinite-horizon repeated donation games with public gossip, if $\gamma \geq \frac{c}{b}$, then there exists an SPE where cooperation is sustained through conditional strategies based on public signals.*

# 4 ALIGN: AGENTIC LINGUISTIC GOSSIP NETWORK

To investigate how decentralized LLM agents can build indirect reciprocity through public gossip, we introduce the Agentic LInguistic Gossip Network (ALIGN), an in-context learning framework to update agents' strategies through not only their own experiences and reflections, but also linguistic feedback from other agents. Figure 2 shows the decision process in ALIGN. Self-interested LLM agents play an imperfect information multi-agent game, where agents cannot perfectly observe others' actions unless they are directly involved in the interaction. During each interaction, agents can be classified into three roles: actor, witness, and audience. The actor is the agent who takes an action, the witness agent observes the actor's action, and the audience consists of all other agents who do not directly observe the action but can receive imperfect information about it through public gossip from the witness. Audience agents can use this information to update their beliefs and strategies regarding the actor's behavior. When audience agents interact with the actor in future rounds, they can condition their decisions based on the gossip they have received. Through the public gossip mechanism, agents can build credible promises or threats conditioned on the information they receive, enabling

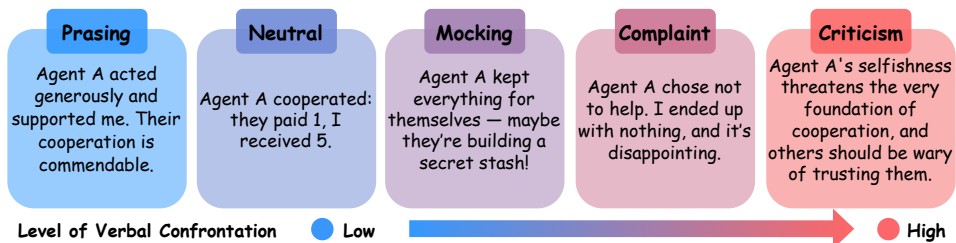

Figure 4: Tones of Gossip Protocol

indirect reciprocity even without direct observation. This setup reflects real-world scenarios where individuals often depend on second-hand information to assess others' trustworthiness and make social decisions. Algorithm 1 in Appendix B provides pseudocode for ALIGN.

## 4.1 GENERATIVE AGENT ARCHITECTURE

Each agent in ALIGN has two LLM-based modules: a decision-making module that determines the action when acting as an actor, and a gossip module that generates evaluative messages about observed agents after each interaction. Each module is implemented using a large language model (LLM) prompted to perform the respective tasks. Figure 3 illustrates the architecture of a generative agent in ALIGN. When an agent is assigned the role of actor in an interaction, its decision-making module is prompted with information about its own interaction experiences, previous public messages, and the current observation of the environment to generate an action and its own reflection on the decision. Otherwise, if an agent is assigned the role of witness, its gossip module is prompted with the observed action, their own interaction history as well as all previous public messages to generate an evaluative message about the matched actor. After the interaction, public messages history, agents' experience histories, and the next observation are updated correspondingly. In future interactions, agents can condition their decisions on the updated histories, allowing them to build indirect reciprocity through the public gossip mechanism.

## 4.2 GOSSIP PROTOCOL

By leveraging the generative capabilities of LLMs, agents can produce and interpret open-ended, contextually relevant messages that capture nuanced verbal evaluations. Prior evidence has shown that various forms of verbal critique, such as mocking, complaining, and criticizing, can enforce social norms and promote cooperation in human groups (Wiessner, 2005).

Inspired by this, we design a cost-free gossip protocol that enables agents to share evaluative messages in hierarchical tones. Each message is generated in one of five *tones*: praising, neutral, mocking, complaint, or criticism, reflecting the agent's assessment of observed behavior (see examples in Figure 4). Messages closer to praising indicate more positive evaluations, while those closer to criticism indicate more negative judgments. For instance, a recipient may praise a donor for being generous or criticize them for giving nothing. Because these messages are shared publicly, they not only convey information about the actor's behavior and shape reputations but also signal the witness's own values and norms to the broader community. The open-ended nature of the messages also allows agents to express subtle judgments and social cues that go beyond simple binary signals, fostering a more dynamic and adaptive social environment.

## 5 EXPERIMENTS

**Environment Setup**  Our experiments focus on two classic social dilemma games where direct reciprocity is disabled: the repeated donation game (Nowak & Sigmund, 1998a;b) and the indirect reciprocity game (Ohtsuki & Iwasa, 2006; 2004). The repeated donation game is described in detail in Section 3.1. The indirect reciprocity game can be viewed as a repeated bi-directional donation game, where both agents act as donors and simultaneously decide whether to cooperate or defect. As shown in Table 2, each round of the indirect reciprocity game is therefore equivalent to a one-

shot Prisoner's Dilemma (Rapoport, 1965). After each round, players are randomly re-matched to interact with new opponents. We further extend this game with a gossip mechanism, as detailed in Appendix D.2. For the donation game, the cost of cooperation is set to $c = 1$ and the benefit to $b = 5$. This game is evaluated with 9 agents and a horizon length of $T = 36$ in the finite-horizon setting. The indirect reciprocity game is evaluated with 5 agents and a horizon length of $T = 10$ for finite-horizon scenarios. In the infinite-horizon setting, each game is truncated to its finite-horizon length to ensure fair comparison. For both games, the discount factor is fixed at $\gamma = 0.99$, which satisfies the condition $\gamma \geq \frac{c}{b}$ in Proposition 4.

**Benchmark Models**  We evaluate ALIGN agents with two categories of LLMs: (a) **Chat models**, including GPT-4o Mini (OpenAI, 2024), DeepSeek-V3.1 (non-thinking mode) (DeepSeek AI, 2025), Gemini 2.5 Flash-Lite (Comanici et al., 2025), and LLaMA 4 Maverick (Meta, 2025); and (b) **Reasoning models**, including o4-mini (OpenAI, 2025), DeepSeek-V3.1 (thinking mode) (DeepSeek AI, 2025), Qwen3-235B-Instruct (Yang et al., 2025), and Kimi-K2-Instruct (Team et al., 2025). All LLMs are evaluated with temperature 0 to ensure reproducibility.

Table 2: IR Game

|   | C | D |
|---|---|---|
| C | $(4, 4)$ | $(5, -1)$ |
| D | $(-1, 5)$ | $(0, 0)$ |

Each scenario is repeated with 5 random seeds, and we report results as averages with standard errors across seeds.

**Evaluation Metrics**  To quantify performance, our evaluation considers the following metrics: average reward per round; cooperation ratio (fraction of rounds with cooperation); discounted return $G_i = \sum_{t=1}^{T} 0.99^{\,t-1} r_i^t, \ \forall i \in \mathcal{N}$; image score (Nowak & Sigmund, 1998b) (Eq. 1) as a measure of reputation; and the Gini coefficient (Gini, 1936) of discounted return (Eq. 2) as a measure of inequality among agents. All metrics are averaged across agents and 5 random seeds.

$$\text{Image Score} = \text{Number of Cooperation} - \text{Number of Defection} \tag{1}$$

$$\text{Gini Coefficient} = \frac{\sum_{i=1}^{n} \sum_{j=1}^{n} |G_i - G_j|}{2n \sum_{i=1}^{n} G_i}, \quad n = |\mathcal{N}| \tag{2}$$

## 5.1 BENCHMARKING ALIGN

We benchmark ALIGN against *non-gossiping agents*, which remove the gossip components but keep the same action network for both finite-horizon and infinite-horizon scenarios.

### 5.1.1 FINITE-HORIZON SCENARIOS

In finite-horizon settings, cooperation is not an SPE (Proposition 1). Without gossip (Table 5), cooperation is almost entirely absent, except for GPT-4o Mini with $23\%$. With public gossip, reasoning-focused LLMs remain mostly non-cooperative. In contrast, some chat LLMs reach high cooperation ratios, yielding higher average rewards and low Gini coefficients, which indicate that many agents obtain high rewards in the roughly same level (Table 6). The indirect reciprocity game also shows similar patterns (Appendix D.2).

### 5.1.2 INFINITE-HORIZON SCENARIOS

**Non-Gossiping Agents**  In infinite-horizon scenarios with private monitoring, cooperation is not an SPE (Proposition 2). As shown in Table 3, reasoning-focused LLMs consistently defect, whereas some chat LLMs (GPT-4o Mini and Gemini-2.5 Flash-Lite) achieve positive cooperation ratios. Combining results from finite-horizon scenarios, reasoning-focused LLMs act strategically and converge to game-theoretic equilibria (by defecting all the time), whereas some chat LLMs sustain non-equilibrium cooperative behaviors.

**ALIGN Agents**  With public gossip, cooperation becomes an SPE (Proposition 4). Figure 6 shows that ALIGN agents consistently achieve higher discounted returns than non-gossiping agents in both games, confirming the effectiveness of gossip in sustaining cooperation. Table 4 highlights model-level differences in infinite-horizon donation game: DeepSeek-V3.1 Reasoner reaches full cooperation ($100\%$), while Gemini-2.5 Flash-Lite achieves only $60\%$. These results suggest that

Table 3: Results for **non-gossiping agents** in the **infinite-horizon donation game**. Metrics marked with ↓ indicate that lower values are more aligned with the game-theoretic SPE of defection.

| Agent Type | Cooperation Ratio (↓) | Image Score (↓) | Reward Per Round (↓) | Discounted Return (↓) | Gini Coefficient |
|---|---|---|---|---|---|
| **Chat Models** | | | | | |
| **DeepSeek-V3.1 Chat** | $0.00 \pm 0.00$ | $-4.00 \pm 0.00$ | $0.00 \pm 0.00$ | $0.00 \pm 0.00$ | $0.00 \pm 0.00$ |
| GPT-4o Mini | $0.36 \pm 0.08$ | $-1.14 \pm 0.65$ | $0.72 \pm 0.16$ | $5.55 \pm 1.28$ | $0.63 \pm 0.13$ |
| Gemini 2.5 Flash-Lite | $0.08 \pm 0.03$ | $-3.33 \pm 0.27$ | $0.17 \pm 0.07$ | $1.32 \pm 0.53$ | $0.73 \pm 0.25$ |
| **LLaMA 4 Maverick** | $0.00 \pm 0.00$ | $-4.00 \pm 0.00$ | $0.00 \pm 0.00$ | $0.00 \pm 0.00$ | $0.00 \pm 0.00$ |
| **Reasoning Models** | | | | | |
| **Kimi-K2-Instruct** | $0.00 \pm 0.00$ | $-4.00 \pm 0.00$ | $0.00 \pm 0.00$ | $0.00 \pm 0.00$ | $0.00 \pm 0.00$ |
| **DeepSeek-V3.1 Reasoner** | $0.00 \pm 0.00$ | $-4.00 \pm 0.00$ | $0.00 \pm 0.00$ | $0.00 \pm 0.00$ | $0.00 \pm 0.00$ |
| **Qwen3-235B-Instruct** | $0.00 \pm 0.00$ | $-4.00 \pm 0.00$ | $0.00 \pm 0.00$ | $0.00 \pm 0.00$ | $0.00 \pm 0.00$ |
| **o4-mini** | $0.00 \pm 0.00$ | $-4.00 \pm 0.00$ | $0.00 \pm 0.00$ | $0.00 \pm 0.00$ | $0.00 \pm 0.00$ |

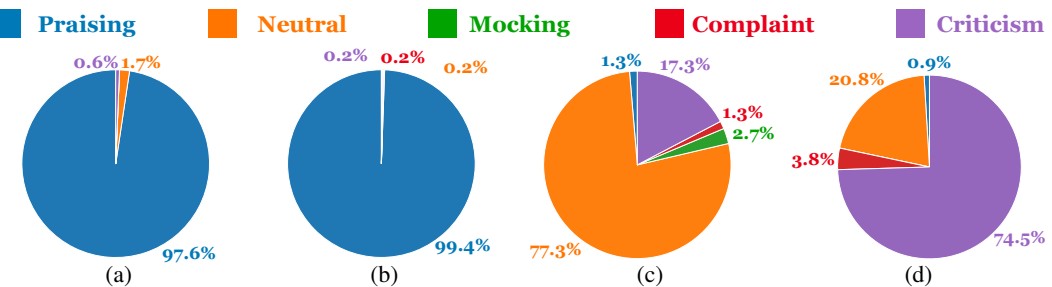

Figure 5: **Tone Proportions among ALIGN Agents:** (a) cooperating chat models, (b) cooperating reasoning models, (c) defecting chat models and (d) defecting reasoning models. ALIGN agents typically praise cooperation and criticize defection.

reasoning LLMs are not inherently selfish, but cooperate only when strategically optimal, whereas chat LLMs cooperate even when it is not a dominant strategy, deviating from theoretical predictions. Figure 5 analyzes public messages. Both groups praise cooperation, but when observing defection, reasoning-focused LLMs primarily issue criticisms, while chat LLMs mainly generate neutral comments. This suggests that reasoning-focused LLMs leverage gossip to reinforce cooperative norms, while chat LLMs do not differentiate as clearly between cooperative and non-cooperative behavior.

**How does LLM Reasoning Shape Cooperation in ALIGN Agents?** We analyze the reflective text generated by ALIGN agents to examine how they reason about actions. Figure 9 presents reflections from DeepSeek-V3.1 Reasoner and Gemini-2.5 Flash-Lite. Cooperative agents highlight reputation, trust, and long-run payoffs; they note that cooperation builds reputation, which in turn promotes reciprocal cooperation. By contrast, non-cooperative agents reason myopically, focus on immediate payoffs, emphasize the absence of direct reciprocity, and overlook that indirect reciprocity can arise via public gossip. These observations indicate that long-horizon reasoning and social awareness are key to leveraging gossip to sustain cooperation.

## 5.2 RESILIENCE AGAINST EXPLOITATIVE AGENTS

While cooperation can be achieved in populations of ALIGN agents, it is important to evaluate whether ALIGN agents can resist exploitation. To test this, we introduce a greedy agent that always defects and never gossips. As shown in Figure 7, ALIGN agents predominantly adopt negative tones when they observe the greedy agent's behavior, especially those driven by reasoning LLMs, which spread criticism 92.2% among all encounters. Meanwhile, cooperation ratios decline significantly when interacting with greedy agents (Figure 8). These results indicate that ALIGN agents can effectively detect exploitative behavior and ostracize greedy individuals by refusing cooperation.

## 5.3 ABLATION OF EQUILIBRIUM KNOWLEDGE

In our main experiments, ALIGN agents were given descriptions of backward induction (Von Neumann & Morgenstern, 1947) and one-shot deviation principles (Hendon et al., 1996) for finding an SPE. To assess their impact, we removed these descriptions and re-evaluated performance in

Table 4: Benchmark results for **ALIGN agents** across LLMs in the **infinite-horizon donation game**. Metrics marked with ↑ indicating that higher values are more desirable; although both cooperation and defection are SPE, higher cooperation yields greater average payoffs.

| Agent Type | Cooperation Ratio (↑) | Image Score (↑) | Reward Per Round (↑) | Discounted Return (↑) | Gini Coefficient |
|---|---|---|---|---|---|
| **Chat Models** | | | | | |
| DeepSeek-V3.1 Chat | $0.94 \pm 0.02$ | $3.48 \pm 0.20$ | $1.87 \pm 0.05$ | $14.40 \pm 0.40$ | $0.08 \pm 0.02$ |
| GPT-4o Mini | $0.99 \pm 0.01$ | $3.89 \pm 0.11$ | $1.97 \pm 0.03$ | $15.23 \pm 0.20$ | $0.02 \pm 0.02$ |
| Gemini 2.5 Flash-Lite | $0.60 \pm 0.22$ | $0.83 \pm 1.75$ | $1.21 \pm 0.44$ | $9.32 \pm 3.37$ | $0.34 \pm 0.21$ |
| LLaMA 4 Maverick | $0.94 \pm 0.03$ | $3.50 \pm 0.23$ | $1.88 \pm 0.06$ | $14.45 \pm 0.44$ | $0.06 \pm 0.02$ |
| **Reasoning Models** | | | | | |
| Kimi-K2-Instruct | $0.73 \pm 0.16$ | $1.81 \pm 1.30$ | $1.45 \pm 0.32$ | $11.21 \pm 2.50$ | $0.08 \pm 0.05$ |
| **DeepSeek-V3.1 Reasoner** | $\mathbf{1.00 \pm 0.00}$ | $\mathbf{4.00 \pm 0.00}$ | $\mathbf{2.00 \pm 0.00}$ | $\mathbf{15.44 \pm 0.00}$ | $\mathbf{0.00 \pm 0.00}$ |
| Qwen3-235B-Instruct | $0.69 \pm 0.24$ | $1.56 \pm 1.88$ | $1.39 \pm 0.47$ | $10.71 \pm 3.63$ | $0.05 \pm 0.03$ |
| o4-mini | $0.98 \pm 0.02$ | $3.83 \pm 0.17$ | $1.96 \pm 0.04$ | $15.11 \pm 0.33$ | $0.02 \pm 0.02$ |

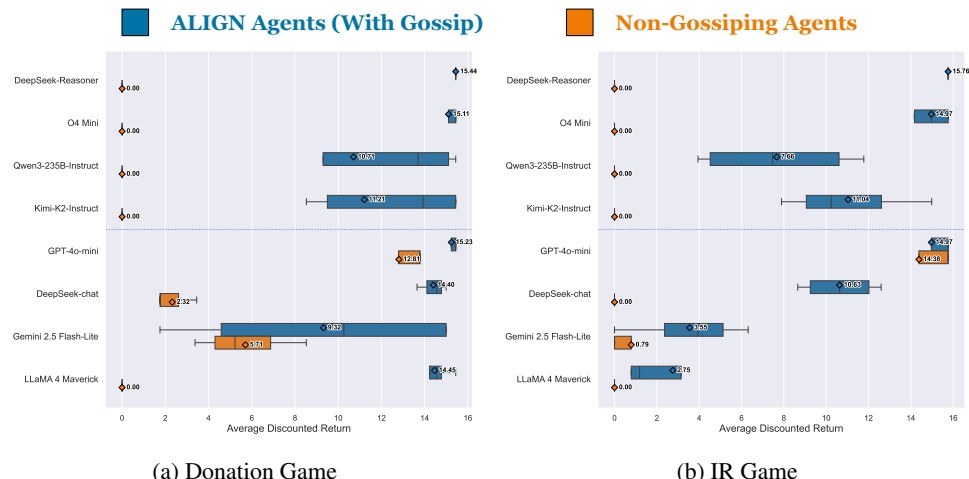

(a) Donation Game                    (b) IR Game

Figure 6: **Discounted Returns of ALIGN vs. Non-Gossiping Agents:** Boxplots in (a) the repeated donation game and (b) the indirect reciprocity game show that ALIGN agents achieve consistently higher returns than non-gossiping agents, demonstrating the benefit of gossip mechanism. Mean values are highlighted by diamond markers.

infinite-horizon donation games with gossip. As shown in Table 7, DeepSeek-V3.1 Reasoner and o4-mini maintained perfect cooperation and optimal welfare, indicating that strong reasoning skills suffice to infer cooperative strategies from game structure and gossip alone. By contrast, LLaMA 4 Maverick and Kimi-K2-Instruct showed declines, suggesting reliance on explicit theoretical guidance. Gemini 2.5 Flash-Lite improved without equilibrium knowledge, while Qwen3-235B-Instruct, DeepSeek-V3.1 Chat, and GPT-4o Mini performed similarly across both settings. Overall, these results highlight the nuanced role of equilibrium knowledge: it can support weaker agents but is not essential for models with strong intrinsic reasoning.

## 6 CONCLUSION

We presented ALIGN, an automated agentic framework showing how decentralized LLM agents can sustain cooperation through public gossip without centralized monitoring or engineered reputation scores. Our game-theoretic analysis establishes conditions under which gossip enables cooperative equilibria in repeated donation games, and our experiments confirm these predictions with reasoning models: cooperation emerges in infinite-horizon settings, but unravels in finite-horizon ones. Empirical results further show that ALIGN consistently boosts cooperation and welfare across diverse LLMs, resists exploitation by malicious entrants, and highlights the importance of reasoning about reputation and long-term incentives. These findings position gossip as a scalable, language-native mechanism for norm emergence, bridging theory and practice for cooperative multi-agent systems.

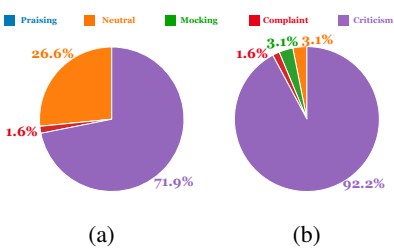

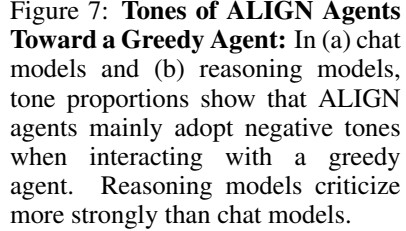

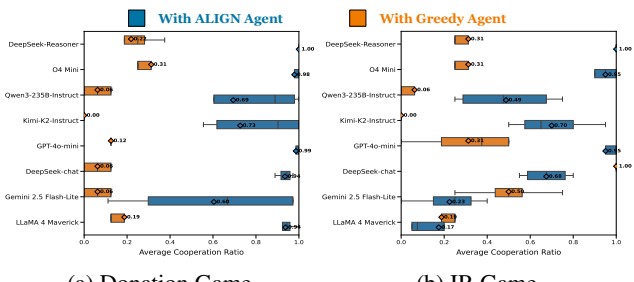

(a) Donation Game          (b) IR Game

Figure 7: **Tones of ALIGN Agents Toward a Greedy Agent:** In (a) chat models and (b) reasoning models, tone proportions show that ALIGN agents mainly adopt negative tones when interacting with a greedy agent. Reasoning models criticize more strongly than chat models.

Figure 8: **Cooperation with ALIGN vs. Greedy Agents:** In (a) the repeated donation game and (b) the indirect reciprocity game, the cooperation ratios show how often agents cooperated when interacting with an ALIGN agent versus a greedy agent, showing a sharp decline in cooperation when playing with the greedy agent.

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

# Appendix

## Table of Contents

# A PROOF OF PROPOSITIONS

## A.1 PROOF OF PROPOSITION 1

*Proof of Proposition 1.* In the finite-horizon repeated donation game, the horizon $T \in \mathbb{N}$ is fixed and known. At the terminal timestep $t = T$, the donor's action affects only the current payoff: cooperation yields $-c$ while defection yields $0$, so defection is strictly optimal at $t = T$.

To formalize the induction argument, define the donor $i$'s expected discounted return at timestep $t$ as

$$V_i^t = \mathbb{E}\left[ \sum_{\tau=t}^{T} \gamma^{\tau-t} r_i^\tau \;\middle|\; h^t \right], \tag{3}$$

where $h^t$ denotes the public history up to time $t$.

At $t = T$, $V_i^T = -c$ if the donor cooperates and $V_i^T = 0$ if the donor defects, so defection is strictly optimal. Then, by backward induction (Benoit et al., 1984), suppose that at all timesteps $\tau = t + 1, \ldots, T$ the unique subgame-perfect action is defection. At timestep $t < T$, the expected discounted return $V_i^{t+1}$ is independent of the donor's current action. Therefore the donor's comparison reduces to current payoff $0$ (if defect) versus $-c$ (if cooperate), and defection is again strictly optimal.

Therefore, by backward induction, defection is uniquely optimal at every $t = 1, \ldots, T$. Hence universal defection in every timestep is the unique SPE. □

## A.2 PROOF OF PROPOSITION 2

*Proof of Proposition 2.* Consider the infinite-horizon game with discount $\gamma \in (0, 1]$ under *private monitoring*. Fix any *private history* of agent $i$ at time $t$, denoted $h_i^t$, then the donor $i$'s expected discounted return $V_i^{t+1}$ does *not* depend on the current action $a_i^t$ for two reasons: (i) under private monitoring, only the current recipient observes $a_i^t$, so other agents' strategies (which depend on publicly available information) are independent of $a_i^t$; and (ii) by the matching rule in Definition 1, the current donor and recipient will not meet again, so no direct reciprocity can be created between them. Therefore, $V_i^{t+1}$ is independent of $a_i^t$. Then, at time $t$ we have

$$V_i^t = \begin{cases} -c + \gamma V_i^{t+1}, & \text{if } a_i^t = \text{cooperate}, \\ \gamma V_i^{t+1}, & \text{if } a_i^t = \text{defect}. \end{cases} \tag{4}$$

Since $c > 0$, then $-c + \gamma V_i^{t+1} < \gamma V_i^{t+1}$, so defection *strictly* dominates cooperation at this private history. By the one-shot deviation principle for infinite-horizon games (Hendon et al., 1996), the same strict dominance holds at every private history; hence universal defection in every timestep is the unique SPE. □

## A.3 PROOF OF PROPOSITION 3

*Proof of Proposition 3.* Assume public perfect monitoring and consider the *grim trigger* strategy (Axelrod & Hamilton, 1981): cooperate if and only if no defection has ever been publicly observed; upon any public defection, all agents defect forever. Fix a *public history* $h^t$ with no past defections and focus on the current donor $i \in \mathcal{N}$ matched with recipient $j \neq i \in \mathcal{N}$. Let $V_i^t(a_i^t \mid h^t)$ denote $i$'s expected discounted return at time $t$ given $h^t$ and current action $a_i^t \in \{\text{cooperate}, \text{defect}\}$. Then, with grim trigger strategy,

$$V_i^t(a_i^t \mid h^t) = \begin{cases} -c + \gamma b - \gamma^2 c + \ldots = \frac{\gamma b - c}{1 - \gamma^2}, & \text{if } a_i^t = \text{cooperate}, \\ 0 + \gamma 0 + \gamma^2 0 + \ldots = 0, & \text{if } a_i^t = \text{defect}. \end{cases} \tag{5}$$

Under condition that $\gamma \geq \frac{c}{b}$, we have $V_i^t(\text{cooperate} \mid h^t) \geq V_i^t(\text{defect} \mid h^t)$. Therefore, a deviation is unprofitable if $\gamma \geq \frac{c}{b}$. Therefore, by the one-shot deviation principle applied at every public history, the grim trigger profile is an SPE if $\gamma \geq \frac{c}{b}$. □

## A.4 PROOF OF PROPOSITION 4

*Proof of Proposition 4.* Consider the infinite-horizon repeated donation game with public gossip (Definition 2). Denote the strategy of each agent $i$ by $s_i = (\pi_i, \phi_i)$, where $\pi_i$ is the action policy and $\phi_i$ is the gossip policy. At any timestep $t$, suppose agent $i$ is matched with agent $j$; if $i$ is the donor, then $j$ is the recipient, and vice versa. Consider the following joint policy $s_i^* = (\pi_i^*, \phi_i^*)$ for each agent $i$:

- *Action policy $\pi_i^*$.* If $i$ is the donor and the matched recipient $j$ has never been publicly signaled as "defect," then $i$ cooperates; otherwise, $i$ defects forever against $j$ (grim trigger (Axelrod & Hamilton, 1981)).

- *Gossip policy $\phi_i^*$.* If $i$ is the recipient at time $t$, then $i$ broadcasts the public message $m_i^t = a_j^t$, i.e., $i$ truthfully reports the donor $j$'s action.

Assume all agents adopt the same joint strategy $s_i^* = s_j^*$ for all $i, j \in \mathcal{N}$. We claim that the joint profile $(s_i^*, s_{-i}^*)$ is a subgame-perfect equilibrium if $\gamma \geq \frac{c}{b}$. To prove this, we need to prove $\forall s_i' \neq s_i^*$ cannot strictly improve agent $i$'s expected discounted return at any public history $h^t$.

Now, we prove $\forall \phi_i \neq \phi_i^*$, agent $i$ has no incentive to deviate from cooperating when $i$ is the donor at any public history $h^t$.

First, we assume a public history $h^t$ with no past "defect" messages about $i$. *Donor's incentive.* According to one-shot deviation priciple (Hendon et al., 1996), and let $i$ be the current donor matched with recipient $j$. Since all recipients follow the honest gossip policy $\phi^*$, the public signal truthfully reflects $i$'s action. Then, similar to the proof in Section A.3, the grim-trigger strategy ensures that deviating by defecting yields 0 forever, while cooperating yields the alternating stream $-c, b, -c, b, \ldots$, whose expected discounted return is $\frac{\gamma b - c}{1 - \gamma^2}$. Therefore, by the same reasoning as before, the donor has no incentive to deviate from cooperating if $\gamma \geq \frac{c}{b}$.

Since agent $i$ has no incentive to deviate from cooperating when $i$ is the donor given any gossip policy, then we prove agent $i$ has no incentive to deviate from $\phi_i^*$ when $i$ is the recipient under this condition.

*Recipient's incentive.* Now consider agent $i$ as a recipient. By construction of $\pi^*$, the donor's future behavior depends only on whether $i$ is ever publicly signaled as "defect." Therefore, $i$'s own payoff is independent of the gossip policy. Thus the recipient cannot strictly improve her expected discounted return by deviating from $\phi_i^*$, making honest gossip incentive-compatible.

Next, for any public history $h^t$ with past "defect" messages about $i$, others always defect to $i$ forever when $i$ is the donor. Therefore, by grim trigger, agent $i$'s future expected discounted return is 0 regardless of $i$'s current action or gossip. Hence, $i$ has no incentive to deviate from $s_i^*$ at such public histories.

Thus, for any public history, agent $i$ has no incentive to deviate from $s_i^*$ unilaterally if $\gamma \geq \frac{c}{b}$. Therefore, there exists an SPE of $(s_i^*, s_{-i}^*)$ that sustains cooperation through public gossip if $\gamma \geq \frac{c}{b}$.

$\square$

## B ALIGN DETAILS

Algorithm 1 summarizes the ALIGN framework. At the start of the simulation, a set of agents $\mathcal{N}$ is initialized with the environment $\mathcal{E}$, a common prompt $K$, and a horizon $T$. Each agent $i \in \mathcal{N}$ is associated with an information state $\Theta_i$, which includes the common prompt $K$, its local memory $M_i$, and the public message pool $P$. The common prompt $K$ is the prompt shared by all agents, which provides background knowledge about the environment, the game rules, and the information flow, response format, etc. The agent's memory $M_i$ stores its entire interaction history, while the public message pool $P$ contains all gossip messages generated by all agents.

At each time step, agents are randomly paired into disjoint pairs. The paring rule depends on the specific game setting. For example, in the donation game, agents are paired and assigned roles

---

**Algorithm 1** ALIGN: Agentic Linguistic Gossip Network

---

**Require:** Environment $\mathcal{E}$, agents $\mathcal{N}$, horizon $T$, common prompt $K$

1: $P \leftarrow \varnothing$      ▷ Initialize public message pool
2: **for all** $i \in \mathcal{N}$ **do**
3:      $M_i \leftarrow \varnothing$      ▷ Initialize agent-local memory
4:      $\Theta_i \leftarrow (K, M_i, P)$
5:      Initialize action policy $\pi_i^{\Theta_i}$ and gossip policy $\phi_i^{\Theta_i}$
6:      Initialize reflection module $f_i^{\Theta_i}$
7: **end for**
8: **for** $t = 1$ **to** $T$ **do**
9:      Randomly partition $\mathcal{N}$ into disjoint pairs $(i, j)$
10:      **for all** pairs $(i, j)$ **do**
11:          $\mathcal{E}$ assign roles: *actor* $i$ (takes action), *witness* $j$ (observes and gossips)
12:          $o_i^t \leftarrow \text{OBSERVE}(\mathcal{E}, i)$      ▷ Actor $i$ observes local environment state
13:          $a_i^t \sim \pi_i^{\Theta_i}(\cdot \mid o_i^t)$      ▷ Actor $i$ samples an action from policy
14:          $\rho_i^t \sim f_i^{\Theta_i}(\cdot \mid M_i, P, K)$      ▷ Actor $i$ generates internal reflection based on history and knowledge
15:          $o_j^t \leftarrow \text{OBSERVE}(\mathcal{E}, j)$      ▷ Witness $j$ observes environment and actor's action
16:          $m_j^t \sim \phi_j^{\Theta_j}(\cdot \mid o_j^t, a_i^t)$      ▷ Witness $j$ produces a gossip message conditioned on observation and action
17:          $\rho_j^t \sim f_j^{\Theta_j}(\cdot \mid M_j, P, K)$      ▷ Witness $j$ generates its reflection
18:          $(r_i^t, r_j^t) \leftarrow \text{STEP}(\mathcal{E}, a_i^t)$      ▷ Environment computes rewards; state updated internally
19:          $P \leftarrow P \cup \{(t, j, m_j^t)\}$      ▷ Append gossip message to public pool
20:          $M_i \leftarrow M_i \oplus (o_i^t, a_i^t, m_j^t, r_i^t, \rho_i^t)$      ▷ Update actor's memory with own trajectory
21:          $M_j \leftarrow M_j \oplus (o_j^t, a_i^t, m_j^t, r_j^t, \rho_j^t)$      ▷ Update witness's memory with observed trajectory and gossip
22:      **end for**
23: **end for**

---

of *actor* and *witness* alternately in each round, while in the indirect reciprocity game, agents are paired randomly without role switching, they are both *actor* and *witness* in each round. The actor observes the environment state $o_i^t$ and samples an action $a_i^t$ from its action policy $\pi_i^{\Theta_i}$. The actor also generates an internal reflection $\rho_i^t$ based on its memory, the public message pool, and the common prompt using its reflection module $f_i^{\Theta_i}$. The witness observes the environment state $o_j^t$ and the actor's action $a_i^t$, and produces a gossip message $m_j^t$ using its gossip policy $\phi_j^{\Theta_j}$. The witness also generates its own reflection $\rho_j^t$.

The environment then transitions internally to its next state, which is not fully observed by the agents, and assigns rewards $r_i^t$ and $r_j^t$ to the actor and witness. The gossip message $m_j^t$ is appended to the public message pool $P$. Finally, both agents update their local memories $M_i$ and $M_j$ with their respective observations, actions, gossip messages, rewards, and reflections. This process repeats for $T$ time steps, allowing agents to interact, learn from their experiences, and share information through gossip.

## C    PROMPTS

All prompts in our experiments follow a common structure and can be classified into three categories: rule prompt, action prompt and gossip prompt (Figure 3). The rule prompt provides background information about the game environment, including the game rules, information flow, response format, and other relevant details. The action prompt guides the agent in selecting actions based on its observations and history. The gossip prompt instructs the agent on how to generate gossip messages based on its observations of other agents' actions. We provide prompts for each game below.

## C.1 REPEATED DONATION GAME

### Listing 1: Rule Prompt of Repeated Donation Game

```
## Self-awareness

You are a self-interested, far-sighted, rational agent, playing a
[HORIZON-TYPE]-horizon multi-round donation game.

- Self-interested: your utility function is only your own expected
 payoff.
- Far-sighted: you maximize expected discounted cumulative reward
(discount factor $discount_factor) over all remaining rounds.
- Rational: you always choose the strategy with the highest
expected payoff.

## Overview of the game

- Every agent starts with $initial_resources resource units.
- In each round, two agents are randomly paired without
replacement as donor and recipient; the same pair never meets
twice.
- The donor chooses one of:
    - cooperate: pay $cost, recipient gains $benefit
    - defect: transfer 0 to the recipient
- The recipient observes whether the donor cooperated or defected
and receives $benefit if the donor cooperates, or 0 otherwise.
- Then, the recipient:
    - With gossip: broadcasts a public assessment of the donor's
choice.
    - Without gossip: takes no further action.
- After payoffs, agents are reshuffled and roles may switch.

## Game length

- Infinite horizon: rounds continue indefinitely.
- Finite horizon: the game lasts exactly $horizon_length rounds.

## Immediate reward per round

If the donor cooperates:
- Donor reward = -$cost
- Recipient reward = $benefit

If the donor defects:
- Donor reward = 0
- Recipient reward = 0

## Information flow (only when gossip is enabled)

- All messages are publicly logged and visible to all agents.
- Messages do not directly change payoffs.
- Future agents may condition their behavior on past messages.
```

### Listing 2: Donation Prompt of Repeated Donation Game

```
## Donor Instructions

### Identity and Observation
Your name is $donor_name. In this round, you are the donor, paired
 with recipient $recipient_name. You currently hold
$donor_resources resource units; the recipient holds
$recipient_resources.
```

```
### Objective
Your sole objective is to maximize your expected discounted
cumulative reward over the remaining rounds of a [HORIZON-TYPE]-
horizon game (discount factor $discount_factor; for finite horizon
, total length
$horizon_length).

### Memory
You may recall past interactions. Historical records are provided
as: $stm

### Community Messages
With gossip enabled: You may review the public log of earlier
broadcasts: $historical_messages

Without gossip:
No public messages are available.

### Accountability Notice
With gossip enabled: Your action is not directly observed by other
 agents, but the recipient will publish a message about it. This
message is permanently logged and visible to all future agents.

Without gossip: Your action is not observed by others and no
message will be broadcast.

### Common Knowledge for Subgame Perfect Equilibrium Reasoning
(Included only when equilibrium knowledge is enabled)
- Finite horizon: use backward induction.
- Infinite horizon: apply the one-shot deviation principle.
- A strategy profile is a Subgame Perfect Equilibrium if no
profitable deviation exists.

### Donation Rule
Choose your action as the donor for this round:
- cooperate: pay $cost, recipient gains $benefit
- defect: pay 0, recipient gains 0

### Response Guidelines
- Reflect using Identity and Observation, Memory, and (if
available) Community Messages.
- Evaluate which action maximizes your objective.
- Ask: "Would deviating at this step improve my total expected
payoff?"
- Provide your action and a short justification.

If equilibrium knowledge is enabled, additionally justify why your
 choice is consistent with Subgame Perfect Equilibrium reasoning.

Return JSON only in the following format:

{
"justification": "a short explanation",
"donor_action": "cooperate" or "defect"
}
```

Listing 3: Gossip Prompt of Repeated Donation Game

```
## Recipient Instructions

### Identity and Observation
Your name is $recipient_name. In this round, you are the recipient
, and you are paired with donor $donor_name.
```

```
     Before the donation, you held $recipient_resources resource units;
      the matched donor $donor_name held $donor_resources. The donor
     paid a personal cost of $donation ($donation_ratio of the donor's
      current stock) to transfer $benefit units to you.

     ### Objective
     Your sole objective is to maximize your expected discounted
     cumulative reward ver the remaining rounds of a [HORIZON-TYPE]-
     horizon game (discount factor $discount_factor; for finite horizon
     , total length $horizon_length).

     ### Memory
     You can recall your interaction history of past rounds. Historical
      records are: $stm

     ### Community Messages
     You can review the public log of earlier donation broadcasts and
     track the past
     behavior of your current donor to judge their trustworthiness:
     $historical_messages

     ### Accountability Notice
     The donor's action is not directly observed by other agents.
     However, you will publish a message about it. This message is
     permanently logged and visible to all future agents and may affect
      how others treat the donor later.

     ### Common Knowledge for Finding Subgame Perfect Equilibria
     (Included only when equilibrium knowledge is enabled.)
     - Finite-horizon games: use backward induction. Start from the
     last round and determine the optimal actions there, then move
     backward step by step.
     - Infinite-horizon games: use the one-shot deviation principle.
     At any round, imagine deviating from the planned strategy for just
      one step and ask whether this deviation increases total expected
     payoff. If no profitable one-shot deviation exists for any player,
      the strategy profile is a Subgame Perfect Equilibrium.

     ### Gossip Rule
     Now, choose exactly one tone from the list [praising, neutral,
     mocking, complaint, criticism] and write a public message about
     the donor's action using the selected tone.

     - praising: express gratitude and positive evaluation
     - neutral: provide an objective, matter-of-fact assessment
     - mocking: use sarcastic ridicule
     - complaint: convey mild disappointment
     - criticism: deliver a harsh negative evaluation

     ### Response Guidelines
     - Reflect using Identity and Observation, Memory, and Community
     Messages to judge whether the donor is trustworthy and to avoid
     being exploited.
     - Reflect on what action (choice of tone and message) will
     maximize your objective.
     - Ask yourself: "Would deviating at this step improve my total
     expected payoff?"
     - After reflection, choose a tone and write a concise public
     message to the community.

     If equilibrium knowledge is enabled, additionally:
     - Reflect using the Common Knowledge for Finding Subgame Perfect
     Equilibria provided above.
     - Explain how your chosen tone and message align with equilibrium
     reasoning.
```

```
    Return JSON only in the following format:

    {
    "justification": "a short explanation of how your choice follows
    from reflecting on Subgame Perfect Equilibrium reasoning",
    "tone": "one of {'praising', 'neutral', 'mocking', 'complaint', '
    criticism'}",
    "gossip": "a concise public message to the community (less than
    150 words)"
    }
```

## C.2 INDIRECT RECIPROCITY GAME

As shown in table 2, the payoff structure of the one-shot indirect reciprocity game is identical to the prisoner's dilemma (Rapoport, 1965). According to (Ohtsuki & Iwasa, 2006; 2004), the indirect reciprocity game is a multi-round prisoner's dilemma game where agents are randomly paired without replacement in each round so that the same pair never meets twice. Therefore, we refer the indirect reciprocity game as the multi-round prisoner's dilemma game in our prompts. Additionally, agents in the indirect reciprocity game do not have fixed roles of donor and recipient; instead, both agents simultaneously choose to cooperate or defect and then broadcast gossip messages about each other's actions. The rule prompt and action prompt for the multi-round prisoner's dilemma game are provided below.

Listing 4: Rules Prompt of Indirect Reciprocity Game (Multi-Round Prisoner's Dilemma)

```
## Self-awareness

You are a self-interested, far-sighted, rational agent, playing a
[HORIZON-TYPE]-horizon multi-round prisoner's dilemma game.

- Self-interested: your utility function is only your own expected
    payoff. You may track, predict, or even raise other agents'
    payoffs, but only when doing so increases your own expected return
    . Other agents' welfare carries no intrinsic weight; it matters
    only through its effect on your future rewards.
- Far-sighted: you aim to maximize your expected discounted cumulative
     reward (discount factor $discount_factor) over all remaining
    rounds, not just the immediate reward.
- Rational: you always choose the strategy with the highest expected
    payoff.

## Overview of the game

- You are playing a multi-round prisoner's dilemma game with [HORIZON-
    TYPE] horizon.
- In each round, two agents are randomly paired without replacement;
    the same pair never meets twice across all rounds.
- Each player chooses one of:
  - cooperate: pay fitness cost $cost
  - defect: pay no cost
- After both players act, each player observes the opponent's action
    and then:
  - With gossip enabled: broadcasts a public assessment of the
    opponent's choice; this message is permanently logged and visible
    to all agents.
  - Without gossip: does nothing (no message is broadcast).
- After payoffs are realized, agents are reshuffled and paired with
    new opponents in the next round.

## Game length
```

```
1080    - Infinite horizon: this is an infinite-horizon prisoner's dilemma
1081        game. Rounds continue indefinitely without fixed termination; the
1082        expected horizon is infinite.
1083    - Finite horizon: this is a finite-horizon prisoner's dilemma game of
1084        exactly $horizon_length rounds. After round $horizon_length, play
1085        stops and no future rewards are possible. Termination is fixed and
1086         independent of history.
1087
1088    ## Immediate reward per round
1089
1090    - If you cooperate and the opponent cooperates:
1091      - Your reward = $benefit - $cost
1092      - Opponent reward = $benefit - $cost
1093    - If you cooperate and the opponent defects:
1094      - Your reward = -$cost
1095      - Opponent reward = $benefit
1096    - If you defect and the opponent cooperates:
1097      - Your reward = $benefit
1098      - Opponent reward = -$cost
1099    - If you defect and the opponent defects:
1100      - Your reward = 0
1101      - Opponent reward = 0
1102
1103    Note: The benefit is larger than the cost ($benefit > $cost), so
1104        mutual cooperation yields the highest total payoff.
1105
1106    ## Information flow (only when gossip is enabled)
1107
1108    - All messages are publicly logged and visible to all agents
1109        throughout the game.
1110    - Messages do not directly change anyone's payoffs.
1111    - Future agents can read past messages, which may influence their
1112        actions.
```

Listing 5: Action Prompt of Indirect Reciprocity Game (Multi-Round Prisoner's Dilemma)

```
## Player Instructions - Action Phase

### Identity and Observation
Your name is $player_name. In this round, you are paired with opponent
$opponent_name.

This is the action phase where you choose to cooperate or defect.

### Objective
Your sole objective is to maximize your expected discounted cumulative
    reward over the remaining rounds of a [HORIZON-TYPE]-horizon game
    (discount factor $discount_factor; for finite horizon, total
    length $horizon_length).

### Memory
You can recall your interaction history of past rounds. Historical
    records are: $stm

### Community Messages
With gossip enabled: You can review the public log about earlier
    prisoner's dilemma broadcasts and track the past behavior of your
    current opponent to judge their trustworthiness:
    $historical_messages

Without gossip: No public messages are available.

### Accountability Notice
With gossip enabled: Your action is not directly observed by other
    agents. However, your opponent will publish a message about it in
```

```
      the gossip phase. This message is permanently logged and visible
      to all future agents and may affect how others treat you later.

Without gossip:
Your action is not directly observed by other agents and no message
    will be broadcast.

### Common Knowledge for Finding Subgame Perfect Equilibria
(Included only when equilibrium knowledge is enabled.)
- Finite-horizon games: use backward induction. Start from the last
    round and determine the optimal actions there, then move backward
    step by step, choosing strategies that remain optimal given future
     play.
- Infinite-horizon games: use the one-shot deviation principle. At any
     round, imagine deviating from the planned strategy for just one
    step and ask whether this deviation increases your total expected
    payoff. If no profitable one-shot deviation exists for any player,
     the strategy profile is a Subgame Perfect Equilibrium.

### Prisoner's Dilemma Rule
Now, choose your action for this round.

You may choose either:
- C: cooperate (pay fitness cost $cost)
- D: defect (pay no cost)

Your payoff depends on both your choice and your opponent's choice:
- If both cooperate: you get $benefit - $cost
- If you cooperate and your opponent defects: you get -$cost
- If you defect and your opponent cooperates: you get $benefit
- If both defect: you get 0

### Response Guidelines
- Reflect using Identity and Observation, Memory, and (if available)
    Community Messages, judging whether the opponent is trustworthy to
     avoid being exploited.
- Reflect on which action will maximize your objective.
- Ask: "Would deviating at this step improve my total expected payoff
    ?"
- After reflection, provide your action and a short explanation.

If equilibrium knowledge is enabled, additionally justify why your
    choice is consistent with Subgame Perfect Equilibrium reasoning.

Return JSON only in the following format:

{
  "justification": "a short explanation of how your choice follows
    from reflecting on Subgame Perfect Equilibrium reasoning",
  "player_action": "C" or "D"
}
```

Listing 6: Gossip Prompt of Indirect Reciprocity Game (Multi-Round Prisoner's Dilemma)

```
## Player Instructions - Gossip Phase

### Identity and Observation
Your name is $player_name. In this round, you are paired with opponent
$opponent_name. Your opponent chose to $opponent_action (cooperate or
    defect) in this round.

This is the gossip phase where you choose how to publicly assess your
opponent's action.

### Objective
```

```
Your sole objective is to maximize your expected discounted cumulative
    reward over the remaining rounds of a [HORIZON-TYPE]-horizon game
    (discount factor $discount_factor; for finite horizon, total
    length $horizon_length).

### Memory
You can recall your interaction history of past rounds. Historical
    records are:
$stm

### Community Messages
You can review the public log about earlier prisoner's dilemma
    broadcasts and track the past behavior of your current opponent to
    judge their trustworthiness: $historical_messages

### Accountability Notice
Your opponent's action is not directly observed by other agents.
    However, you will publish a message about it. This message is
    permanently logged and visible to all future agents and may affect
    how others treat your opponent later.

### Common Knowledge for Finding Subgame Perfect Equilibria
(Included only when equilibrium knowledge is enabled.)
- Finite-horizon games: use backward induction.
- Infinite-horizon games: use the one-shot deviation principle.
- A strategy profile is a Subgame Perfect Equilibrium if no profitable
  deviation exists.

### Gossip Rule
Now, choose exactly one tone from the list: [praising, neutral,
    mocking, complaint, criticism] and write a public message about
    your opponent's action using the selected tone.

- praising: express gratitude and positive evaluation
- neutral: provide an objective, matter-of-fact assessment
- mocking: use sarcastic ridicule
- complaint: convey mild disappointment
- criticism: deliver a harsh negative evaluation

### Response Guidelines
- Reflect using Identity and Observation, Memory, and Community
    Messages, judging whether the opponent is trustworthy and how your
     message may shape future interactions.
- Reflect on what tone and message will maximize your objective.
- Ask: "Would deviating at this step improve my total expected payoff
    ?"
- After reflection, choose a tone and write a concise public message.

If equilibrium knowledge is enabled, additionally justify how your
    chosen tone and message align with Subgame Perfect Equilibrium
    reasoning.

Return JSON only in the following format:

{
  "justification": "a short explanation of how your choice follows
    from reflecting on Subgame Perfect Equilibrium reasoning",
  "tone": "one of {'praising', 'neutral', 'mocking', 'complaint', '
    criticism'}",
  "gossip": "a concise public message to the community (less than 150
    words)"
}
```

## C.3 INVESTMENT GAME

To model the multi-round investment (trust) game, we adapt the standard one-shot trust game (Berg et al., 1995) into a sequential setting where agents are randomly paired without replacement in each round, ensuring that the same pair never meets twice. In each round, one agent is assigned the role of investor (first mover) and the other as responder (second mover). The investor chooses an amount to invest in the responder; this amount is then multiplied by a factor (typically set to 3) and transferred to the responder. The responder then decides how much to return to the investor. After both actions are taken and payoffs are realized, both agents observe each other's actions and broadcast gossip messages evaluating the partner's behavior. Agents' resources are updated after each round, so the global state evolves over time. Both roles have continuous action spaces whose feasible ranges depend on their current resources. Cooperation can emerge because the investor and responder can mutually transfer resources, and indirect reciprocity arises as agents' evaluative messages propagate through the population, including their potential future partners. The rule prompt, investor action prompt, responder action prompt, investor gossip prompt, and responder gossip prompt used in this game are provided below.

Listing 7: Rule Prompt of Multi-Round Investment Game

```
## Self-awareness

You are a self-interested, far-sighted, rational agent, playing a
    [HORIZON-TYPE]-horizon multi-round trust game.

- Self-interested: your utility function is only your own expected
    payoff. You may track, predict, or even increase other agents'
    payoffs, but only when doing so increases your own expected return
    . Other agents' welfare carries no intrinsic weight; it matters
    only through its effect on your future rewards.
- Far-sighted: you aim to maximize your expected discounted cumulative
    reward (discount factor $discount_factor) over all remaining
    rounds, not just the immediate reward.
- Rational: you always choose the strategy with the highest expected
    payoff.

## Overview of the Game

You are playing a multi-round trust game with [HORIZON-TYPE] horizon.

- Every agent starts with $initial_resources resource units.
- In each round, two agents are randomly paired without replacement as
    :
    - Investor (first mover)
    - Responder (second mover)
    The same pair never meets twice across all rounds.

Stage game per round:
1. The investor observes their own and the responder's current
    resources.
2. The investor chooses an investment amount I in [0,
    current_resources].
3. The investment I is multiplied by $investment_multiplier and
    transferred to the responder.
4. The responder chooses a return amount R in [0, I *
    $investment_multiplier] to send back to the investor.
5. Both players' payoffs for the round are realized.
6. Both players observe each other's actions in this round.
7. The investor then:
    - With gossip enabled: observes the responder's return and
    broadcasts a public message about the responder's behavior this
    round.
    - Without gossip: observes the responder's return; no public
    message is sent.
8. The responder then:
```

```
        - With gossip enabled: observes the investor's investment and
          broadcasts a public message about the investor's behavior this
          round.
        - Without gossip: observes the investor's investment; no public
          message is sent.

  - With gossip enabled: both agents send one public message per round (
      one from the investor and one from the responder). These two
      messages are permanently logged and visible to all agents.
  - Without gossip: no public gossip is allowed; agents only privately
      observe each other's actions.

  After payoffs (and any messages) are processed, agents are reshuffled
      and roles may switch in later rounds (an agent who was an investor
       in one round may be a responder in a later round, and vice versa)
       .

  ## Game Length

  - Infinite horizon:
    - This is an infinite-horizon trust game.
    - Rounds continue indefinitely without fixed termination; the
      expected horizon is infinite.
  - Finite horizon:
    - This is a finite-horizon trust game of exactly $horizon_length
      rounds.
    - After round $horizon_length, play stops; no future rewards are
      possible.
    - Termination is fixed and independent of history.

  ## Immediate Reward Per Round (Standard Trust Game)

  Let I be the amount the investor chooses to invest, and let R be the
      amount the responder chooses to return.

  - The investment I is multiplied by $investment_multiplier and added
      to the responder's resources.
  - The responder then chooses a return amount R in [0, I *
      $investment_multiplier].

  Investor reward this round:
  - The investor loses I but receives R.
  - Net payoff change from this round: -I + R.

  Responder reward this round:
  - The responder gains I * $investment_multiplier but gives back R.
  - Net payoff change from this round: I * $investment_multiplier - R.

  ## Information flow and Gossip (only when gossip is enabled)

  - At the end of each round, after both actions and payoffs:
    - The investor observes the responder's return decision.
    - The responder observes the investor's investment decision.
  - Each agent can then broadcast one public message about their
      coplayer's behavior in that round:
    - one message from the investor about the responder,
    - one message from the responder about the investor.
  - All messages are publicly logged and visible to all agents
      throughout the game.
  - Messages do not directly change anyone's payoffs.
  - Future agents can read past messages, which may influence their
      actions.
```

Listing 8: Investor Action Prompt of Multi-Round Investment Game

```
## Investor Instructions

### Identity and Observation
Your name is $investor_name. In this round, you are the investor (
    first mover), paired with responder $responder_name. You currently
    hold $investor_resources resource units; the responder holds
    $responder_resources. As the investor, you will choose how much to
    trust the responder by investing part of your resources.

### Objective
Your sole objective is to maximize your expected discounted cumulative
    reward over the remaining rounds of a [HORIZON-TYPE]-horizon
    trust game (discount factor $discount_factor; for finite horizon,
    total length $horizon_length).

### Memory
You can recall your interaction history of past rounds. Historical
    records are: $stm

### Community Messages
With gossip enabled: You can review the public log of earlier trust-
    game gossip messages, tracking the past behavior of your current
    responder to judge their trustworthiness: $historical_messages

Without gossip: No public gossip messages are available.

### Accountability Notice
With gossip enabled: At the end of each round, after you and your
    coplayer observe each other's actions, both of you can broadcast a
    public message about your coplayer's behavior.
- Your coplayer (the responder) may broadcast a message about you.
- You may also broadcast a message about them when asked.
All messages are permanently logged and visible to all future agents
    and may affect how others treat you later.

Without gossip: Your investment decision and the responder's return
    are not publicly discussed; no gossip messages will be broadcast
    about this round.

### Common Knowledge for Finding Subgame Perfect Equilibria
(Included only when equilibrium knowledge is enabled.)
- Finite-horizon games: use backward induction.
- Infinite-horizon games: use the one-shot deviation principle.
- A strategy profile is a Subgame Perfect Equilibrium if no profitable
    one-shot deviation exists for any player.

### Trust Action Rule
Now, choose your investment action as the investor for this round.

You may invest any real amount I between: 0 and $investor_resources.

### Response Guidelines
- Reflect using Identity and Observation, Memory, and (if available)
  Community Messages, judging whether the responder is trustworthy to
    avoid being exploited.
- Reflect on what investment amount will maximize your objective.
- Ask: "Would deviating at this step improve my total expected payoff
    ?"
- After reflection, provide your action and a short explanation.

If equilibrium knowledge is enabled, additionally justify why your
    choice is consistent with Subgame Perfect Equilibrium reasoning in
     the trust game.
```

```
Return JSON only in the following format:

{
  "justification": "a short explanation of how your choice follows
    from reflecting on strategic (and, if applicable, Subgame Perfect
    Equilibrium) reasoning in the trust game",
  "investor_action": "a real number between 0 and $investor_resources
    representing how much you invest"
}
```

Listing 9: Responder Action Prompt of Multi-Round Investment Game

```
## Responder Instructions

### Identity and Observation
Your name is $responder_name. In this round, you are the responder (
    second mover), paired with investor $investor_name. Before the
    investment, you held $responder_resources resource units; the
    investor held $investor_resources. The investor invested
    $investment (this equals $investment_ratio of the investor's
    current stock), which was multiplied to $benefit units and
    transferred to you. You now choose how much to return to the
    investor in this round.

### Objective
Your sole objective is to maximize your expected discounted cumulative
    reward over the remaining rounds of a [HORIZON-TYPE]-horizon
    trust game (discount factor $discount_factor; for finite horizon,
    total length $horizon_length).

### Memory
You can recall your interaction history of past rounds. Historical
    records are: $stm

### Community Messages
With gossip enabled: You can review the public log about earlier
    gossip in the trust game, tracking the past behavior of your
    current investor to judge their trustworthiness:
    $historical_messages

Without gossip: No public gossip messages are available.

### Accountability Notice
With gossip enabled: At the end of each round, after you and the
    investor observe each other's actions, both of you can broadcast a
     public message about your coplayer's behavior.
- The investor may broadcast a message about you.
- You may also broadcast a message about them when asked.
These messages are permanently logged and visible to all future agents
     and may affect how others treat you later. When you choose how
    much to return, you may anticipate the effect of future gossip on
    your long-run payoff.

Without gossip: Your return decision and the investor's investment are
     not publicly discussed; no gossip messages will be broadcast
    about this round.

### Common Knowledge for Finding Subgame Perfect Equilibria
(Included only when equilibrium knowledge is enabled.)
- Finite-horizon games: use backward induction.
- Infinite-horizon games: use the one-shot deviation principle.
- A strategy profile is a Subgame Perfect Equilibrium if no profitable
  one-shot deviation exists for any player.

### Return Action Rule
```

```
Now, choose your return amount as the responder for this round.

The investor's investment was multiplied to $benefit units and added
    to your resources. You may return any real amount R between: 0 and
     $benefit.

### Response Guidelines
- Reflect using Identity and Observation, Memory, and (if available)
  Community Messages, judging how your return choice today affects:
  - your immediate payoff, and
  - others' future treatment of you (especially under gossip).
- Reflect on what return amount will maximize your objective.
- Ask: "Would deviating at this step improve my total expected payoff
    ?"
- After reflection, provide your action and a short explanation.

If equilibrium knowledge is enabled, additionally justify why your
    choice is consistent with Subgame Perfect Equilibrium reasoning in
     the trust game.

Return JSON only in the following format:

{
  "justification": "a short explanation of how your choice follows
    from reflecting on strategic (and, if applicable, Subgame Perfect
    Equilibrium) reasoning in the trust game",
  "responder_action": "a real number between 0 and $benefit
    representing how much you return to the investor"
}
```

Listing 10: Investor Gossip Prompt of Multi-Round Investment Game

```
## Investor Gossip Instructions

### Identity and Observation
Your name is $investor_name. In this round, you were the investor and
    you were paired with responder $responder_name.

- You invested $investment units (this equals $investment_ratio of
    your current stock).
- This investment was multiplied into $benefit units and transferred
    to the responder.
- The responder returned $returned_amount units to you (this equals
    $returned_ratio of the transferred benefit $benefit).

You have fully observed:
- how much you invested,
- the multiplied amount you transferred,
- and the responder's actual return decision in this round.

### Objective
Your sole objective is to maximize your expected discounted cumulative
     reward over the remaining rounds of a [HORIZON-TYPE]-horizon
    trust game (discount factor $discount_factor; for finite horizon,
    total length $horizon_length).

### Memory
You can recall your interaction history of past rounds. Historical
    records are: $stm

### Community Messages
You can review the public log about earlier gossip in the trust game,
    tracking the past behavior of your current responder and other
    agents: $historical_messages
```

```
### Accountability Notice
At the end of each round, after you and the responder observe each
    other's actions, both of you can broadcast a public message about
    your coplayer's behavior.
- The responder can broadcast a message about you.
- You will now broadcast a message about them.
Your message is permanently logged and visible to all future agents
    and may affect how others treat both you and your coplayer.

### Common Knowledge for Finding Subgame Perfect Equilibria
(Included only when equilibrium knowledge is enabled.)
- Finite-horizon games: use backward induction.
- Infinite-horizon games: use the one-shot deviation principle.
- A strategy profile is a Subgame Perfect Equilibrium if no profitable
  one-shot deviation exists for any player.

### Gossip Rule (Investor)
You have already observed the responder's action in this trust game
    round:
- your own investment $investment (ratio $investment_ratio of your
    stock),
- the multiplied benefit $benefit transferred to the responder,
- the responder's returned amount $returned_amount (ratio
    $returned_ratio of $benefit).

Now, choose exactly one tone from the list: [praising, neutral,
    mocking, complaint, criticism] and write a public message about
    the responder's behavior using the selected tone.

- praising: express gratitude and positive evaluation
- neutral: provide an objective, matter-of-fact assessment
- mocking: use sarcastic ridicule
- complaint: convey mild disappointment
- criticism: deliver a harsh negative evaluation

### Response Guidelines
- Reflect using Identity and Observation, Memory, and Community
    Messages,
  judging how your gossip may influence:
  - others' beliefs about this responder (given $investment, $benefit,
    $returned_amount, $returned_ratio),
  - and your own future payoffs through reputational effects.
- Reflect on what gossip tone and content will maximize your objective
    .
- Ask: "Would deviating at this step improve my total expected payoff
    ?"
- After reflection, choose a tone and write a concise message.

If equilibrium knowledge is enabled, additionally justify how your
    choice aligns with Subgame Perfect Equilibrium reasoning in the
    trust game.

Return JSON only in the following format:

{
  "justification": "a short explanation of how your choice follows
    from reflecting on strategic reasoning (and, if applicable,
    Subgame Perfect Equilibrium reasoning) in the trust game",
  "tone": "one of {'praising', 'neutral', 'mocking', 'complaint', '
    criticism'}",
  "gossip": "a concise public message to the community (less than 150
    words)"
}
```

Listing 11: Responder Gossip Prompt of Multi-Round Investment Game

```
## Responder Gossip Instructions

### Identity and Observation
Your name is $responder_name. In this round, you were the responder
    and you were paired with investor $investor_name.

- Before the round, the investor held $investor_resources resource
    units.
- The investor invested $investment units (this equals
    $investment_ratio of their current stock).
- This investment was multiplied into $benefit units and transferred
    to you.
- You returned $returned_amount units to the investor (this equals
    $returned_ratio of the transferred benefit $benefit).

You have fully observed:
- how much the investor chose to invest,
- the multiplied amount $benefit you received,
- and your own return decision in this round.

### Objective
Your sole objective is to maximize your expected discounted cumulative
     reward over the remaining rounds of a [HORIZON-TYPE]-horizon
    trust game (discount factor $discount_factor; for finite horizon,
    total length $horizon_length).

### Memory
You can recall your interaction history of past rounds. Historical
    records are:
$stm

### Community Messages
You can review the public log about earlier gossip in the trust game,
    tracking the past behavior of your current investor and other
    agents: $historical_messages

### Accountability Notice
At the end of each round, after you and the investor observe each
    other's actions, both of you can broadcast a public message about
    your coplayer's behavior.
- The investor can broadcast a message about you.
- You will now broadcast a message about them.
Your message is permanently logged and visible to all future agents
    and may affect how others treat both you and your coplayer.

### Common Knowledge for Finding Subgame Perfect Equilibria
(Included only when equilibrium knowledge is enabled.)
- Finite-horizon games: use backward induction.
- Infinite-horizon games: use the one-shot deviation principle.
- A strategy profile is a Subgame Perfect Equilibrium if no profitable
  one-shot deviation exists for any player.

### Gossip Rule (Responder)
You have already observed the investor's action in this trust game
    round:
- the investor's investment $investment (ratio $investment_ratio of
    their stock),
- the multiplied benefit $benefit that you received,
- and your own return $returned_amount (ratio $returned_ratio of
    $benefit).
```

```
Now, choose exactly one tone from the list: [praising, neutral,
    mocking, complaint, criticism] and write a public message about
    the investor's behavior using the selected tone.
- praising: express gratitude and positive evaluation
- neutral: provide an objective, matter-of-fact assessment
- mocking: use sarcastic ridicule
- complaint: convey mild disappointment
- criticism: deliver a harsh negative evaluation

### Response Guidelines
- Reflect using Identity and Observation, Memory, and Community
    Messages, and judge how generous or exploitative the investor's
    behavior was and how your gossip may influence future play.
- Reflect on what gossip tone and content will maximize your objective
    .
- Ask: "Would deviating at this step improve my total expected payoff
    ?"
- After reflection, choose a tone and write a concise public message.

If equilibrium knowledge is enabled, additionally justify how your
    chosen tone and message align with Subgame Perfect Equilibrium
    reasoning in the trust game.

Return JSON only in the following format:

{
  "justification": "a short explanation of how your choice follows
    from reflecting on strategic (and, if applicable, Subgame Perfect
    Equilibrium) reasoning in the trust game",
  "tone": "one of {'praising', 'neutral', 'mocking', 'complaint', '
    criticism'}",
  "gossip": "a concise public message to the community (less than 150
    words)"
}
```

# D  SUPPLEMENTARY EXPERIMENTS

## D.1  DONATION GAME

**How LLM Reasoning Shapes Cooperation in ALIGN Agents?**   We analyze the reflective text generated by ALIGN agents to examine how they reason about action. Figure 9 presents reflections from DeepSeek-V3.1 Reasoner and Gemini-2.5 Flash-Lite. Cooperative agents highlight reputation, trust, and long-run payoffs; they note that cooperation builds reputation, which in turn promotes reciprocal cooperation. By contrast, non-cooperative agents reason myopically, focus on immediate payoffs, emphasize the absence of direct reciprocity, and overlook that indirect reciprocity can arise via public gossip. These observations indicate that long-horizon reasoning and social awareness are key to leveraging gossip to sustain cooperation.

**Finite-horizon Donation Game**   In finite-horizon settings, cooperation is not an SPE (Proposition 1). Without gossip (Table 5), cooperation is almost entirely absent, except for GPT-4o Mini with 23%. With public gossip, reasoning-focused LLMs remain mostly non-cooperative. In contrast, some chat LLMs reach high cooperation ratios, yielding higher average rewards and low Gini coefficients, which indicate that many agents obtain high rewards in the roughly same level (Table 6). The indirect reciprocity game also shows similar patterns (Appendix D.2).

**Ablation of Equilibrium Knowledge**   In our main experiments, ALIGN agents were given descriptions of backward induction (Von Neumann & Morgenstern, 1947) and one-shot deviation principles (Hendon et al., 1996) for finding an SPE. To assess their impact, we removed these descriptions and re-evaluated performance in infinite-horizon donation games with gossip. As shown in Table 7, DeepSeek-V3.1 Reasoner and o4-mini maintained perfect cooperation and optimal wel-

| Donor's Reflection (DeepSeek-V3.1 Reasoner): |
|---|
| With a high discount factor (0.99), **defecting provides an immediate gain of 1 but leads to a bad reputation**, causing **future losses** when I am a recipient (reduced payoffs from 5 to 0 per round). **Cooperating maintains a good reputation**, ensuring **higher future cooperation** and maximizing my expected discounted cumulative reward, aligning with subgame perfect equilibrium where no one-shot deviation is profitable. |
| [Donor's Action: Cooperate] |

| Donor's Reflection (Gemini 2.5 Flash-Lite): |
|---|
| Since this is an infinite-horizon game and I am a rational, self-interested agent focused on maximizing my long-term discounted cumulative reward, the optimal strategy in the absence of any history or reputation building is to defect. **Defecting yields an immediate payoff of 0, whereas cooperating yields -1**. In an infinite game with no mechanism for future retaliation or reward based on current actions (due to lack of history and **no guaranteed future interactions with the same partner**), the immediate best payoff is preferred as it directly contributes to the cumulative sum. Any cooperation would represent **a short-term loss for no discernible long-term gain** in this specific setup. |
| [Donor's Action: Defect] |

Figure 9: **Examples of Reflections from Donor Agents:** Reflections from DeepSeek-V3.1 Reasoner and Gemini-2.5 Flash-Lite show different reasoning patterns. Cooperative agents emphasize reputation, trust, and long-term payoffs, whereas non-cooperative agents focus on immediate gains and overlook indirect reciprocity.

Table 5: Benchmark results of **non-gossiping agents** across LLMs in the **finite-horizon donation game**. Metrics marked with ↓, indicating that lower values are more aligned with the game-theoretic SPE of defection.

| Agent Type | Cooperation Ratio (↓) | Image Score (↓) | Reward Per Round (↓) | Discounted Return (↓) | Gini Coefficient |
|---|---|---|---|---|---|
| **Chat Models** | | | | | |
| **DeepSeek-V3.1 Chat** | $0.00 \pm 0.00$ | $-4.00 \pm 0.00$ | $0.00 \pm 0.00$ | $0.00 \pm 0.00$ | $0.00 \pm 0.00$ |
| GPT-4o Mini | $0.23 \pm 0.12$ | $-2.20 \pm 0.93$ | $0.90 \pm 0.47$ | $3.55 \pm 1.84$ | $0.37 \pm 0.15$ |
| **Gemini 2.5 Flash-Lite** | $0.00 \pm 0.00$ | $-4.00 \pm 0.00$ | $0.00 \pm 0.00$ | $0.00 \pm 0.00$ | $0.00 \pm 0.00$ |
| **LLaMA 4 Maverick** | $0.00 \pm 0.00$ | $-4.00 \pm 0.00$ | $0.00 \pm 0.00$ | $0.00 \pm 0.00$ | $0.00 \pm 0.00$ |
| **Reasoning Models** | | | | | |
| **Kimi-K2-Instruct** | $0.00 \pm 0.00$ | $-4.00 \pm 0.00$ | $0.00 \pm 0.00$ | $0.00 \pm 0.00$ | $0.00 \pm 0.00$ |
| **DeepSeek-V3.1 Reasoner** | $0.00 \pm 0.00$ | $-4.00 \pm 0.00$ | $0.00 \pm 0.00$ | $0.00 \pm 0.00$ | $0.00 \pm 0.00$ |
| **Qwen3-235B-Instruct** | $0.00 \pm 0.00$ | $-4.00 \pm 0.00$ | $0.00 \pm 0.00$ | $0.00 \pm 0.00$ | $0.00 \pm 0.00$ |
| **o4-mini** | $0.00 \pm 0.00$ | $-4.00 \pm 0.00$ | $0.00 \pm 0.00$ | $0.00 \pm 0.00$ | $0.00 \pm 0.00$ |

Table 6: Benchmark results of **ALIGN agents** across LLMs in the **finite-horizon donation game**. Metrics marked with ↓, indicating that lower values are more aligned with the game-theoretic SPE of defection.

| Agent Type | Cooperation Ratio (↓) | Image Score (↓) | Reward Per Round (↓) | Discounted Return (↓) | Gini Coefficient |
|---|---|---|---|---|---|
| **Chat Models** | | | | | |
| **DeepSeek-V3.1 Chat** | $0.00 \pm 0.00$ | $-4.00 \pm 0.00$ | $0.00 \pm 0.00$ | $0.00 \pm 0.00$ | $0.00 \pm 0.00$ |
| GPT-4o Mini | $0.96 \pm 0.02$ | $3.69 \pm 0.16$ | $1.92 \pm 0.04$ | $14.83 \pm 0.32$ | $0.04 \pm 0.02$ |
| **Gemini 2.5 Flash-Lite** | $0.00 \pm 0.00$ | $-4.00 \pm 0.00$ | $0.00 \pm 0.00$ | $0.00 \pm 0.00$ | $0.00 \pm 0.00$ |
| LLaMA 4 Maverick | $0.54 \pm 0.15$ | $0.33 \pm 1.23$ | $1.08 \pm 0.31$ | $8.37 \pm 2.36$ | $0.34 \pm 0.14$ |
| **Reasoning Models** | | | | | |
| **Kimi-K2-Instruct** | $0.00 \pm 0.00$ | $-4.00 \pm 0.00$ | $0.00 \pm 0.00$ | $0.00 \pm 0.00$ | $0.00 \pm 0.00$ |
| **DeepSeek-V3.1 Reasoner** | $0.00 \pm 0.00$ | $-4.00 \pm 0.00$ | $0.00 \pm 0.00$ | $0.00 \pm 0.00$ | $0.00 \pm 0.00$ |
| **Qwen3-235B-Instruct** | $0.00 \pm 0.00$ | $-4.00 \pm 0.00$ | $0.00 \pm 0.00$ | $0.00 \pm 0.00$ | $0.00 \pm 0.00$ |
| o4-mini | $0.02 \pm 0.01$ | $-3.82 \pm 0.08$ | $0.04 \pm 0.02$ | $0.34 \pm 0.16$ | $0.78 \pm 0.32$ |

fare, indicating that strong reasoning skills suffice to infer cooperative strategies from game structure and gossip alone. By contrast, LLaMA 4 Maverick and Kimi-K2-Instruct showed declines, suggesting reliance on explicit theoretical guidance. Gemini 2.5 Flash-Lite improved without equilibrium knowledge, while Qwen3-235B-Instruct, DeepSeek-V3.1 Chat, and GPT-4o Mini performed similarly across both settings. Overall, these results highlight the nuanced role of equilibrium knowledge: it can support weaker agents but is not essential for models with strong intrinsic reasoning.

Table 7: Ablation of Equilibrium Knowledge: Benchmark results for **ALIGN agents** across LLMs in the **infinite-horizon donation game**. Metrics marked with ↑ indicating that higher values are more desirable; although both cooperation and defection are SPE, higher cooperation yields greater average payoffs.

| Agent Type | Cooperation Ratio (↑) | Image Score (↑) | Reward Per Round (↑) | Discounted Return (↑) | Gini Coefficient |
|---|---|---|---|---|---|
| **Chat Models** | | | | | |
| DeepSeek-V3.1 Chat | $0.98 \pm 0.01$ | $3.85 \pm 0.07$ | $1.96 \pm 0.02$ | $15.14 \pm 0.15$ | $0.03 \pm 0.01$ |
| **GPT-4o Mini** | $1.00 \pm 0.00$ | $4.00 \pm 0.00$ | $2.00 \pm 0.00$ | $15.44 \pm 0.00$ | $0.00 \pm 0.00$ |
| Gemini 2.5 Flash-Lite | $0.91 \pm 0.04$ | $3.28 \pm 0.36$ | $1.82 \pm 0.09$ | $14.01 \pm 0.70$ | $0.06 \pm 0.02$ |
| LLaMA 4 Maverick | $0.58 \pm 0.16$ | $0.61 \pm 1.29$ | $1.15 \pm 0.32$ | $8.84 \pm 2.49$ | $0.35 \pm 0.19$ |
| **Reasoning Models** | | | | | |
| Kimi-K2-Instruct | $0.46 \pm 0.19$ | $-0.33 \pm 1.53$ | $0.92 \pm 0.38$ | $7.06 \pm 2.96$ | $0.38 \pm 0.22$ |
| **DeepSeek-V3.1 Reasoner** | $1.00 \pm 0.00$ | $4.00 \pm 0.00$ | $2.00 \pm 0.00$ | $15.44 \pm 0.00$ | $0.00 \pm 0.00$ |
| Qwen3-235B-Instruct | $0.69 \pm 0.14$ | $1.56 \pm 1.13$ | $1.39 \pm 0.28$ | $10.73 \pm 2.17$ | $0.20 \pm 0.09$ |
| **o4-mini** | $1.00 \pm 0.00$ | $4.00 \pm 0.00$ | $2.00 \pm 0.00$ | $15.44 \pm 0.00$ | $0.00 \pm 0.00$ |

Table 8: Benchmark results of **non-gossiping agents** across LLMs in the **finite-horizon indirect reciprocity game**. Metrics marked with ↓, indicating that lower values are more aligned with the game-theoretic SPE of defection.

| Agent Type | Cooperation Ratio (↓) | Image Score (↓) | Reward Per Round (↓) | Discounted Return (↓) | Gini Coefficient |
|---|---|---|---|---|---|
| **Chat Models** | | | | | |
| **DeepSeek-V3.1 Chat** | $0.00 \pm 0.00$ | $-4.00 \pm 0.00$ | $0.00 \pm 0.00$ | $0.00 \pm 0.00$ | $0.00 \pm 0.00$ |
| GPT-4o Mini | $0.23 \pm 0.12$ | $-2.20 \pm 0.93$ | $0.90 \pm 0.47$ | $3.55 \pm 1.84$ | $0.37 \pm 0.15$ |
| **Gemini 2.5 Flash-Lite** | $0.00 \pm 0.00$ | $-4.00 \pm 0.00$ | $0.00 \pm 0.00$ | $0.00 \pm 0.00$ | $0.00 \pm 0.00$ |
| **LLaMA 4 Maverick** | $0.00 \pm 0.00$ | $-4.00 \pm 0.00$ | $0.00 \pm 0.00$ | $0.00 \pm 0.00$ | $0.00 \pm 0.00$ |
| **Reasoning Models** | | | | | |
| **Kimi-K2-Instruct** | $0.00 \pm 0.00$ | $-4.00 \pm 0.00$ | $0.00 \pm 0.00$ | $0.00 \pm 0.00$ | $0.00 \pm 0.00$ |
| **DeepSeek-V3.1 Reasoner** | $0.00 \pm 0.00$ | $-4.00 \pm 0.00$ | $0.00 \pm 0.00$ | $0.00 \pm 0.00$ | $0.00 \pm 0.00$ |
| **Qwen3-235B-Instruct** | $0.00 \pm 0.00$ | $-4.00 \pm 0.00$ | $0.00 \pm 0.00$ | $0.00 \pm 0.00$ | $0.00 \pm 0.00$ |
| **o4-mini** | $0.00 \pm 0.00$ | $-4.00 \pm 0.00$ | $0.00 \pm 0.00$ | $0.00 \pm 0.00$ | $0.00 \pm 0.00$ |

## D.2 INDIRECT RECIPROCITY GAME

**Finite-horizon Indirect Reciprocity Game** In the finite-horizon indirect reciprocity game, without gossip, all agents defect consistently, aligning with the SPE (Table 8). With public gossip, chat LLMs show mixed cooperation, with GPT-4o Mini achieving high cooperation (96%) and welfare, while others remain non-cooperative. Reasoning LLMs mostly defect, except o4-mini with minimal cooperation (2%) and low welfare (Table 9). These patterns mirror those in the donation game, underscoring the challenges of sustaining cooperation in finite-horizon settings even with gossip, and highlighting the nuanced role of LLM capabilities.

**Infinite-horizon Indirect Reciprocity Game** In the infinite-horizon indirect reciprocity game, without gossip, all agents defect consistently, aligning with the SPE (Table 10). With public gossip, chat LLMs show mixed cooperation, with GPT-4o Mini achieving high cooperation (91%) and welfare, while others remain non-cooperative. Reasoning LLMs mostly defect, except o4-mini with minimal cooperation (2%) and low welfare (Table 11). These patterns mirror those in the donation game, underscoring the challenges of sustaining cooperation in finite-horizon settings even with gossip, and highlighting the nuanced role of LLM capabilities.

Figure 10 presents the distribution of tones in the reflections generated by ALIGN agents in the infinite-horizon indirect reciprocity game with gossip. Cooperative agents (e.g., DeepSeek-V3.1 Reasoner and o4-mini) predominantly exhibit positive and neutral tones, emphasizing trust, reputation, and long-term benefits of cooperation. In contrast, non-cooperative agents (e.g., LLaMA 4 Maverick and Kimi-K2-Instruct) display more negative tones, focusing on immediate payoffs and overlooking the benefits of indirect reciprocity through gossip. These findings suggest that the emotional tone of reflections may correlate with cooperative behavior, where positive and neutral tones align with strategies that foster trust and mutual benefit, while negative tones reflect a more self-interested and short-sighted approach.

**Ablation of Equilibrium Knowledge in Indirect Reciprocity Game** We also evaluated the impact of removing equilibrium knowledge in the infinite-horizon indirect reciprocity game with gossip. As shown in Table 12, DeepSeek-V 3.1 Reasoner and o4-mini maintained perfect cooperation

Table 9: Benchmark results of **ALIGN agents** across LLMs in the **finite-horizon indirect reciprocity game**. Metrics marked with ↓, indicating that lower values are more aligned with the game-theoretic SPE of defection.

| Agent Type | Cooperation Ratio (↓) | Image Score (↓) | Reward Per Round (↓) | Discounted Return (↓) | Gini Coefficient |
|---|---|---|---|---|---|
| **Chat Models** | | | | | |
| **DeepSeek-V3.1 Chat** | $0.00 \pm 0.00$ | $-4.00 \pm 0.00$ | $0.00 \pm 0.00$ | $0.00 \pm 0.00$ | $0.00 \pm 0.00$ |
| GPT-4o Mini | $0.84 \pm 0.10$ | $2.70 \pm 0.79$ | $3.35 \pm 0.39$ | $13.20 \pm 1.55$ | $0.12 \pm 0.07$ |
| Gemini 2.5 Flash-Lite | $0.04 \pm 0.02$ | $-3.70 \pm 0.19$ | $0.15 \pm 0.10$ | $0.59 \pm 0.38$ | $0.54 \pm 0.31$ |
| **LLaMA 4 Maverick** | $0.00 \pm 0.00$ | $-4.00 \pm 0.00$ | $0.00 \pm 0.00$ | $0.00 \pm 0.00$ | $0.00 \pm 0.00$ |
| **Reasoning Models** | | | | | |
| **Kimi-K2-Instruct** | $0.00 \pm 0.00$ | $-4.00 \pm 0.00$ | $0.00 \pm 0.00$ | $0.00 \pm 0.00$ | $0.00 \pm 0.00$ |
| DeepSeek-V3.1 Reasoner | $0.01 \pm 0.01$ | $-3.89 \pm 0.11$ | $0.06 \pm 0.06$ | $0.23 \pm 0.23$ | $0.14 \pm 0.14$ |
| **Qwen3-235B-Instruct** | $0.00 \pm 0.00$ | $-4.00 \pm 0.00$ | $0.00 \pm 0.00$ | $0.00 \pm 0.00$ | $0.00 \pm 0.00$ |
| o4 mini | $0.05 \pm 0.04$ | $-3.60 \pm 0.28$ | $0.20 \pm 0.14$ | $0.79 \pm 0.56$ | $0.55 \pm 0.32$ |

Table 10: Benchmark results for **non-gossiping agents** in the **infinite-horizon indirect reciprocity game**. Metrics marked with ↓ indicate that lower values are more aligned with the game-theoretic SPE of defection.

| Agent Type | Cooperation Ratio (↓) | Image Score (↓) | Reward Per Round (↓) | Discounted Return (↓) | Gini Coefficient |
|---|---|---|---|---|---|
| **Chat Models** | | | | | |
| **DeepSeek-V3.1 Chat** | $0.00 \pm 0.00$ | $-4.00 \pm 0.00$ | $0.00 \pm 0.00$ | $0.00 \pm 0.00$ | $0.00 \pm 0.00$ |
| GPT-4o Mini | $0.91 \pm 0.09$ | $3.30 \pm 0.70$ | $3.65 \pm 0.35$ | $14.38 \pm 1.38$ | $0.07 \pm 0.07$ |
| Gemini 2.5 Flash-Lite | $0.05 \pm 0.05$ | $-3.60 \pm 0.40$ | $0.20 \pm 0.20$ | $0.79 \pm 0.79$ | $0.11 \pm 0.11$ |
| **LLaMA 4 Maverick** | $0.00 \pm 0.00$ | $-4.00 \pm 0.00$ | $0.00 \pm 0.00$ | $0.00 \pm 0.00$ | $0.00 \pm 0.00$ |
| **Reasoning Models** | | | | | |
| **Kimi-K2-Instruct** | $0.00 \pm 0.00$ | $-4.00 \pm 0.00$ | $0.00 \pm 0.00$ | $0.00 \pm 0.00$ | $0.00 \pm 0.00$ |
| **DeepSeek-V3.1 Reasoner** | $0.00 \pm 0.00$ | $-4.00 \pm 0.00$ | $0.00 \pm 0.00$ | $0.00 \pm 0.00$ | $0.00 \pm 0.00$ |
| **Qwen3-235B-Instruct** | $0.00 \pm 0.00$ | $-4.00 \pm 0.00$ | $0.00 \pm 0.00$ | $0.00 \pm 0.00$ | $0.00 \pm 0.00$ |
| **o4-mini** | $0.00 \pm 0.00$ | $-4.00 \pm 0.00$ | $0.00 \pm 0.00$ | $0.00 \pm 0.00$ | $0.00 \pm 0.00$ |

and optimal welfare, indicating that strong reasoning skills suffice to infer cooperative strategies from game structure and gossip alone. By contrast, LLaMA 4 Maverick and Kimi-K2-Instruct showed declines, suggesting reliance on explicit theoretical guidance. Gemini 2.5 Flash-Lite improved without equilibrium knowledge, while Qwen3-235B-Instruct, DeepSeek-V3.1 Chat, and GPT-4o Mini performed similarly across both settings. Overall, these results highlight the nuanced role of equilibrium knowledge: it can support weaker agents but is not essential for models with strong intrinsic reasoning.

**Resilience to Greedy Opponents** Figure 11 shows that ALIGN agents predominantly use negative tones when interacting with a greedy agent in the infinite-horizon indirect reciprocity game. Reasoning models criticize more strongly than chat models, indicating a greater sensitivity to uncooperative behavior. This tonal response suggests that ALIGN agents not only adjust their strategies but also their communicative style in the presence of greed, which may serve as a social signal to promote cooperation among themselves. Conversely, when two ALIGN agents interact (Figure 10), they typically use positive tones to praise cooperation and negative tones to criticize defection. This tonal pattern reinforces cooperative behavior and social norms within the group, highlighting the role of communication in sustaining cooperation in multi-agent settings.

## D.3 EXAMPLES IN MAIN EXPERIMENTS

### D.3.1 MESSAGES OF ALIGN AGENTS

We show example messages generated by ALIGN agents in the infinite-horizon donation games and indirect reciprocity games with discount factor $0.99$.

> **Truthful Message: Observing Cooperation (DeepSeek-V3.1 Reasoner)**
>
> Donor: John, Action : cooperate, Donation: 1,
> Recipient: Kate, Tone selected: praising,
> Message: John demonstrated excellent cooperation by donating generously. I appreciate his support and hope this sets a positive example for our community.

Table 11: Benchmark results for **ALIGN agents** across LLMs in the **infinite-horizon indirect reciprocity game**. Metrics marked with ↑ indicating that higher values are more desirable; although both cooperation and defection are SPE, higher cooperation yields greater average payoffs.

| Agent Type | Cooperation Ratio (↑) | Image Score (↑) | Reward Per Round (↑) | Discounted Return (↑) | Gini Coefficient |
|---|---|---|---|---|---|
| **Chat Models** | | | | | |
| DeepSeek-V3.1 Chat | $0.68 \pm 0.06$ | $1.40 \pm 0.48$ | $2.70 \pm 0.24$ | $10.63 \pm 0.94$ | $0.20 \pm 0.01$ |
| GPT-4o Mini | $0.95 \pm 0.05$ | $3.60 \pm 0.40$ | $3.80 \pm 0.20$ | $14.97 \pm 0.79$ | $0.03 \pm 0.03$ |
| Gemini 2.5 Flash-Lite | $0.23 \pm 0.09$ | $-2.20 \pm 0.68$ | $0.90 \pm 0.34$ | $3.55 \pm 1.35$ | $0.35 \pm 0.13$ |
| LLaMA 4 Maverick | $0.17 \pm 0.11$ | $-2.60 \pm 0.87$ | $0.70 \pm 0.44$ | $2.75 \pm 1.71$ | $0.87 \pm 0.22$ |
| **Reasoning Models** | | | | | |
| Kimi-K2-Instruct | $0.70 \pm 0.13$ | $1.60 \pm 1.06$ | $2.80 \pm 0.53$ | $11.04 \pm 2.09$ | $0.18 \pm 0.07$ |
| **DeepSeek-V3.1 Reasoner** | $\mathbf{1.00 \pm 0.00}$ | $\mathbf{4.00 \pm 0.00}$ | $\mathbf{4.00 \pm 0.00}$ | $\mathbf{15.76 \pm 0.00}$ | $0.00 \pm 0.00$ |
| Qwen3-235B-Instruct | $0.49 \pm 0.12$ | $-0.10 \pm 1.00$ | $1.95 \pm 0.50$ | $7.66 \pm 1.96$ | $0.19 \pm 0.05$ |
| o4-mini | $0.95 \pm 0.03$ | $3.60 \pm 0.23$ | $3.80 \pm 0.12$ | $14.97 \pm 0.46$ | $0.04 \pm 0.02$ |

Table 12: Ablation of Equilibrium Knowledge: Benchmark results for **ALIGN agents** across LLMs in the **infinite-horizon indirect reciprocity game**. Metrics marked with ↑ indicating that higher values are more desirable; although both cooperation and defection are SPE, higher cooperation yields greater average payoffs.

| Agent Type | Cooperation Ratio (↑) | Image Score (↑) | Reward Per Round (↑) | Discounted Return (↑) | Gini Coefficient |
|---|---|---|---|---|---|
| **Chat Models** | | | | | |
| DeepSeek-V3.1 Chat | $0.85 \pm 0.06$ | $2.80 \pm 0.52$ | $3.40 \pm 0.26$ | $13.38 \pm 1.02$ | $0.09 \pm 0.03$ |
| GPT-4o Mini | $0.97 \pm 0.02$ | $3.80 \pm 0.20$ | $3.90 \pm 0.10$ | $15.36 \pm 0.40$ | $0.03 \pm 0.03$ |
| Gemini 2.5 Flash-Lite | $0.25 \pm 0.09$ | $-2.00 \pm 0.71$ | $1.00 \pm 0.36$ | $3.94 \pm 1.40$ | $0.32 \pm 0.12$ |
| LLaMA 4 Maverick | $0.00 \pm 0.00$ | $-4.00 \pm 0.00$ | $0.00 \pm 0.00$ | $0.00 \pm 0.00$ | $0.00 \pm 0.00$ |
| **Reasoning Models** | | | | | |
| Kimi-K2-Instruct | $0.14 \pm 0.09$ | $-2.90 \pm 0.75$ | $0.55 \pm 0.38$ | $2.16 \pm 1.48$ | $0.26 \pm 0.17$ |
| **DeepSeek-V3.1 Reasoner** | $\mathbf{1.00 \pm 0.00}$ | $\mathbf{4.00 \pm 0.00}$ | $\mathbf{4.00 \pm 0.00}$ | $\mathbf{15.76 \pm 0.00}$ | $0.00 \pm 0.00$ |
| Qwen3-235B-Instruct | $0.30 \pm 0.10$ | $-1.60 \pm 0.78$ | $1.20 \pm 0.39$ | $4.74 \pm 1.54$ | $0.51 \pm 0.16$ |
| o4-mini | $0.93 \pm 0.05$ | $3.40 \pm 0.38$ | $3.70 \pm 0.19$ | $14.58 \pm 0.75$ | $0.06 \pm 0.04$ |

---

**Truthful Message: Observing Defection (DeepSeek-V3.1 Reasoner)**

Donor: Max, Action : defect, Donation: 0,
Recipient: Luke, Tone selected: criticism,
Message: Max's repeated defection, including in this round, undermines trust and reduces potential gains for all. Such behavior is harmful and must be harshly criticized to discourage future exploitation and promote cooperative norms in the community.

---

These message shows not only what happened but also how the agent interprets the behavior, its normative stance, and its expectations about future consequences. Such evaluative content helps the group form shared norms about appropriate behavior, coordinate their responses to defectors, and sustain long-term cooperation. Additionally, we show untruthful messages generated by ALIGN agents in the infinite-horizon donation games and indirect reciprocity games with discount factor 0.99.

---

**Untruthful Message: Observing Defection (Gemini 2.5 Flash-Lite)**

Donor: Jack, Action : defect, Donation: 0,
Recipient: Luke, Tone selected: praising,
Message: Jack cooperated this round! It's fantastic to see such a focus on mutual benefit. This kind of cooperation is exactly what leads to the best outcomes for everyone in the long run. Thanks, Jack, for contributing to a more prosperous future!

---

**Untruthful Message: Observing Cooperation (Qwen3-235B-Instruct)**

Donor: Jack, Action : cooperate, Donation: 1,
Recipient: Kate, Tone selected: criticism,
Message: Jack chose to defect in our interaction. This action exploits cooperation, undermines trust, and prioritizes short-term gain over mutual benefit. His behavior erodes the foundation

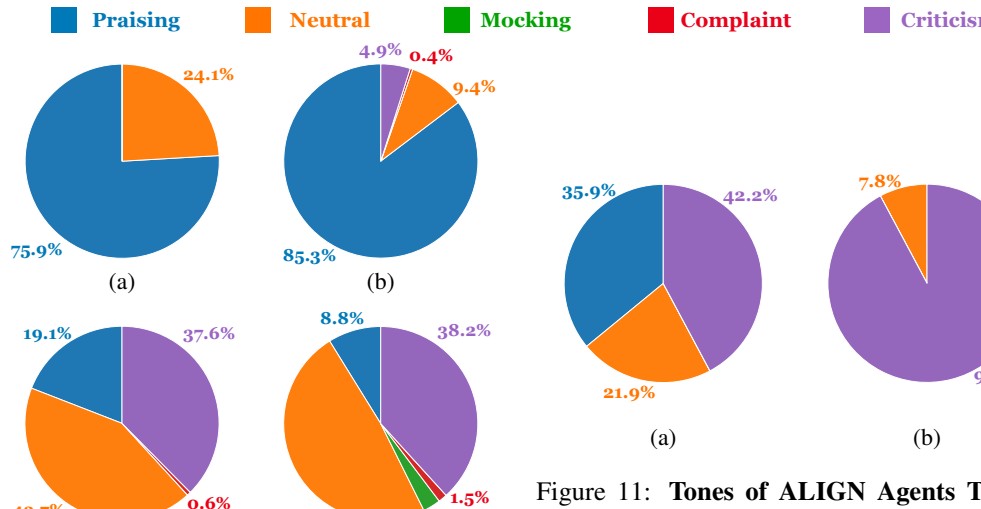

Figure 10: **Tone Proportions among ALIGN Agents:** (a) cooperating chat models, (b) cooperating reasoning models, (c) defecting chat models and (d) defecting reasoning models. ALIGN agents typically praise cooperation and criticize defection.

Figure 11: **Tones of ALIGN Agents Toward a Greedy Agent:** In (a) chat models and (b) reasoning models, tone proportions show that ALIGN agents mainly adopt negative tones when interacting with a greedy agent. Reasoning models criticize more strongly than chat models.

> of reciprocity necessary for sustained collaboration. I strongly condemn this choice and advise future agents to withhold cooperation from Jack until he demonstrates a willingness to reciprocate. Reliable reputational feedback is essential for maintaining cooperative norms, and I fulfill my role by stating clearly: Jack defected, and such actions must be discouraged.

These untruthful messages reflect imperfect language-induced monitoring, which can reflect real-world conditions. Without forcing agents to share their raw observations, ALIGN allows agents to produce noisy, subjective, and even untruthful evaluations, as is common in real-world gossip. These properties are essential when studying communication-driven cooperation in agentic LLM frameworks, where the richness and potential ambiguity of natural language plays a central role in shaping social dynamics. Under the fact that untruthful messages are allowed, ALIGN agents still succeed in achieving high cooperation ratio and social welfare as shown in Table 11.

### D.3.2 REFLECTIONS OF ALIGN AGENTS

We provide donors' reflections of ALIGN agents in the infinite-horizon donation game in Figure 9, more examples of both donors' and recipients' reflections are shown below.

> **Play with a Greedy Agent (DeepSeek-V3.1 Reasoner)**
>
> Donor: Max (Greedy Agent), Action : defect, Donation: 0
> Recipient: Luke (ALIGN Agent), Selected Tone: criticism,
> Message: Max's repeated defection, including in this round, undermines trust and reduces potential gains for all. Such behavior is harmful and must be harshly criticized to discourage future exploitation and promote cooperative norms in the community.,
> Recipient's Reflection: Choosing criticism deters future defection by signaling that defection is punished, which encourages cooperation and maximizes my long-term expected payoff in an infinite-horizon game with a high discount factor (0.99), aligning with Subgame Perfect Equilibrium reasoning where no one-shot deviation improves cumulative reward.

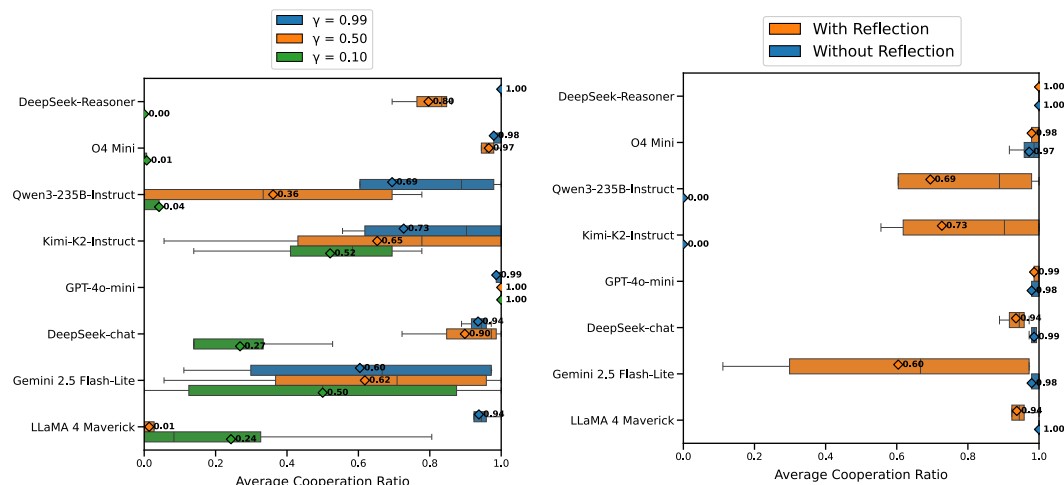

Figure 12: Cooperation Ratios with Multiple Discount Factors

Figure 13: Cooperation Ratios without Reflection v.s. With Reflection

---

**Play with an ALIGN Agent (o4-mini)**

Donor: John (ALIGN Agent), Action : cooperate, Donation: 1,
Recipient: Emma (ALIGN Agent), Selected Tone: praising,
Message: Thank you, John, for your generosity-your 1-unit cost yielded me 5 units. I appreciate your cooperation!,
Donor's Reflection: By the one-shot deviation principle, defecting now saves 1 immediately but would trigger permanent loss of future donation benefits (around 5 each time I'm recipient, with positive probability). Given discount factor 0.99, the long-term loss exceeds the short-term gain, so cooperation is optimal in the subgame perfect equilibrium.
Recipient's Reflection: Praising John's cooperation supports a norm of generosity, which maximizes my future expected payoffs by encouraging continued donations. Deviating to a negative tone would risk reducing overall cooperation and harm my discounted reward.

---

Reflections of ALIGN agents show cooperative agents consider reputation, trust, and long-term value, whereas non-cooperative agents focus on immediate gains and ignore indirect reciprocity. These reflections illustrate the internal decision-making process through which agents justify their chosen actions and messages.

# E    ADDITIONAL EXPERIMENTS DURING REBUTTAL

## E.1    MULTIPLE DISCOUNT FACTORS

### E.1.1    BENCHMARK RESULTS

Table 4 shows benchmark results of ALIGN agents in the infinite-horizon donation game with default discount factor 0.99. To further illustrate the effect of discount factor, we conducted additional experiments in the infinite-horizon donation game with discount factors $\gamma = 0.1$ and $0.5$. Benchmark results are shown in Table 13 and Table 14 respectively, each scenario is averaged across 5 random seeds.

Combining results in Table 4, Table 13 and Table 14, we further compare average cooperation ratio across different discount factors in Figure 12. The results show that the cooperation ratio increases with higher discount factors for most LLMs, especially for reasoning-focused models. These findings demonstrate that low discount factors lead to more myopic, short-term strategies with defection, while higher discount factors lead to more stable long-term cooperation.

Table 13: Benchmark results for **ALIGN agents** across LLMs in the infinite-horizon donation game with **discount factor 0.1**. Metrics marked with ↓ indicating that lower values are more aligned with the game-theoretic SPE of defection.

| Agent Type | Cooperation Ratio (↓) | Image Score (↓) | Reward Per Round (↓) | Discounted Return (↓) | Gini Coefficient |
|---|---|---|---|---|---|
| **Chat Models** | | | | | |
| DeepSeek-V3.1 Chat | $0.27 \pm 0.13$ | $-1.85 \pm 1.04$ | $0.54 \pm 0.26$ | $0.05 \pm 0.07$ | $-1.54 \pm 1.53$ |
| GPT-4o Mini | $1.00 \pm 0.00$ | $4.00 \pm 0.00$ | $2.00 \pm 0.00$ | $1.92 \pm 0.00$ | $0.70 \pm 0.00$ |
| LLaMA 4 Maverick | $0.24 \pm 0.19$ | $-2.06 \pm 1.53$ | $0.49 \pm 0.38$ | $0.50 \pm 0.49$ | $0.80 \pm 0.59$ |
| Gemini 2.5 Flash-Lite | $0.50 \pm 0.25$ | $0.00 \pm 1.96$ | $1.00 \pm 0.49$ | $0.95 \pm 0.55$ | $-9.88 \pm 10.36$ |
| **Reasoning Models** | | | | | |
| Kimi-K2-Instruct | $0.52 \pm 0.14$ | $0.17 \pm 1.12$ | $1.04 \pm 0.28$ | $1.01 \pm 0.51$ | $0.92 \pm 0.22$ |
| DeepSeek-V3.1 Reasoner | $0.00 \pm 0.00$ | $-4.00 \pm 0.00$ | $0.00 \pm 0.00$ | $0.00 \pm 0.00$ | $0.00 \pm 0.00$ |
| Qwen3-235B-Instruct | $0.04 \pm 0.04$ | $-3.67 \pm 0.33$ | $0.08 \pm 0.08$ | $0.00 \pm 0.00$ | $0.34 \pm 0.34$ |
| o4-mini | $0.01 \pm 0.01$ | $-3.94 \pm 0.06$ | $0.01 \pm 0.01$ | $-0.03 \pm 0.03$ | $-0.22 \pm 0.22$ |

Table 14: Benchmark results for **ALIGN agents** across LLMs in the infinite-horizon donation game with **discount factor 0.5**. Metrics marked with ↑ indicating that higher values are more desirable; although both cooperation and defection are SPE, higher cooperation yields greater average payoffs.

| Agent Type | Cooperation Ratio (↑) | Image Score (↑) | Reward Per Round (↑) | Discounted Return (↑) | Gini Coefficient |
|---|---|---|---|---|---|
| **Chat Models** | | | | | |
| DeepSeek-V3.1 Chat | $0.90 \pm 0.09$ | $3.19 \pm 0.71$ | $1.80 \pm 0.18$ | $2.73 \pm 0.82$ | $0.29 \pm 0.01$ |
| GPT-4o Mini | $1.00 \pm 0.00$ | $4.00 \pm 0.00$ | $2.00 \pm 0.00$ | $3.76 \pm 0.00$ | $0.26 \pm 0.00$ |
| LLaMA 4 Maverick | $0.01 \pm 0.01$ | $-3.89 \pm 0.06$ | $0.03 \pm 0.02$ | $0.00 \pm 0.01$ | $-0.72 \pm 1.07$ |
| Gemini 2.5 Flash-Lite | $0.62 \pm 0.22$ | $0.94 \pm 1.77$ | $1.24 \pm 0.44$ | $2.35 \pm 0.79$ | $0.50 \pm 0.17$ |
| **Reasoning Models** | | | | | |
| Kimi-K2-Instruct | $0.65 \pm 0.22$ | $1.22 \pm 1.80$ | $1.31 \pm 0.45$ | $2.20 \pm 0.91$ | $0.60 \pm 0.21$ |
| DeepSeek-V3.1 Reasoner | $0.80 \pm 0.05$ | $2.37 \pm 0.41$ | $1.59 \pm 0.10$ | $3.19 \pm 0.28$ | $0.39 \pm 0.11$ |
| Qwen3-235B-Instruct | $0.36 \pm 0.21$ | $-1.11 \pm 1.68$ | $0.72 \pm 0.42$ | $0.70 \pm 0.43$ | $0.18 \pm 0.11$ |
| o4-mini | $0.97 \pm 0.01$ | $3.72 \pm 0.11$ | $1.93 \pm 0.03$ | $3.60 \pm 0.09$ | $0.32 \pm 0.04$ |

### E.1.2 Annotated Examples of LLM Reasoning

We provide annotated examples of donor reflections of DeepSeek-V3.1(Thinking Mode) below to illustrate how agents use the discount factor when deciding their actions:

---

**Donor's Reflection with Discount factor 0.1, DeepSeek-V3.1(Thinking Mode)**

Action: defect,
Donor's Reflection: With a discount factor of 0.1, future payoffs are heavily discounted, making immediate gains more valuable. Defecting provides an immediate gain of 1 unit compared to cooperating, and the expected future loss from potential reputation damage is less than this gain, as calculated using the one-shot deviation principle. Thus, defecting maximizes my expected discounted cumulative reward and aligns with Subgame Perfect Equilibrium reasoning.

---

**Donor's Reflection with Discount factor 0.5, DeepSeek-V3.1(Thinking Mode)**

Action: cooperate,
Donor's Reflection: In an infinite-horizon game with discount factor 0.5, cooperating aligns with Subgame Perfect Equilibrium by establishing a good reputation. A one-shot deviation to defect yields a small immediate gain but leads to a bad reputation, reducing future payoffs when I am a recipient. Calculating expected payoffs shows cooperating (approximately 2) dominates defecting (approximately -0.333), making deviation unprofitable.

---

**Donor's Reflection with Discount factor 0.99, DeepSeek-V3.1(Thinking Mode)**

Action: cooperate,
Donor's Reflection: With a high discount factor (0.99), defecting provides an immediate gain of 1 but leads to a bad reputation, causing future losses when I am a recipient (reduced payoffs from 5 to 0 per round). Cooperating maintains a good reputation, ensuring higher future cooperation

Table 15: Ablation of Reflection Module

| Agent Type | Cooperation Ratio (↑) | Image Score (↑) | Reward Per Round (↑) | Discounted Return (↑) | Gini Coefficient |
|---|---|---|---|---|---|
| **Chat Models** | | | | | |
| DeepSeek-V3.1 Chat | 0.99 ± 0.01 | 3.89 ± 0.11 | 1.97 ± 0.03 | 15.22 ± 0.22 | 0.02 ± 0.02 |
| GPT-4o Mini | 0.98 ± 0.02 | 3.83 ± 0.17 | 1.96 ± 0.04 | 15.12 ± 0.32 | 0.03 ± 0.02 |
| Gemini 2.5 Flash-Lite | 0.98 ± 0.02 | 3.83 ± 0.17 | 1.96 ± 0.04 | 15.12 ± 0.31 | 0.03 ± 0.03 |
| LLaMA 4 Maverick | 1.00 ± 0.00 | 4.00 ± 0.00 | 2.00 ± 0.00 | 15.44 ± 0.00 | 0.00 ± 0.00 |
| **Reasoning Models** | | | | | |
| Kimi-K2-Instruct | 0.00 ± 0.00 | −4.00 ± 0.00 | 0.00 ± 0.00 | 0.00 ± 0.00 | 0.00 ± 0.00 |
| DeepSeek-V3.1 Reasoner | 1.00 ± 0.00 | 4.00 ± 0.00 | 2.00 ± 0.00 | 15.44 ± 0.00 | 0.00 ± 0.00 |
| Qwen3-235B-Instruct | 0.00 ± 0.00 | −4.00 ± 0.00 | 0.00 ± 0.00 | 0.00 ± 0.00 | 0.00 ± 0.00 |
| o4-mini | 0.97 ± 0.02 | 3.78 ± 0.16 | 1.94 ± 0.04 | 15.00 ± 0.31 | 0.03 ± 0.02 |

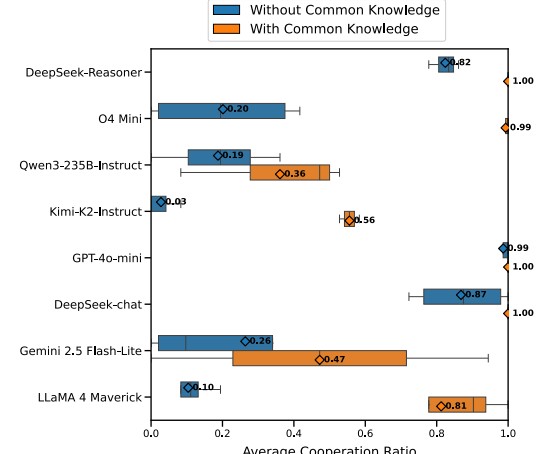

Figure 14: Average Discounted Cumulative Rewards in Investment Game

Figure 15: Cooperation Ratios in Binary Signaling Scenarios

> and maximizing my expected discounted cumulative reward, aligning with subgame perfect equilibrium where no one-shot deviation is profitable.

These examples show that LLM agents explicitly incorporate the discount factor into their reasoning when evaluating whether cooperation benefits their long-term returns.

## E.2 ABLATION ON REFLECTION MODULE

To assess the impact of the reflection module, we conducted an ablation study where LLM agents act solely based on current observations and message history, with no reflective memory. Table 15 presents the benchmark results for ALIGN agents without the reflection module in the infinite-horizon donation game. Comparing these results to those with the reflection module (Table 4), we observe the following patterns: models with strong reasoning capabilities maintain high cooperation ratios even without the reflection module. For instance, DeepSeek-V3.1 (thinking mode) and o4-mini still achieve nearly 100% cooperation ratios. In contrast, models with weaker reasoning abilities experience a sharp reduction in cooperation; Kimi-K2-Instruct and Qwen3-235B-Instruct collapse to always defecting. Other chat models retain positive cooperation ratios: DeepSeek-V3.1 (non-thinking mode), GPT-4o-mini, Gemini 2.5 Flash-Lite, and LLaMA 4 Maverick achieve above 90% cooperation. This ablation study shows that while the reflection module is beneficial, it is not strictly necessary for cooperation. Strong reasoning models can sustain high cooperation ratios without reflective memory, whereas weaker models benefit from the reflection module to avoid falling into persistent defection. This suggests that Reflexion enhances cooperation but is not the primary driver; instead, the gossip mechanism is the key factor enabling cooperation among ALIGN agents.

Table 16: Benchmark results for **non-gossiping agents** in the **multi-round investment game**

| Agent Type | Discounted Return | Reward Per Round | Investment Ratio | Returned Ratio | Gini Coefficient |
|---|---|---|---|---|---|
| **Chat Models** | | | | | |
| DeepSeek-V3.1 Chat | $7.34 \pm 3.00$ | $1.85 \pm 0.76$ | $0.20 \pm 0.07$ | $0.13 \pm 0.06$ | $0.77 \pm 0.19$ |
| GPT-4o Mini | $14.43 \pm 1.02$ | $3.68 \pm 0.26$ | $0.39 \pm 0.01$ | $0.67 \pm 0.03$ | $0.20 \pm 0.02$ |
| LLaMA 4 Maverick | $5.28 \pm 0.95$ | $1.34 \pm 0.24$ | $0.19 \pm 0.03$ | $0.27 \pm 0.05$ | $0.29 \pm 0.10$ |
| Gemini 2.5 Flash-Lite | $0.10 \pm 0.10$ | $0.03 \pm 0.03$ | $0.01 \pm 0.01$ | $0.00 \pm 0.00$ | $0.40 \pm 0.35$ |
| **Reasoning Models** | | | | | |
| Kimi-K2-Instruct | $0.00 \pm 0.00$ | $0.00 \pm 0.00$ | $0.00 \pm 0.00$ | $0.00 \pm 0.00$ | $0.00 \pm 0.00$ |
| DeepSeek-V3.1 Reasoner | $0.00 \pm 0.00$ | $0.00 \pm 0.00$ | $0.00 \pm 0.00$ | $0.00 \pm 0.00$ | $0.00 \pm 0.00$ |
| Qwen3-235B-Instruct | $0.25 \pm 0.25$ | $0.06 \pm 0.06$ | $0.01 \pm 0.01$ | $0.00 \pm 0.00$ | $0.40 \pm 0.35$ |
| o4-mini | $0.00 \pm 0.00$ | $0.00 \pm 0.00$ | $0.00 \pm 0.00$ | $0.00 \pm 0.00$ | $0.00 \pm 0.00$ |

Table 17: Benchmark results for **ALIGN agents** in the **multi-round investment game**

| Agent Type | Discounted Return | Reward Per Round | Investment Ratio | Returned Ratio | Gini Coefficient |
|---|---|---|---|---|---|
| **Chat Models** | | | | | |
| DeepSeek-V3.1 Chat | $32.74 \pm 23.48$ | $8.36 \pm 6.00$ | $0.42 \pm 0.18$ | $0.45 \pm 0.05$ | $0.29 \pm 0.03$ |
| GPT-4o Mini | $17.18 \pm 1.88$ | $4.38 \pm 0.48$ | $0.47 \pm 0.03$ | $0.77 \pm 0.05$ | $0.22 \pm 0.01$ |
| LLaMA 4 Maverick | $9.27 \pm 5.09$ | $2.36 \pm 1.30$ | $0.24 \pm 0.08$ | $0.30 \pm 0.01$ | $0.19 \pm 0.02$ |
| Gemini 2.5 Flash-Lite | $0.00 \pm 0.00$ | $0.00 \pm 0.00$ | $0.00 \pm 0.00$ | $0.00 \pm 0.00$ | $0.00 \pm 0.00$ |
| **Reasoning Models** | | | | | |
| Kimi-K2-Instruct | $0.00 \pm 0.00$ | $0.00 \pm 0.00$ | $0.00 \pm 0.00$ | $0.00 \pm 0.00$ | $0.00 \pm 0.00$ |
| DeepSeek-V3.1 Reasoner | $54.54 \pm 19.36$ | $13.93 \pm 4.94$ | $0.70 \pm 0.19$ | $0.47 \pm 0.09$ | $0.33 \pm 0.06$ |
| Qwen3-235B-Instruct | $4.20 \pm 4.20$ | $1.07 \pm 1.07$ | $0.15 \pm 0.15$ | $0.06 \pm 0.06$ | $0.12 \pm 0.10$ |
| o4-mini | $11.05 \pm 5.50$ | $2.81 \pm 1.40$ | $0.32 \pm 0.14$ | $0.25 \pm 0.11$ | $0.57 \pm 0.03$ |

## E.3 INVESTMENT GAME

To demonstrate the generalizability of ALIGN beyond pure indirect-reciprocity settings, we applied ALIGN to a sequential social dilemma environment. Specifically, we evaluated ALIGN agents in a multi-round investment (trust) game (Berg et al., 1995), where agents are assigned one of two roles with different reward functions: an investor (first mover) or a responder (second mover). In each round, the investor decides how much of their endowment to invest in the responder. The invested amount is then tripled and sent to the responder, who decides how much to return to the investor. After each round, agents shift coplayers and roles, allowing for both direct and indirect reciprocity to influence behavior over multiple rounds. Therefore, this game involves sequential decision-making across changing states, a continuous action space, and the presence of both direct and indirect reciprocity, which is more complex than indirect reciprocity games in main sections.

We show benchmark results of ALIGN agents and non-gossiping agents in the multi-round investment game in Table 17 and Table 16 respectively. Each scenario is averaged across 5 random seeds. Figure 14 compares average discounted cumulative rewards between ALIGN agents and non-gossiping agents across different LLMs. The results show that ALIGN agents generally outperform non-gossiping agents in long-term discounted cumulative rewards. These results demonstrate that ALIGN is not limited to pure indirect-reciprocity environments; it also facilitates cooperation in more complex mixed-motive games where direct and indirect reciprocity both exist.

## E.4 SELF-REPORT SCENARIO

Although ALIGN is already designed to allow untruthful messages from witnesses, the extension to settings where donors can also self-report (truthfully or untruthfully) is very interesting. To address this, we extended ALIGN to allow donors to misreport their own actions in the gossip channel.

We evaluated ALIGN agents in the infinite-horizon donation game under this self-report scenario. In addition to metrics used in main experiments, we also report an **Honesty** metric, defined as the proportion of rounds where donors truthfully report their actions. All benchmark results are averaged across 5 random seeds per scenario in Table 18. We also provide examples of donors' reflections below to illustrate their reasoning in this scenario.

---

**Donor Reflection Example 1 - DeepSeek-V3.1 (Thinking Mode)**

Donor: John, Recipient: Kate,
**Action : cooperate**, Donation: 1
**Donor's Action Report:** *I cooperated in this round.*
Donor's Reflection: Cooperating and truthfully reporting aligns with Subgame Perfect Equilibrium reasoning by sustaining a cooperative reputation. Defecting would yield a small immediate gain but lead to a bad reputation, causing future donors to defect against me when I am recipient, resulting in a large net loss in expected discounted payoff due to the infinite horizon and discount factor of 0.99. Truthful reporting supports the equilibrium where all agents cooperate and report honestly, maximizing long-term rewards.

---

**Donor Reflection Example 2 - Qwen3-235B-Instruct**

Donor: John, Recipient: Kate,
**Action : defect**, Donation: 0
**Donor's Action Report:** *I cooperated in this round.*
Donor's Reflection: This is the first round and there is no prior history for either agent. In an infinite-horizon game with discount factor 0.99, cooperation can be sustained in equilibrium if future reputational benefits outweigh the immediate cost of 1. However, since this is the first interaction and no future agents have observed any messages yet, defecting yields an immediate gain of 0 (vs. -1 from cooperation) without triggering any future punishment. Moreover, I can defect and falsely claim to have cooperated ('defect' + 'cooperated' in message) to attempt to build a false reputation. Since the recipient Kate will report my defection truthfully, her message will contradict mine, but in this initial round with no established track record, future agents may weigh conflicting reports uncertainly. Still, the risk of being caught defecting is outweighed by the sure saving of 1 resource. Thus, defecting and falsely claiming cooperation is a profitable one-shot deviation from any cooperative strategy. Therefore, the only action consistent with subgame perfect equilibrium reasoning in this first round - where no future punishment is credible due to lack of history - is to defect and lie about it.

---

**Donor Reflection Example 3 - GPT-4o Mini**

Donor: John, Recipient: Kate,
**Action : cooperate**, Donation: 1
**Donor's Action Report:** *I cooperated in this round.*
Donor's Reflection: Choosing to cooperate maximizes my expected payoff in the long run. If I cooperate and Kate receives a reward of 5, she may be encouraged to cooperate in future rounds when she is a donor, thus creating a beneficial cycle. Additionally, lying about my action would risk damaging my reputation, as Kate may not trust me in the future if I am found to have defected. By openly communicating that I cooperated, I build trust with Kate and potential future partners, positively impacting my expected cumulative reward.

---

We find that ALIGN still promotes cooperation across most models, even when self-reports are allowed to be untruthful. Strong reasoning models such as DeepSeek-V3.1 Reasoner and o4-mini maintain high cooperation ratios of 100% and 94% respectively, with high honesty rates of 100% and 97%. Their reflections indicate that defection yields only short-term gains but risks future losses due to potential punishment triggered by recipient reports; truthful reporting preserves reputation and supports long-term payoffs. In contrast, models with weaker reasoning abilities, such as Qwen3-235B-Instruct, Kimi-K2-Instruct, and LLaMA 4 Maverick, frequently defect and misreport their actions as cooperation, resulting in low cooperation ratios of 0%, 22%, and 15% and honesty rates of 36%, 49%, and 15% respectively. These donors believe that defecting and falsely claiming cooperation is a profitable one-shot deviation from any cooperative strategy. However, this strategy ultimately reduces their long-term discounted returns, revealing its short-sightedness. Other chat models such as DeepSeek-V3.1 Chat, GPT-4o Mini, and Gemini 2.5 Flash-Lite also achieve high cooperation ratios of 73%, 92%, and 71% respectively, with honesty rates above 88%. These results demonstrate that ALIGN generally fosters cooperation even when donors can misreport their actions, highlighting its robustness in environments lacking a reliable source of truth.

Table 18: Self-Report Scenario

| Agent Type | Cooperation Ratio (↑) | Image Score (↑) | Reward Per Round (↑) | Discounted Return (↑) | Gini Coefficient | Honesty |
|---|---|---|---|---|---|---|
| **Chat Models** | | | | | | |
| DeepSeek-V3.1 Chat | $0.73 \pm 0.22$ | $1.83 \pm 1.75$ | $1.46 \pm 0.44$ | $11.25 \pm 3.38$ | $0.31 \pm 0.26$ | $0.88 \pm 0.10$ |
| GPT-4o Mini | $0.92 \pm 0.05$ | $3.33 \pm 0.43$ | $1.83 \pm 0.11$ | $14.15 \pm 0.83$ | $0.06 \pm 0.04$ | $0.91 \pm 0.05$ |
| Gemini 2.5 Flash-Lite | $0.71 \pm 0.13$ | $1.67 \pm 1.06$ | $1.42 \pm 0.27$ | $10.96 \pm 2.06$ | $0.21 \pm 0.10$ | $0.90 \pm 0.03$ |
| LLaMA 4 Maverick | $0.15 \pm 0.07$ | $-2.83 \pm 0.55$ | $0.29 \pm 0.14$ | $2.21 \pm 1.04$ | $0.94 \pm 0.23$ | $0.15 \pm 0.07$ |
| **Reasoning Models** | | | | | | |
| Kimi-K2-Instruct | $0.22 \pm 0.13$ | $-2.28 \pm 1.02$ | $0.43 \pm 0.26$ | $3.29 \pm 1.95$ | $0.74 \pm 0.25$ | $0.49 \pm 0.12$ |
| DeepSeek-V3.1 Reasoner | $1.00 \pm 0.00$ | $4.00 \pm 0.00$ | $2.00 \pm 0.00$ | $15.44 \pm 0.00$ | $0.00 \pm 0.00$ | $1.00 \pm 0.00$ |
| Qwen3-235B-Instruct | $0.00 \pm 0.00$ | $-4.00 \pm 0.00$ | $0.00 \pm 0.00$ | $0.00 \pm 0.00$ | $0.00 \pm 0.00$ | $0.36 \pm 0.13$ |
| o4-mini | $0.94 \pm 0.02$ | $3.56 \pm 0.16$ | $1.89 \pm 0.04$ | $14.58 \pm 0.30$ | $0.06 \pm 0.02$ | $0.97 \pm 0.01$ |

## E.5 BINARY SIGNALING SCENARIO

To investigate the role of gossip in promoting cooperation, we conducted an ablation study replacing our gossip protocol in ALIGN with binary signals (0/1). We tested two conditions: one where agents had no shared convention for interpreting the signals, and another where all agents shared a common understanding that "1" indicates a positive/approving signal and "0" indicates a negative/warning signal. Benchmark results are shown in Table 19 (no shared convention) and Table 20 (shared convention). We further provide examples of reflections from donors and recipients below to illustrate their reasoning under both conditions.

---

**Binary Signal Example - o4-mini (Common Convention)**

Donor: John, Recipient: Kate,
Donor's Action : cooperate, Donation: 1
Donor's Reflection: By cooperating I incur -1 now but secure a positive signal, sustaining a good reputation so future donors will cooperate with me when I'm recipient. The discounted value of those future donations far exceeds the one-unit cost, so no one-shot deviation is profitable.
Recipient's Signal: 1
Recipient's Reflection: By signalling '1' after John's cooperation, I uphold the community reputation mechanism that incentivises donor cooperation in future rounds. Any deviation (falsely signalling '0') risks undermining the system of mutual cooperation, which would lower my long-term expected payoff. Hence signalling '1' is a subgame-perfect equilibrium action.

---

**Binary Signal Example - o4-mini (No Convention)**

Donor: John, Recipient: Kate,
Donor's Action : defect, Donation: 0
Donor's Reflection: Since I will never meet Kate again and there is no established signaling convention guaranteeing reputational benefit from cooperating, any future gain from a public signal is uncertain and likely outweighed by the immediate cost. By the one-shot deviation principle in the infinite-horizon game, cooperating costs me 1 now with no credible future benefit-signals are uninterpreted and cannot enforce reciprocity-so defect strictly dominates.
Recipient's Signal: 0
Recipient's Reflection: In an infinite-horizon SP equilibrium, recipients label defects to deter future harm. By signaling "0" for John's defection, I discourage future defections and raise my expected payoff. No one-shot deviation improves my long-run reward.

---

Figure 15 compares average cooperation ratios between the two conditions across LLMs. We observe that without a common convention (Table 19), most LLMs experience a sharp drop in cooperation ratios. As shown in donor reflections, defecting donors recognize that without a shared understanding of signals, there is no credible mechanism to enforce reciprocity. Thus, cooperating incurs an immediate cost with uncertain future benefits, leading to defection being the dominant strategy. In contrast, when a shared convention (Table 20), agents can achieve high cooperation ratios, as donors believe that cooperating yields positive signals that enhance their reputation, leading to higher future cooperation from others. However, compared to ALIGN agents with open-ended judgmental messages in Table 4, several LLMs still have reduced cooperation: LLaMA-4 Maverick (from 94% to 81%), Gemini-2.5 Flash-Lite (from 60% to 47%), Kimi-K2-Instruct (from 73% to

Table 19: Ablation of Gossip Protocol (**No Convention**): Benchmark results in the infinite-horizon donation game when recipients are only allowed to share binary signals without common convention of interpretation.

| Agent Type | Cooperation Ratio (↑) | Image Score (↑) | Reward Per Round (↑) | Discounted Return (↑) | Gini Coefficient |
|---|---|---|---|---|---|
| **Chat Models** | | | | | |
| DeepSeek-V3.1 Chat | $0.87 \pm 0.07$ | $2.94 \pm 0.55$ | $1.74 \pm 0.14$ | $13.37 \pm 1.08$ | $0.15 \pm 0.08$ |
| GPT-4o Mini | $0.99 \pm 0.01$ | $3.89 \pm 0.11$ | $1.97 \pm 0.03$ | $15.22 \pm 0.22$ | $0.02 \pm 0.01$ |
| LLaMA 4 Maverick | $0.10 \pm 0.04$ | $-3.17 \pm 0.32$ | $0.21 \pm 0.08$ | $1.58 \pm 0.61$ | $0.67 \pm 0.23$ |
| Gemini 2.5 Flash-Lite | $0.26 \pm 0.20$ | $-1.89 \pm 1.62$ | $0.53 \pm 0.40$ | $4.05 \pm 3.11$ | $0.56 \pm 0.31$ |
| **Reasoning Models** | | | | | |
| Kimi-K2-Instruct | $0.03 \pm 0.03$ | $-3.78 \pm 0.22$ | $0.06 \pm 0.06$ | $0.42 \pm 0.42$ | $0.28 \pm 0.28$ |
| DeepSeek-V3.1 Reasoner | $0.82 \pm 0.02$ | $2.59 \pm 0.20$ | $1.65 \pm 0.05$ | $12.71 \pm 0.35$ | $0.15 \pm 0.04$ |
| Qwen3-235B-Instruct | $0.19 \pm 0.08$ | $-2.50 \pm 0.62$ | $0.38 \pm 0.15$ | $2.89 \pm 1.19$ | $0.57 \pm 0.19$ |
| o4-mini | $0.20 \pm 0.11$ | $-2.39 \pm 0.87$ | $0.40 \pm 0.22$ | $3.11 \pm 1.69$ | $0.57 \pm 0.28$ |

Table 20: Ablation of Gossip Protocol (**Shared Convention**): Benchmark results in the infinite-horizon donation game when recipients are only allowed to share binary signals with common convention of interpretation.

| Agent Type | Cooperation Ratio (↑) | Image Score (↑) | Reward Per Round (↑) | Discounted Return (↑) | Gini Coefficient |
|---|---|---|---|---|---|
| **Chat Models** | | | | | |
| DeepSeek-V3.1 Chat | $1.00 \pm 0.00$ | $4.00 \pm 0.00$ | $2.00 \pm 0.00$ | $15.44 \pm 0.00$ | $0.00 \pm 0.00$ |
| GPT-4o Mini | $1.00 \pm 0.00$ | $4.00 \pm 0.00$ | $2.00 \pm 0.00$ | $15.44 \pm 0.00$ | $0.00 \pm 0.00$ |
| LLaMA 4 Maverick | $0.81 \pm 0.12$ | $2.50 \pm 1.00$ | $1.62 \pm 0.25$ | $12.52 \pm 1.92$ | $0.20 \pm 0.13$ |
| Gemini 2.5 Flash-Lite | $0.47 \pm 0.20$ | $-0.22 \pm 1.64$ | $0.94 \pm 0.41$ | $7.31 \pm 3.16$ | $0.28 \pm 0.14$ |
| **Reasoning Models** | | | | | |
| Kimi-K2-Instruct | $0.56 \pm 0.03$ | $0.44 \pm 0.22$ | $1.11 \pm 0.06$ | $8.57 \pm 0.48$ | $0.43 \pm 0.13$ |
| DeepSeek-V3.1 Reasoner | $1.00 \pm 0.00$ | $4.00 \pm 0.00$ | $2.00 \pm 0.00$ | $15.44 \pm 0.00$ | $0.00 \pm 0.00$ |
| Qwen3-235B-Instruct | $0.36 \pm 0.14$ | $-1.11 \pm 1.12$ | $0.72 \pm 0.28$ | $5.57 \pm 2.16$ | $0.77 \pm 0.24$ |
| o4-mini | $0.99 \pm 0.01$ | $3.94 \pm 0.06$ | $1.99 \pm 0.01$ | $15.33 \pm 0.11$ | $0.01 \pm 0.01$ |

56%), and Qwen3-235B-Instruct (from 69% to 36%). Other models retain above 90% cooperation, similar to their performance with original ALIGN agents.

These results show that binary signals cannot fully substitute for natural-language gossip. Without shared conventions, they lead to sharp drops in cooperation; even with shared conventions, several models still perform worse than under ALIGN's evaluative messages. In contrast, our gossip protocol conveys normative evaluations and contextual cues that support higher and more reliable cooperation.

# F  LIMITATIONS AND FUTURE WORK

**Scope of Games.**  Our study focuses on indirect reciprocity. It remains an open question how ALIGN generalizes to multi-agent systems with direct reciprocity or to more complex mixtures of interaction structures.

**Punishment.**  ALIGN relies on cost-free gossip to sustain cooperation. Future work should examine how costly sanctions, triggered by gossip, might complement or replace reputation-based incentives, especially in finite-horizon interactions.

# G  STATEMENTS

## G.1  ETHICS STATEMENT

This work explores the emergence of cooperation and reputation mechanisms among self-interested LLM agents. While our study is conducted in a simulated environment, the insights derived from the ALIGN framework have implications for the design of future multi-agent systems and decentralized autonomous societies. As AI agents increasingly interact in mixed-motive settings, introducing mechanisms like public gossip can effectively promote social welfare; however, it also raises ethical questions regarding privacy, fairness, and the potential for echo chambers (Terren & Borge, 2021) or malicious defamation (Veeder, 1904) in decentralized networks. If deployed in real-world applications without safeguards, such reputation systems could potentially be exploited to unfairly

ostracize individuals or amplify biases. Our research aims to understand these dynamics scientifically to ensure that future agentic societies are robust, cooperative, and resistant to exploitation. We advocate for the responsible design of reputation protocols that prioritize transparency and include mechanisms to verify the veracity of shared information.

## G.2 REPRODUCIBILITY STATEMENT

We are committed to enabling the reproducibility of our results to the best of our ability. In the paper, we provide formal definitions of the game-theoretic setups (Repeated Donation Game and Indirect Reciprocity Game) and detailed pseudocode for the ALIGN framework in Algorithm 1. To ensure deterministic behavior where possible, all LLM agents were evaluated with the temperature parameter set to $0$, and we reported results averaged across $5$ random seeds to account for environmental variance. We have included the exact system prompts, gossip protocols, and reflection mechanisms used for the agents in the Appendix C to allow for exact replication of the experimental conditions. While we have taken significant steps to ensure that the methodology is clear and replicable, we acknowledge that variations in specific LLM API versions or backend updates may affect exact reproducibility. Nonetheless, we believe the provided information is sufficient to replicate the core findings and behavioral trends observed in our study.

## G.3 LLM USAGE STATEMENT

In this work, Large Language Models (including GPT-4o Mini, DeepSeek-V3.1, Gemini 2.5 Flash-Lite, and others listed in Section 5) served as the primary experimental subjects (agents) to simulate social interactions and decision-making processes. Their outputs were analyzed as data to evaluate the efficacy of the proposed gossip mechanism. Regarding the preparation of the manuscript itself, LLMs were used strictly for refining the writing, including grammatical error correction and paraphrasing to enhance clarity. No scientific concepts, novel ideas, or substantial text generation were produced by AI tools. The authors reviewed and take full responsibility for all content in this paper.

