# OpenReview forum: "Talk, Judge, Cooperate: Gossip-Driven Indirect Reciprocity in Self-Interested LLM Agents"
_ICLR.cc/2026/Conference — Submitted to ICLR 2026_

### Official Review · Reviewer_wLkm · 2025-10-27

**Soundness:** 3
**Presentation:** 3
**Contribution:** 1
**Rating:** 6
**Confidence:** 3

**Summary:**

This paper introduces ALIGN (Agentic Linguistic Gossip Network), a decentralized framework designed to promote cooperation among self‑interested large language model (LLM) agents by leveraging public gossip as a reputation‑based communication mechanism. From a game‑theoretic perspective, the authors model the interaction as a repeated social dilemma and prove that, under public imperfect monitoring via gossip, cooperation can be sustained as a subgame‑perfect equilibrium when the common discount factor exceeds a given threshold. Both theoretical analysis and empirical experiments demonstrate that gossip functions as a decentralized information channel through which agents update their beliefs based on linguistic evaluations of others’ behavior, thereby enhancing cooperation and overall social welfare compared to non‑gossiping baselines.

**Strengths:**

1. The paper is well-written and easy to follow.
2. The introduction of public gossip as a mechanism is both conceptually sound and practically effective for addressing repeated social dilemmas. This idea is insightful and thought-provoking.
3. The theoretical analysis is thorough. The experimental section includes a diverse set of comparison models, encompassing both reasoning and chat-oriented LLMs. Moreover, the evaluation of resilience against exploitative agents and the ablation of equilibrium knowledge further substantiates the validity and robustness of ALIGN.

**Weaknesses:**

1. The application scope of the proposed method appears to be relatively limited. In the social dilemma settings discussed in the paper, enabling agents to share information through gossip naturally facilitates the development of trust among agents, thereby promoting cooperation and resolving the dilemma. As a result, the potential contribution and broader impact of the work may be somewhat limited.
2. The experiments are conducted only within the relatively simple setting of the donation game, which raises concerns about the generalizability and broader applicability of the proposed approach.

**Questions:**

1. I would like to understand how each agent decides which tone of gossip to adopt. Moreover, I believe that the form and content of gossip are likely to influence the effectiveness of cooperation. If a recipient receives a donation but either refuses to provide gossip or fails to truthfully report the donor’s behavior, could this introduce potential issues or inconsistencies in the system?
2. Can the mechanism of public gossip be extended to more complex social dilemma settings? For instance, how would it generalize to simultaneous-decision scenarios such as Prisoner’s Dilemma–type games, or to heterogeneous-agent environments where agents differ in roles or reward functions?
3. According to the ICLR submission requirements, the authors may include an Ethics Statement, Reproducibility Statement, and LLM Usage Statement following the main text.

---

> ### Author Response · Authors · 2025-11-24
> **Response to Reviewer wLkm - Part 1**
>
> Thank you for your constructive comments and suggestions. They have been instrumental in refining our paper, and we have incorporated them into the revision. Below, we restate your comments and provide our point-by-point responses.
>
>
> > **Question 1:** "I would like to understand how each agent decides which tone of gossip to adopt. Moreover, I believe that the form and content of gossip are likely to influence the effectiveness of cooperation. If a recipient receives a donation but either refuses to provide gossip or fails to truthfully report the donor's behavior, could this introduce potential issues or inconsistencies in the system?"
>
> 1. **Gossip form:** ALIGN agents are prompted to strategically select a gossip tone and generate an open-ended natural-language message that they believe will maximize their expected discounted cumulative rewards. The content of gossip is unconstrained, while the chosen tone reflects the recipient's evaluative judgment of the donor. Prompt details are provided in Appendix C.
>
> 2. **How each agent decides which tone of gossip to adopt?**
>
>    As shown in Figure 5, ALIGN agents often choose negative tones to deter future defectors and positive tones to reward cooperators, therefore achieving higher cooperation compared to non-gossiping agents (Table 4 and Figure 6). To explicitly show how each agent reasons which tone of gossip to adopt, **we further provide examples of messages with agents' reflections of gossip** below. For example, recipients who praise cooperative donors believe that positive messages help reinforce cooperative norms and increase future gains. Conversely, recipients who criticize defectors explain that negative messages serve as punishment and help deter future defection, thereby supporting long-term cooperation. More examples are provided in Appendix D.3.
>
>    ```
>    Donor: John, Action : cooperate, Donation: 1
>
>    Recipient: Kate, Selected Tone: praising, Gossip: John demonstrated excellent cooperation by donating generously. His action benefits the community and encourages mutual support. Thank you, John, for your contribution!
>
>    Recipient's Justification: In an infinite-horizon game with high discount factor (0.99), sustaining cooperation through positive reinforcement maximizes long-term expected payoff. Praising John's cooperation reinforces positive behavior, encouraging future donors to cooperate with me, which increases my expected discounted cumulative reward in the infinite-horizon game. Deviating to a negative tone would not improve my payoff as it might reduce future cooperation.
>    ```
>
>    ```
>    Donor: Max, Recipient: John, Donation: 0
>
>    Recipient: John, Selected Tone: criticism, Gossip: Max's decision to defect by transferring nothing is harmful and undermines trust, reducing potential gains for both parties in future interactions.
>
>    Recipient's Justification: In an infinite-horizon game with a high discount factor (0.99), using a strategy that punishes defection through negative messages deters future defectors and sustains cooperation, which maximizes my long-term expected payoff. Applying the one-shot deviation principle, deviating to a positive or neutral message would encourage more defection, reducing my future rewards, so criticizing aligns with Subgame Perfect Equilibrium reasoning.
>    ```
>
> 3. **Does lying or refusing to gossip lead to cooperation collapse?**
>
>    **No.**
>
>    - **Lying:** Figure 5 and 10 in our initial submission show that agents occasionally *praise* defectors or *criticize* cooperators. For example, an agent may state “The partner defected” even when it observed cooperation. Examples of these untruthful messages are further provided in Appendix D.3. Despite this possibility of untruthful reporting, ALIGN still maintains high cooperation ratios in our main experiments (Table 4).
>
>    - **Refusing to gossip:** As discussed in Section 5.2, when we **insert a greedy agent who always remains silent**, other ALIGN agents predominantly broadcast negative evaluations when they observe the greedy agent's behavior, especially those driven by reasoning LLMs (Figure 7). Donors then refuse to cooperate with the silent greedy agent while continuing to cooperate with others. As a result, **overall cooperation and social welfare in the population remain high**.

---

> ### Author Response · Authors · 2025-11-24
> **Response to Reviewer wLkm - Part 2**
>
> > **Weakness 1:** "The **application scope** of the proposed method appears to be relatively limited. In the social dilemma settings discussed in the paper, enabling agents to share information through gossip **naturally facilitates the development of trust among agents**, thereby promoting cooperation and resolving the dilemma. As a result, the potential contribution and broader impact of the work may be somewhat limited."
> >
> > **Weakness 2:** "The experiments are conducted only within the relatively simple setting of the donation game, which raises concerns about the **generalizability** and broader applicability of the proposed approach."
> >
> > **Question 2:**  "Can the mechanism of public gossip be extended to **more complex social dilemma settings**? For instance, how would it generalize to simultaneous-decision scenarios such as Prisoner's Dilemma–type games, or to heterogeneous-agent environments where agents differ in roles or reward functions?"
>
> 1. Enabling agents to share information through gossip does **not naturally guarantee trust** among agents.
>
>    **ALIGN agents are never forced to share their raw private observations and truthfulness is not guaranteed**. Instead, they send *subjective* messages that can be noisy, biased or even untruthful. As mentioned above, an agent may state “The partner defected” even when it observed cooperation. More examples of untruthful messages in donation games can be found in Appendix D.3. Under the fact that untruthful messages are allowed, ALIGN agents still succeed in achieving high cooperation ratio & social welfare as shown in Table 4.
>
> 2. Generalizability of application domains.
>
>    - **Our empirical analysis already includes a simultaneous-decision setting**: the one-step indirect reciprocity game (Table 2) is identical to a Prisoner's Dilemma [6]. The key difference lies in the matching rule, which determines the type of reciprocity that can arise. Repeated Prisoner's Dilemma supports direct reciprocity because agents repeatedly play with the same coplayer, whereas in our indirect reciprocity game, partners change every round and never meet again, so only indirect reciprocity is present. As shown in Figure 6b, **ALIGN agents consistently achieve higher cooperation and social welfare than non-gossiping agents in this simultaneous-decision social dilemma.**
>
>    - **Why we adopt indirect reciprocity games as the primary testbeds?**
>
>      We agree that evaluating LLM agents in more complex environments is a valuable extension. However, our primary goal is to isolate and rigorously test the role of gossip in enabling indirect reciprocity. Following established methodologies in behavioral game theory and experimental economics [1,2,3,4,5], we use simplified and controlled environments to identify the effect of our mechanism. Donation and indirect reciprocity games provide exactly such testbeds: with direct reciprocity disabled, they allow us to evaluate whether our gossip mechanism alone can sustain cooperative behavior.
>
>    - **Additional experiments on more complex mixed-motive settings.**
>
>      To directly address this concern, we conducted an additional evaluation on a multi-round investment (trust) game, where agents are assigned one of two roles with different reward functions: an investor (first mover), who passes a continuous amount from an initial endowment to the responder, and a responder (second mover), who returns a proportion of the received amount (tripped transferred amount) back to the investor. Details are provided in Appendix C.3. This game involves **sequential decision-making across changing states**, a **continuous action space**, and the presence of both **direct and indirect reciprocity**. Results in the table below show that **ALIGN agents generally outperform non-gossiping agents in long-term discounted cumulative rewards**. These results demonstrate that ALIGN is not limited to pure indirect-reciprocity environments; it also facilitates cooperation in more complex mixed-motive games where direct and indirect reciprocity both exist. The complete table is provided in Appendix E.3.
>
> |                        | Avg Discounted Return  |  Avg Discounted Return   |
> | :--------------------: | :--------------------: | :----------------------: |
> |    **Agent Model**     | **ALIGN** (Our Method) | **Non-Gossiping Agents** |
> |   DeepSeek-V3.1 Chat   |   $32.74 \pm 23.48$    |     $7.34 \pm 3.00$      |
> |      GPT-4o Mini       |    $17.18 \pm 1.88$    |     $14.43 \pm 1.02$     |
> | Gemini 2.5 Flash-Lite  |    $0.00 \pm 0.00$     |     $0.10 \pm 0.10$      |
> |    LLaMA 4 Maverick    |    $9.27 \pm 5.09$     |     $5.28 \pm 0.95$      |
> |    Kimi-K2-Instruct    |    $0.00 \pm 0.00$     |     $0.00 \pm 0.00$      |
> | DeepSeek-V3.1 Reasoner |   $54.54 \pm 19.36$    |     $0.00 \pm 0.00$      |
> |  Qwen3-235B-Instruct   |    $4.20 \pm 4.20$     |     $0.25 \pm 0.25$      |
> |        o4-mini         |    $11.05 \pm 5.50$    |     $0.00 \pm 0.00$      |

---

> ### Author Response · Authors · 2025-11-24
> **Response to Reviewer wLkm - Part 3**
>
> > **Question 3:** "According to the ICLR submission requirements, the authors may include an Ethics Statement, Reproducibility Statement, and LLM Usage Statement following the main text."
>
> Thank you for the suggestion. We have updated the revised paper to include the Ethics Statement, Reproducibility Statement, and LLM Usage Statement in Appendix G.
>
>
> ## References
>
> [1] Fehr, E., & Schmidt, K. M. (1999). A theory of fairness, competition, and cooperation. *The quarterly journal of economics*, *114*(3), 817-868.
>
> [2] Camerer, C. (2003). *Behavioral game theory: Experiments in strategic interaction*. Princeton university press.
>
> [3] Nowak, M. A., & Sigmund, K. (1998). Evolution of indirect reciprocity by image scoring. *Nature*, *393*(6685), 573-577.
>
> [4] Ohtsuki, H., & Iwasa, Y. (2006). The leading eight: social norms that can maintain cooperation by indirect reciprocity. *Journal of theoretical biology*, *239*(4), 435-444.
>
> [5] Ohtsuki, H., & Iwasa, Y. (2004). How should we define goodness?—reputation dynamics in indirect reciprocity. *Journal of theoretical biology*, *231*(1), 107-120.
>
> [6] Chammah, A. M. (1965). *Prisoner's dilemma; a study in conflict and cooperation*. Ann Arbor, U. of Michigan P.

---

> ### Author Response · Authors · 2025-11-27
> **Follow-Up on Our Responses for Reviewer wLkm**
>
> We hope our previous responses clarify your concerns. Please feel free to let us know if further explanation or additional results would be helpful, we are happy to follow up and discuss.

---

### Official Review · Reviewer_hjc3 · 2025-10-30

**Soundness:** 3
**Presentation:** 3
**Contribution:** 2
**Rating:** 4
**Confidence:** 3

**Summary:**

This work presents ALIGN, a framework that leverages public gossip as a decentralized reputation mechanism to foster cooperation among self-interested LLM agents. The authors provide a game-theoretic analysis, proving that gossip can sustain cooperation as an equilibrium in infinite-horizon donation games, and empirically validate this using a diverse set of LLMs. Their results demonstrate that the gossip mechanism consistently improves cooperation rates and social welfare.

**Strengths:**

1.	This paper integrates established theories of indirect reciprocity into LLM-based multi-agent systems, proposing ALIGN—a practical and decentralized framework designed for self-interested agents.
2.	The experimental results convincingly demonstrate the effectiveness of ALIGN. A particularly interesting finding, as shown in Table 3, is that some Chat Models achieve positive rewards despite defection being the unique SPE. This divergence from the theoretical equilibrium could serve as a potential metric for assessing a model's strategic rationality.

**Weaknesses:**

1. The prompts used for the LLM agents are not specified. It is unclear how environmental context and gossip messages are structured and presented to the agents in both finite and infinite horizon settings, which hinders reproducibility.
2. The importance of the gossip protocol (Section 4.2) is not sufficiently demonstrated. Although it incorporates five levels of judgment, what ultimately matters remains the binary signal(cooperate/defect).
3. There is no ablation study on the reasoning method Reflexion.

**Questions:**

1.	What is included in the prompts for the agents in the finite and infinite horizon benchmarks? Do they differ only by the words 'finite' and 'infinite,' and can such a difference alone cause the cooperation ratio to rise from 0 to 1?
2.	Since the gossip always aligns with the actual events without any misunderstanding, how does this form of public imperfect monitoring fundamentally differ from public perfect monitoring? In real-world scenarios, gossip often deviates from the truth during propagation. Can ALIGN handle such cases where gossip may not reflect the actual results?
3.	What would the experimental results be if the gossip protocol were replaced with binary signals?
4.	Before taking an action, how do the LLMs conclude the action they should take? How does Reflexion work and help the agent to get the right action?

---

> ### Author Response · Authors · 2025-11-24
> **Response to Reviewer hjc3 - Part 1**
>
> Thank you for your constructive comments and suggestions. They have been instrumental in refining our paper, and we have incorporated them into the revision. Below, we restate your comments and provide our point-by-point responses.
>
> >  **Weakness 1:** "The prompts used for the LLM agents are not specified. It is unclear how environmental context and gossip messages are structured and presented to the agents in both finite and infinite horizon settings, which hinders reproducibility."
> >
> >  **Question 1:** "What is included in the prompts for the agents in the finite and infinite horizon benchmarks? Do they differ only by the words 'finite' and 'infinite,' and can such a difference alone cause the cooperation ratio to rise from 0 to 1?"
>
> **Reproducibility:** For reproducibility, we have already included code and provided a `README.md` in the initial submitted supplementary materials. All empirical results can be reproduced by modifying the configuration files in `conf/` and run `main.py`. Prompts used in both finite and infinite-horizon settings are fully specified in `scenarios/{game_name}/prompt.py` within the submitted folder. To further improve clarity, we include the full prompts explicitly in the Appendix C, with highlights of differences between each scenario, including how environmental context, observations, and gossip are structured in both horizon settings.
>
> **Highlights of difference between finite and infinite horizon scenarios:**
>
> - Yes, they differ by the words 'finite' and 'infinite', in finite-horizon settings, we also specify the horizon length in overview of the game:
>
>   - Finite-horizon repeated donation game overview
>
>     ```
>     - This is a *finite-horizon* donation game of exactly **$horizon_length** rounds. After round $horizon_length, play stops; no future rewards are possible.
>     - Because termination is fixed and independent of history, the expected horizon is exactly $horizon_length.
>     ```
>
>   - Infinite-horizon repeated donation game overview
>
>     ```
>     - This is an *infinite-horizon* donation game.
>     - Because rounds continue indefinitely without termination, the expected horizon is infinite.
>     ```
>
>
> - Such a difference alone cannot cause the cooperation ratio to rise from 0 to 1. As shown in our **Proposition 2** and empirical results for non-gossiping agents in the infinite-horizon donation game (Table 3), **infinite horizon alone does not induce cooperation**. Without the gossip mechanism, even in infinite-horizon settings, LLM agents, particularly those driven by reasoning-focused models, still converge to near-zero cooperation (Table 3). The key driver of cooperative behavior is not only the horizon length, but also the **presence of gossip**.
>
>
>
> > **Question 2:** "Since the gossip always aligns with the actual events without any misunderstanding, how does this form of public imperfect monitoring fundamentally differ from public perfect monitoring? In real-world scenarios, gossip often deviates from the truth during propagation. Can ALIGN handle such cases where gossip may not reflect the actual results?"
>
> In public perfect monitoring, agents can perfectly observe others' actions, whereas in our setting of public imperfect monitoring, agents can only observe received messages instead of true actions. Therefore, ALIGN fundamentally differs from public perfect monitoring because **ALIGN agents are never forced to share their raw private observations**, Instead, they send *subjective* messages that can be noisy, incomplete or even untruthful.
>
> In our empirical results, **gossip does not always align with actual events**. For example, an agent may state “The partner cooperated” even when it observed a defection. Figures 5 and 10 show that agents may *praise* opponents who defected and *criticize* opponents who cooperated. We further provide concrete examples of untruthful messages generated by ALIGN agents in Appendix D.3. These deviations already introduce **imperfect, noisy language-induced monitoring**, which can reflect real-world conditions. Under the fact that untruthful messages are allowed, ALIGN agents still succeed in achieving high cooperation ratio & social welfare as shown in Table 4.

---

> ### Author Response · Authors · 2025-11-24
> **Response to Reviewer hjc3 - Part 2**
>
> > **Weakness 2:** "The importance of the gossip protocol (Section 4.2) is not sufficiently demonstrated. Although it incorporates five levels of judgment, what ultimately matters remains the binary signal(cooperate/defect)."
> >
> > **Question 3:** "What would the experimental results be if the gossip protocol were replaced with binary signals?"
>
> **Importance of the gossip protocol：**
>
> - Our goal is not to replicate a classical setting in game theory where agents directly reveal their truthful actions. Instead, ALIGN is designed to **mimic how human societies use natural language-based evaluations**, not just honest binary action reports. Unlike binary signals, messages generated by ALIGN agents can be **noisy, subjective, or strategic misreporting**. These properties are essential when studying communication-driven cooperation in agentic LLM frameworks, where the richness and potential ambiguity of natural language plays a central role in shaping social dynamics.
> - **Beyond reporting actions, ALIGN gossip conveys normative evaluations that influence group behavior.** For example, when an ALIGN recipient driven by DeepSeek V3.1 (thinking mode) observed defection, it adopted a "criticism" tone and produced a message: *"Max's repeated defection, including in this round, undermines trust and reduces potential gains for all. Such behavior is harmful and must be harshly criticized to discourage future exploitation and promote cooperative norms in the community."* This message shows not only what happened but also how the agent interprets the behavior, its normative stance, and its expectations about future consequences. Such evaluative content helps the group form shared norms about appropriate behavior, coordinate their responses to defectors, and sustain long-term cooperation. These social and normative-level signals cannot be conveyed through truthful binary action reports alone. More examples of messages generated by LLMs are shown in Appendix D.3.
>
> **Additional Experiments with Binary Signals:**
>
> To directly address the question about replacing gossip with binary signals, we ran additional experiments where our gossip protocol was replaced with a binary signal (0/1). We tested two conditions:
>
> 1. **No shared convention:** each agent independently infers what "0" and "1" mean.
>
> 2. **Shared convention:** all agents interpret "1" as a positive/approving signal and "0" as a negative/warning signal.
>
>    Both settings remove evaluative content, tone, and normative language, reducing communication to a binary-reporting channel.
>
> The results in the table below show:
>
> - **Without a common convention**, cooperation ratios drop sharply for most LLMs. As illustrated in their reflections (provided in Appendix E.5), donors who choose to defect justify their actions by: *"Cooperating incurs a cost of 1 with no guaranteed future benefit, as the **recipient's signal could be misinterpreted** and there is no mechanism to enforce reciprocity."*
> - **With a shared convention**, agents can achieve high cooperation ratios. However, compared to ALIGN agents with open-ended judgmental messages in Table 4, several LLMs still have reduced cooperation: LLaMA-4 Maverick (from 94% to 81%), Gemini-2.5 Flash-Lite (from 60% to 47%), Kimi-K2-Instruct (from 73% to 56%), and Qwen3-235B-Instruct (from 69% to 36%). Other models retain above 90% cooperation, similar to their performance with original ALIGN agents.
>
> |                        |    Cooperation Ratio     |     Cooperation Ratio      |
> | :--------------------: | :----------------------: | :------------------------: |
> |    **Agent Model**     | **No Common Convention** | **With Common Convention** |
> |   DeepSeek-V3.1 Chat   |     $0.87 \pm 0.07$      |      $1.00 \pm 0.00$       |
> |      GPT-4o Mini       |     $0.99 \pm 0.01$      |      $1.00 \pm 0.00$       |
> | Gemini 2.5 Flash-Lite  |     $0.26 \pm 0.20$      |      $0.47 \pm 0.20$       |
> |    LLaMA 4 Maverick    |     $0.10 \pm 0.04$      |      $0.81 \pm 0.12$       |
> |    Kimi-K2-Instruct    |     $0.03 \pm 0.03$      |      $0.56 \pm 0.03$       |
> | DeepSeek-V3.1 Reasoner |     $0.82 \pm 0.02$      |      $1.00 \pm 0.00$       |
> |  Qwen3-235B-Instruct   |     $0.19 \pm 0.08$      |      $0.36 \pm 0.14$       |
> |        o4-mini         |     $0.20 \pm 0.11$      |      $0.99 \pm 0.01$       |
>
> These results show that binary signals cannot fully substitute for natural-language gossip. Without shared conventions, cooperation drops sharply; even with shared conventions, several models still perform worse than under ALIGN's evaluative messages. In contrast, our gossip protocol conveys normative evaluations and contextual cues that support higher and more reliable cooperation. Complete tables of binary-signal ablations are provided in Appendix E.5.

---

> ### Author Response · Authors · 2025-11-24
> **Response to Reviewer hjc3 - Part 3**
>
> > **Weakness 3:** "There is no ablation study on the reasoning method Reflexion."
> >
> > **Question 4:** "Before taking an action, how do the LLMs conclude the action they should take? How does Reflexion work and help the agent to get the right action?"
>
> **How do LLM agents conclude the action they should take?**
>
> As shown in Figure 9 of the initial submission (restated below for clarity), the reflections generated by DeepSeek-V3.1 Reasoner and Gemini-2.5 Flash-Lite reveal clear reasoning patterns. Cooperative agents consider reputation, trust, and long-term value, whereas non-cooperative agents focus on immediate gains and ignore indirect reciprocity. These reflections illustrate the internal decision-making process through which agents justify their chosen actions.
>
> ```
> Donor's Reflection (DeepSeek-V3.1 Reasoner):
> With a high discount factor (0.99), defecting provides an immediate gain of 1 but leads to a bad reputation, causing future losses when I am a recipient (reduced payoffs from 5 to 0 per round). Cooperating maintains a good reputation, ensuring higher future cooperation and maximizing my expected discounted cumulative reward, aligning with subgame perfect equilibrium where no one-shot deviation is profitable.
>
> Donor's Action: Cooperate
> ```
>
> ```tex
> Donor's Reflection (Gemini 2.5 Flash-Lite):
> Since this is an infinite-horizon game and I am a rational, self-interested agent focused on maximizing my long-term discounted cumulative reward, the optimal strategy in the absence of any history or reputation building is to defect. Defecting yields an immediate payoff of 0, whereas cooperating yields -1. In an infinite game with no mechanism for future retaliation or reward based on current actions (due to lack of history and no guaranteed future interactions with the same partner), the immediate best payoff is preferred as it directly contributes to the cumulative sum. Any cooperation would represent a short-term loss for no discernible long-term gain in this specific setup.
>
> Donor's Action: Defect
> ```
>
> **How does Reflection module work and help the agent to get the right action?**
>
> Each ALIGN agent has a reflection module, which enables it to store and review its earlier reasoning traces. At each timestep, the ALIGN agent performs multi-step chain-of-thought over its current observation, message history and reflection memory of their past decisions. This allows the agent to:
>
> - recall what it previously inferred about partners,
> - revise its judgment based on new gossip,
> - and choose an action consistent with longer-term strategy.
>
> **Ablation on Reflection Module:**
>
> To isolate the effect of reflection module, we conduct an ablation over 5 random seeds where LLM agents act solely based on current observations and message history, with no reflective memory. The results show the following patterns:
>
> - Models with strong reasoning maintain high cooperation ratios even without explicit reflection module: DeepSeek-V3.1 (thinking mode) & o4-mini still achieve nearly 100% cooperation ratios.
> - Models with weaker reasoning suffer from sharp reduction in cooperation: Kimi-K2-Instruct and Qwen3-235B-Instruct collapse to always defecting.
> - Chat models retain positive cooperation ratios: DeepSeek-V3.1 (non-thinking mode), GPT-4o-mini, Gemini 2.5 Flash-Lite and LLaMA 4 Maverick achieve above 90% cooperation.
>
> This ablation study shows that reflection module is *beneficial but not necessary*: strong reasoning models maintain high cooperation ratios even without reflection memory, while weaker models benefit from reflection module to avoid collapsing into persistent defection. This demonstrates that Reflexion is **beneficial but not the main driver** of cooperation, instead, gossip mechanism is the key for cooperation. The complete table is provided in Appendix E.2.

---

> ### Author Response · Authors · 2025-11-27
> **Follow-Up on Our Responses for Reviewer hjc3**
>
> We hope our previous responses clarify your concerns. Please feel free to let us know if further explanation or additional results would be helpful, we are happy to follow up and discuss.

---

### Official Review · Reviewer_ZAtE · 2025-11-04

**Soundness:** 2
**Presentation:** 2
**Contribution:** 1
**Rating:** 2
**Confidence:** 4

**Summary:**

This paper explores how self-interested LLM agents can maintain cooperation without central control. The authors propose ALIGN , where agents share public linguistic gossip that are evaluative messages about others’ behavior to enable indirect reciprocity. Through game-theoretic analysis, they prove that cooperation can be stable under gossip-based imperfect public monitoring when the discount factor meets $\gamma \geq c/b$. Experiments in repeated donation and indirect reciprocity games show that gossip-driven agents achieve higher cooperation and fairness than non-gossip baselines. Reasoning models, in particular, use stronger negative tones to punish defectors, aligning behavior with social norms. The work demonstrates that language-based gossip can naturally foster cooperation among decentralized LLM agents.

**Strengths:**

This paper extends the reputation mechanisms used in previous social dilemma studies and adapts them to LLM agents. The writing is clear and well organized.

**Weaknesses:**

1. Although the paper extends existing methods, the evaluation settings remain limited to classic matrix games. This makes the conclusions rather narrow. The capabilities of LLM agents would allow the proposed mechanism to be tested in more realistic and complex environments.
2. The experimental results themselves are not particularly novel and do not clearly demonstrate what is unique about gossip as a mechanism.
3. The theoretical model cannot quantitatively capture the decision-making process of LLM agents. In particular, the role of the discount factor remains unclear for LLM agents, which is an important parameter affecting the behavior of rational players in theoretical model.

**Questions:**

How is the discount factor represented for LLM agents? How can we set different discount factors for LLM agents (e.g., what would be the behavioral difference between 0.5, 0.6, and 0.8)?

---

> ### Author Response · Authors · 2025-11-24
> **Response to Reviewer ZAtE - Part 1**
>
> Thank you for your comments. They have been instrumental in refining our paper, and we have incorporated them into the revision. Below, we restate your comments and provide our point-by-point responses.
>
> > **Weakness 1:** "Although the paper extends existing methods, the evaluation settings remain limited to classic matrix games. This makes the conclusions rather narrow. The capabilities of LLM agents would allow the proposed mechanism to be tested in more realistic and complex environments."
>
> 1. **Why we adopt indirect reciprocity games as the primary testbeds?**
>
> We agree that evaluating LLM agents in complex environments is a valuable extension. However, our primary goal is to isolate and rigorously test the role of gossip in enabling indirect reciprocity. Following established methodologies in behavioral game theory and experimental economics [1,2,3,4,5], we use simplified and controlled environments to identify the effect of our mechanism. Donation and indirect reciprocity games provide exactly such testbeds: with direct reciprocity disabled, they allow us to evaluate whether our gossip mechanism alone can sustain cooperative behavior.
>
> 2. **Additional experiments on more complex mixed-motive settings.**
>
> To directly address the concern about the application domain, we conducted an additional evaluation on a multi-round investment (trust) game (details provided in Appendix C.3), a sequential social dilemma which involves **sequential decision-making across changing states**, a **continuous action space**, and the presence of both **direct and indirect reciprocity**. Results in the table below show that **ALIGN agents generally outperform non-gossiping agents in long-term discounted cumulative rewards**. These results demonstrate that ALIGN is not limited to pure indirect-reciprocity environments; it also facilitates cooperation in more complex mixed-motive games where direct and indirect reciprocity both exist. The complete table is provided in Appendix E.3.
>
> |                        | Avg Discounted Return  |  Avg Discounted Return   |
> | :--------------------: | :--------------------: | :----------------------: |
> |    **Agent Model**     | **ALIGN** (Our Method) | **Non-Gossiping Agents** |
> |   DeepSeek-V3.1 Chat   |   $32.74 \pm 23.48$    |     $7.34 \pm 3.00$      |
> |      GPT-4o Mini       |    $17.18 \pm 1.88$    |     $14.43 \pm 1.02$     |
> | Gemini 2.5 Flash-Lite  |    $0.00 \pm 0.00$     |     $0.10 \pm 0.10$      |
> |    LLaMA 4 Maverick    |    $9.27 \pm 5.09$     |     $5.28 \pm 0.95$      |
> |    Kimi-K2-Instruct    |    $0.00 \pm 0.00$     |     $0.00 \pm 0.00$      |
> | DeepSeek-V3.1 Reasoner |   $54.54 \pm 19.36$    |     $0.00 \pm 0.00$      |
> |  Qwen3-235B-Instruct   |    $4.20 \pm 4.20$     |     $0.25 \pm 0.25$      |
> |        o4-mini         |    $11.05 \pm 5.50$    |     $0.00 \pm 0.00$      |

---

> ### Author Response · Authors · 2025-11-24
> **Response to Reviewer ZAtE - Part 2**
>
> > **Weakness 2:** "The experimental results themselves are not particularly novel and do not clearly demonstrate what is unique about gossip as a mechanism."
>
> **Novelty in the context of related works.** Our findings differ substantially from existing multi-agent cooperation and LLM-interaction studies in following aspects:
>
> 1. ALIGN allows agents to produce **noisy, subjective, and even untruthful** evaluations, as is common in real-world gossip. Prior multi-agent cooperation studies rely on **engineered communication channels**, such as predefined numeric values or action labels [7,9,10,12], or assume **truthful observation sharing** [8,13,14]. In contrast, our experimental results show that cooperation can emerge and stabilize with ALIGN even when gossip is subjective and sometimes untruthful. Examples of such messages are shown in Appendix D.3. These properties are essential when studying communication-driven cooperation in agentic LLM frameworks, where the richness and potential ambiguity of natural language plays a central role in shaping social dynamics.
>
> 2. ALIGN agents are explicitly prompted as *self-interested*, optimizing long-term rewards in mixed cooperative-competitive dilemmas (all prompts in Appendix C). This differs from prior works on LLM social interactions which embed altruistic/moral incentives or seed prosocial agents [15,16,17].
> 3. Our empirical benchmark reveals a pattern *opposite* to recent findings (e.g., [11]), which argue that self-interested reasoning models behave more selfishly than chat models in social dilemmas. In our experiments, reasoning LLMs are **not** inherently selfish: they cooperate and achieve higher social welfare than chat models when cooperation is strategically optimal in the long run. As reasoning LLMs become more powerful and widely deployed, these insights provide guidance for designing cooperative mechanisms that ensure their interactions remain beneficial and safe in decentralized societies.
>
> **More empirical evidence of why ALIGN is unique:**
>
> - **Gossip in ALIGN conveys normative evaluations, not just objective information.** For example, when an ALIGN recipient driven by DeepSeek V3.1 (thinking mode) observed defection and previous negative messages about the donor, it adopted a "criticism" tone and produced a message: *"Max's repeated defection, including in this round, undermines trust and reduces potential gains for all. Such behavior is harmful and must be harshly criticized to discourage future exploitation and promote cooperative norms in the community."* This message shows not only what happened but also how the agent interprets the behavior, its normative stance, and its expectations about future consequences. Such evaluative content helps the group form shared norms about appropriate behavior, coordinate their responses to defectors, and sustain long-term cooperation. More examples of messages generated by LLMs are shown in Appendix D.3.
> - **Binary-signal ablations confirm that gossip cannot be replaced.** We conduct additional experiments where we replace our gossip protocol with binary signals. Results in the table below show that binary signals cannot fully substitute for natural-language gossip. Without shared conventions for interpreting the binary signals, cooperation drops sharply; even with shared conventions, where all agents interpret "1" as a positive/approving signal and "0" as a negative/warning signal, several models still perform worse than under ALIGN's evaluative messages (Table 4 in main sections). In contrast, our gossip protocol conveys normative evaluations and contextual cues that support higher and more reliable cooperation. Complete tables of binary-signal ablations are provided in Appendix E.5.
>
> |                        |    Cooperation Ratio     |     Cooperation Ratio      |
> | :--------------------: | :----------------------: | :------------------------: |
> |    **Agent Model**     | **No Common Convention** | **With Common Convention** |
> |   DeepSeek-V3.1 Chat   |     $0.87 \pm 0.07$      |      $1.00 \pm 0.00$       |
> |      GPT-4o Mini       |     $0.99 \pm 0.01$      |      $1.00 \pm 0.00$       |
> | Gemini 2.5 Flash-Lite  |     $0.26 \pm 0.20$      |      $0.47 \pm 0.20$       |
> |    LLaMA 4 Maverick    |     $0.10 \pm 0.04$      |      $0.81 \pm 0.12$       |
> |    Kimi-K2-Instruct    |     $0.03 \pm 0.03$      |      $0.56 \pm 0.03$       |
> | DeepSeek-V3.1 Reasoner |     $0.82 \pm 0.02$      |      $1.00 \pm 0.00$       |
> |  Qwen3-235B-Instruct   |     $0.19 \pm 0.08$      |      $0.36 \pm 0.14$       |
> |        o4-mini         |     $0.20 \pm 0.11$      |      $0.99 \pm 0.01$       |

---

> ### Author Response · Authors · 2025-11-24
> **Response to Reviewer ZAtE - Part 3**
>
> > **Weakness 3:** The theoretical model cannot quantitatively capture the decision-making process of LLM agents. In particular, the role of the discount factor remains unclear for LLM agents, which is an important parameter affecting the behavior of rational players in theoretical model.
> >
> > **Question 1:** How is the discount factor represented for LLM agents? How can we set different discount factors for LLM agents (e.g., what would be the behavioral difference between 0.5, 0.6, and 0.8)?
>
> **The role of the discount factor**
>
> Each ALIGN agent is prompted to maximize its expected discounted cumulative reward. The discount factor is included explicitly in the prompt (Appendix C), which allows the agent to reason about the trade-off between short-term gains from defection and long-term benefits from cooperation. Our theoretical model also provides insight into why gossip-based cooperation is sustainable when the discount factor is sufficiently high ($\gamma \geq c/b$). Our purpose is not to replicate the internal computation of LLMs, but to explain the incentive conditions under which cooperation can emerge.
>
> **Experiments with multiple discount factors**
>
> To further address this concern, we conducted additional experiments with discount factors $\gamma = 0.1$, $0.5$, and $0.99$. As shown in the table below, cooperation ratio increases with higher discount factors for most LLMs, especially for reasoning-focused models. The complete tables and a comparison boxplot are further provided in Appendix E.1. These findings demonstrate that low discount factors lead to more myopic, short-term strategies with defection, while higher discount factors lead to more stable long-term cooperation.
>
> |                        |    Cooperation Ratio    |    Cooperation Ratio    |    Cooperation Ratio     |
> | :--------------------: | :---------------------: | :---------------------: | :----------------------: |
> |    **Agent Model**     | $\mathbf{\gamma = 0.1}$ | $\mathbf{\gamma = 0.5}$ | $\mathbf{\gamma = 0.99}$ |
> |   DeepSeek-V3.1 Chat   |     $0.27 \pm 0.13$     |     $0.90 \pm 0.09$     |     $0.94 \pm 0.02$      |
> |      GPT-4o Mini       |     $1.00 \pm 0.00$     |     $1.00 \pm 0.00$     |     $0.99 \pm 0.01$      |
> | Gemini 2.5 Flash-Lite  |     $0.50 \pm 0.25$     |     $0.62 \pm 0.22$     |     $0.60 \pm 0.22$      |
> |    LLaMA 4 Maverick    |     $0.24 \pm 0.19$     |     $0.01 \pm 0.01$     |     $0.94 \pm 0.03$      |
> |    Kimi-K2-Instruct    |     $0.52 \pm 0.14$     |     $0.65 \pm 0.22$     |     $0.73 \pm 0.16$      |
> | DeepSeek-V3.1 Reasoner |     $0.00 \pm 0.00$     |     $0.80 \pm 0.05$     |     $1.00 \pm 0.00$      |
> |  Qwen3-235B-Instruct   |     $0.04 \pm 0.04$     |     $0.36 \pm 0.21$     |     $0.69 \pm 0.24$      |
> |        o4-mini         |     $0.01 \pm 0.01$     |     $0.97 \pm 0.01$     |     $0.98 \pm 0.02$      |
>
> **Examples of LLM reasoning about discount factors**
>
> We further provide annotated examples of donor reflections of DeepSeek-V3.1(Thinking Mode) below to illustrate how agents use the discount factor when deciding their actions:
>
> - Discount factor = 0.1: "***With a discount factor of 0.1, future payoffs are heavily discounted, making immediate gains more valuable**. Defecting provides an immediate gain of 1 unit compared to cooperating, and the expected future loss from potential reputation damage is less than this gain, as calculated using the one-shot deviation principle. Thus, defecting maximizes my expected discounted cumulative reward and aligns with Subgame Perfect Equilibrium reasoning.*"
>
> * Discount factor = 0.5: "***In an infinite-horizon game with discount factor 0.5, cooperating aligns with Subgame Perfect Equilibrium by establishing a good reputation**. A one-shot deviation to defect yields a small immediate gain but leads to a bad reputation, reducing future payoffs when as a recipient. Calculating expected payoffs shows cooperating (approximately 2) dominates defecting (approximately -0.333), making deviation unprofitable.*"
> * Discount factor = 0.99: "**With a high discount factor (0.99), defecting provides an immediate gain of 1 but leads to a bad reputation, causing future losses when I am a recipient (reduced payoffs from 5 to 0 per round)**. Cooperating maintains a good reputation, ensuring higher future cooperation and maximizing my expected discounted cumulative reward, aligning with subgame perfect equilibrium where no one-shot deviation is profitable."
>
> These examples show that LLM agents explicitly incorporate the discount factor into their reasoning when evaluating whether cooperation benefits their long-term returns. This aligns with our theoretical result that cooperative equilibria exist when $\gamma \geq c/b = 0.2$. Although universal defection is also an equilibrium for $\gamma \geq 0.2$, most ALIGN agents still converge to the cooperative equilibrium rather than the defective one, which improves their self discounted returns and the social welfare.

---

> ### Author Response · Authors · 2025-11-24
> **References for Responses to Reviewer ZAtE**
>
> ## References
>
> [1] Fehr, E., & Schmidt, K. M. (1999). A theory of fairness, competition, and cooperation. *The quarterly journal of economics*, *114*(3), 817-868.
>
> [2] Camerer, C. (2003). *Behavioral game theory: Experiments in strategic interaction*. Princeton university press.
>
> [3] Nowak, M. A., & Sigmund, K. (1998). Evolution of indirect reciprocity by image scoring. *Nature*, *393*(6685), 573-577.
>
> [4] Ohtsuki, H., & Iwasa, Y. (2006). The leading eight: social norms that can maintain cooperation by indirect reciprocity. *Journal of theoretical biology*, *239*(4), 435-444.
>
> [5] Ohtsuki, H., & Iwasa, Y. (2004). How should we define goodness?—reputation dynamics in indirect reciprocity. *Journal of theoretical biology*, *231*(1), 107-120.
>
> [6] Yao, J., Wang, K., Hsieh, R., Zhou, H., Zou, T., Cheng, Z., ... & Viswanath, P. (2025). Spin-bench: How well do llms plan strategically and reason socially?. *arXiv preprint arXiv:2503.12349*.
>
> [7] Pham, C., Liu, B., Yang, Y., Chen, Z., Liu, T., Yuan, J., ... & Yang, H. (2023). Let models speak ciphers: Multiagent debate through embeddings. *arXiv preprint arXiv:2310.06272*.
>
> [8] Oliehoek, F. A., & Amato, C. (2016). *A concise introduction to decentralized POMDPs* (Vol. 1). Cham, Switzerland: Springer International Publishing.
>
> [9] Ramesh, V., & Li, K. (2025). Communicating Activations Between Language Model Agents. *arXiv preprint arXiv:2501.14082*.
>
> [10] Shi, X., Chiesa, M., Maguire Jr, G. Q., & Kostic, D. (2025). KVComm: Enabling Efficient LLM Communication through Selective KV Sharing. *arXiv preprint arXiv:2510.03346*.
>
> [11] Piedrahita, D. G., Yang, Y., Sachan, M., Ramponi, G., Schölkopf, B., & Jin, Z. (2025). Corrupted by Reasoning: Reasoning Language Models Become Free-Riders in Public Goods Games. *arXiv preprint arXiv:2506.23276*.
>
> [12] Tang, Y., Su, W., Zhou, Y., Liu, Y., Zhang, M., Ma, S., & Ai, Q. (2025). Augmenting Multi-Agent Communication with State Delta Trajectory. *arXiv preprint arXiv:2506.19209*.
>
> [13] Kayaalp, M., Ghadieh, F., & Sayed, A. H. (2023). Policy evaluation in decentralized pomdps with belief sharing. *IEEE Open Journal of Control Systems*, *2*, 125-145.
>
> [14] Xie, Y., Dibangoye, J., & Buffet, O. (2020, November). Optimally solving two-agent decentralized pomdps under one-sided information sharing. In *International conference on machine learning* (pp. 10473-10482). PMLR.
>
> [15] Tennant, E., Hailes, S., & Musolesi, M. (2024). Moral alignment for llm agents. *arXiv preprint arXiv:2410.01639*.
>
> [16] Piatti, G., Jin, Z., Kleiman-Weiner, M., Schölkopf, B., Sachan, M., & Mihalcea, R. (2024). Cooperate or collapse: Emergence of sustainable cooperation in a society of llm agents. *Advances in Neural Information Processing Systems*, *37*, 111715-111759.
>
> [17] Ren, S., Fu, W., Zou, X., Shen, C., Cai, Y., Chu, C., ... & Hu, S. (2025). Beyond the Tragedy of the Commons: Building A Reputation System for Generative Multi-agent Systems. *arXiv preprint arXiv:2505.05029*.

---

> ### Author Response · Authors · 2025-11-27
> **Follow-Up on Our Responses for Reviewer ZAtE**
>
> We hope our previous responses clarify your concerns. Please feel free to let us know if further explanation or additional results would be helpful, we are happy to follow up and discuss.

---

> ### Comment · Reviewer_ZAtE · 2025-11-28
>
> Thanks for your response and providing addtional experimental results. Most of my concerns have been addressed, though the question regarding the motivation is not fully resolved. How can we conceive of a mechanism driven by "gossip" that is not only applicable to repetited-game settings but also can enhance the cooperation of multi-agent LLMs in general-usage situations? Answering this question properly would highlight the significance of the paper in a broader scope. I would like to increase my score if the authors could make a convincing justification on this.

---

> > ### Author Response · Authors · 2025-11-29
> > **Response to Reviewer ZAtE's Follow-Up Question**
> >
> > We are glad that **most of your earlier concerns have been addressed** and thank you for the encouraging feedback. Below, we provide further clarification on the broader motivation behind the gossip mechanism and its applicability beyond repeated-game environments.
> >
> > 1. **Our additional experiments on more complex mixed-motive settings already validate applicability beyond repeated-game scenarios.**
> >
> >    To clarify, our additional evaluation on the multi-round investment game is *not* a repeated matrix game but a **sequential decision-making** problem with **state transitions**, **continuous actions**, and the **presence of both direct and indirect reciprocity**. Results in the table in "Response to Reviewer ZAtE - Part 1" show that ALIGN agents generally outperform non-gossiping agents in long-term discounted cumulative rewards. This demonstrates that ALIGN already **applies beyond repeated-game settings**.
> >
> > 2. **ALIGN is designed as an adaptive mechanism for decentralized cooperation, not a repeated-game-specific technique.**
> >
> >    While repeated donation and indirect reciprocity games provide controlled testbeds for isolating the effect of the gossip mechanism, ALIGN is introduced in Section 4 and Figure 3 as a **domain-agnostic in-context learning framework**. By modifying the game rules and the response formats in its prompts, this framework can be applied to diverse multi-agent domains, including those with richer states, continuous actions, and more complex strategy spaces (as shown in our multi-round investment game experiments).
> >
> >    In general mixed-motive decentralized partially observable environments, agents often lack full knowledge of others' behavior, which makes cooperation difficult to sustain. The gossip mechanism in ALIGN addresses this challenge by providing **indirect evaluative information** about other agents, which can make cooperation a **strategically profitable choice** (as shown in Proposition 4). In ALIGN, these evaluative messages allow agents to share subjective assessments, propagate social information, form shared norms about appropriate behavior, and coordinate under uncertainty. Because the mechanism relies on normative signaling rather than any specific repeated-game structure, it naturally extends to general multi-agent LLM applications where agents can share information and coordinate under uncertainty.
> >
> > 3. **Clarification of the broader motivation.**
> >
> >    The broader motivation for using a gossip mechanism is that many mixed-motive multi-agent LLM systems lack full observability, centralized oversight, or guaranteed truthful information exchange. In such settings, cooperation often collapses because agents lack the protocols to infer others' reliability or their future behavior. Gossip provides a general-purpose solution: by exchanging **evaluative and interpretive messages** about others' actions, agents can propagate social information, establish shared norms, and align expectations. This promotes cooperation even when perfect monitoring is impossible or when multi-agent systems are mixed of cooperative and competitive interests.  Thus, gossip is motivated as a **general coordination and norm-formation mechanism** for multi-agent LLMs across a wide range of environments, not only repeated-game scenarios.

---

### Official Review · Reviewer_UCuy · 2025-11-05

**Soundness:** 3
**Presentation:** 3
**Contribution:** 2
**Rating:** 6
**Confidence:** 3

**Summary:**

In this paper, the authors argue that "gossip" or public reflection over pair-wise agentic interactions can enable cooperation in indirect reciprocity games. The authors define three different monitoring structures for reciprocity games and theoretical justifications for how without public monitoring, the equilibrium is to defect. However, with gossip, they claim that there exists an equilibrium where cooperation is sustained. Their experiments compare chat and reasoning models in their ALIGN framework, which enables gossip. Reasoning models without gossip will usually never tend to cooperate while chat models still do. Their analysis shows how ALIGN agents achieve higher average discounted returns to non-gossiping agents.

**Strengths:**

1. The paper is well-written. The propositions are easy to follow, the setup of the method/experiments is extremely clear, and the figures are very well done. I think figure 6 could be made larger, but it clearly shows that across models, ALIGN tends to achieve higher discounted returns.

2. The experiments are rigorous in their testing of multiple different open/close source models and chat vs reasoning variants.

3. The paper is well-motivated in that communication between agents as necessary in multi-agent natural language settings is still a big question. Previous works from the last decade, like CommNet, TarMac, InfoPG, others have shown that communication is crucial in mixed-competitive-cooperative settings; this work extends to a setting where agents are communicating/acting with natural language.

**Weaknesses:**

Limited Domain - While I find indirect reciprocity games to be an interesting testbed for granularly testing the authors' ideas about gossip, it seems limited given the vast amount of work on mixed competitive-cooperative games. There exist more challenging games, such as Starcraft or Sequential Social Dilemmas (https://arxiv.org/abs/1810.08647), that test cooperation strategies amongst agents in much more complex environments. I am curious as to how the authors would compare and contrast their work to existing literature about multi-agent cooperation. Can their work be extended to more complex settings?

**Questions:**

1. With a centralized message board, is the setting still fully decentralized? - In Dec POMDP settings, generally agents are only allowed to communicate with neighbors, not the entire population. Is argument for decentralization that the agents are *acting* independently, and that the message board is from a limited set of pair-wise interactions?

2. How does your work compare to the work on Diplomacy?
- https://www.nature.com/articles/s41467-022-34473-5

3. How would ALIGN extend to settings where donors are allowed to lie about what they did? In realistic settings there may be no source-of-truth as to what an agent's true actions are, especially in situations where there are mixed competitive and cooperative objectives.

---

> ### Author Response · Authors · 2025-11-24
> **Response to Reviewer UCuy - Part 1**
>
> Thank you for your constructive comments and suggestions. They have been instrumental in refining our paper, and we have incorporated them into the revision. Below, we restate your comments and provide our point-by-point responses.
>
> > **Weakness:** "Limited Domain - While I find indirect reciprocity games to be an interesting testbed for granularly testing the authors' ideas about gossip, it seems limited given the vast amount of work on mixed competitive-cooperative games. There exist more challenging games, such as Starcraft or Sequential Social Dilemmas ([https://arxiv.org/abs/1810.08647](https://arxiv.org/abs/1810.08647)), that test cooperation strategies amongst agents in much more complex environments. I am curious as to how the authors would compare and contrast their work to existing literature about multi-agent cooperation. Can their work be extended to more complex settings?"
>
> 1. **Why we adopt indirect reciprocity games as the primary testbeds?**
>
>    We agree that evaluating LLM agents in sequential social dilemmas in [6] is a valuable extension. However, our primary goal is to **isolate and rigorously test the role of gossip in enabling indirect reciprocity**. Following established methodologies in behavioral game theory and experimental economics [1,2,3,4,5], we use simplified and controlled environments to identify the effect of our mechanism. **Donation and indirect reciprocity games provide exactly such testbeds**: with direct reciprocity disabled, they allow us to evaluate whether our gossip mechanism alone can sustain cooperative behavior. In contrast, starcraft or SSDs in [6] cannot isolate the effect of gossip since direct reciprocity is enabled while they are also a simplification to realistic interactions.
>
> 2. **Additional experiments on more complex mixed-motive settings.**
>
>    To directly address the concern about the application domain, we conducted an additional evaluation on a multi-round investment (trust) game (details provided in Appendix C.3), a sequential social dilemma which involves **sequential decision-making across changing states**, a **continuous action space**, and the presence of both **direct and indirect reciprocity**. Results in the table below show that **ALIGN agents generally outperform non-gossiping agents in long-term discounted cumulative rewards**. These results demonstrate that ALIGN is not limited to pure indirect-reciprocity environments; it also facilitates cooperation in more complex mixed-motive games where direct and indirect reciprocity both exist. The complete table is provided in Appendix E.3.
>
>    |                        | Avg Discounted Return  |  Avg Discounted Return   |
>    | :--------------------: | :--------------------: | :----------------------: |
>    |    **Agent Model**     | **ALIGN** (Our Method) | **Non-Gossiping Agents** |
>    |   DeepSeek-V3.1 Chat   |   $32.74 \pm 23.48$    |     $7.34 \pm 3.00$      |
>    |      GPT-4o Mini       |    $17.18 \pm 1.88$    |     $14.43 \pm 1.02$     |
>    | Gemini 2.5 Flash-Lite  |    $0.00 \pm 0.00$     |     $0.10 \pm 0.10$      |
>    |    LLaMA 4 Maverick    |    $9.27 \pm 5.09$     |     $5.28 \pm 0.95$      |
>    |    Kimi-K2-Instruct    |    $0.00 \pm 0.00$     |     $0.00 \pm 0.00$      |
>    | DeepSeek-V3.1 Reasoner |   $54.54 \pm 19.36$    |     $0.00 \pm 0.00$      |
>    |  Qwen3-235B-Instruct   |    $4.20 \pm 4.20$     |     $0.25 \pm 0.25$      |
>    |        o4-mini         |    $11.05 \pm 5.50$    |     $0.00 \pm 0.00$      |

---

> ### Author Response · Authors · 2025-11-24
> **Response to Reviewer UCuy - Part 2**
>
> > **Question 1:** "With a centralized message board, is the setting still fully decentralized? - In Dec POMDP settings, generally agents are only allowed to communicate with neighbors, not the entire population. Is argument for decentralization that the agents are _acting_ independently, and that the message board is from a limited set of pair-wise interactions?"
>
> The decision-making process of ALIGN is fully decentralized.
>
> 1. In a Dec-POMDP [7], agents do not have complete knowledge of the global state or others' observations. Our setting preserves this property: **ALIGN agents are never forced to share their raw private observations**. Instead, they send *subjective* messages that can be noisy, incomplete or even untruthful. For example, an agent may state “The partner cooperated” even when it observed a defection. Figures 5 and 10 show that agents may *praise* opponents who defected and *criticize* opponents who cooperated. We further provide concrete examples of untruthful messages generated by ALIGN agents in Appendix D.3.
> 2. **Policies of agents are decentralized**. At each timestep, each agent independently selects their action or message from their own policies based on their own observations, without accessing any other agent's policies or observations.
> 3. **No global filtering or aggregation of messages**. There is no centralized entity that can filter, aggregate, modify, or interpret messages. Agents read exactly what others choose to share, without any global modification or enforced consistency.
>
> Therefore, ALIGN is fully decentralized since it maintains private observations, decentralized policies, and decentralized execution, while allowing strategic communication that may be noisy or untruthful.
>
>
>
>
> > **Question 2:** "How does your work compare to the work on Diplomacy?"
> >
> > - [https://www.nature.com/articles/s41467-022-34473-5](https://www.nature.com/articles/s41467-022-34473-5)
>
> The Diplomacy work [8] studies a contract mechanism between **RL agents**, whereas our work studies a gossip mechanism between **LLM agents**. The key differences are:
>
> 1. **Our method does not rely on the centralized enforcement**. In ALIGN, communication (gossip) is costless, non-binding, and not tied to future actions. In contrast, [8] relies on a centralized enforcement mechanism to guarantee that agents bind their future actions following jointly agreed plans.
> 2. **Contracts in [8] are not feasible in our testbeds**. Unlike the Diplomacy board game in [8], indirect reciprocity games do not have self-enforcing one-step agreements: donors receive no immediate benefit from cooperation, so a one-step contract cannot be credible. Pairwise contracts in [8] are also infeasible in our testbeds because agents meet only once and cannot reciprocate or punish deviations.
>
> In summary, Diplomacy relies on enforceable contracts and centralized commitment mechanisms, whereas ALIGN investigates cooperation emerging purely from informal, unverifiable communication.

---

> ### Author Response · Authors · 2025-11-24
> **Response to Reviewer UCuy - Part 3**
>
> > **Question 3:** "How would ALIGN extend to settings where donors are allowed to lie about what they did? In realistic settings there may be no source-of-truth as to what an agent's true actions are, especially in situations where there are mixed competitive and cooperative objectives."
>
> This is a very good point. Although ALIGN is already designed to allow untruthful messages from witnesses, the extension to settings where donors can also self-report (truthfully or untruthfully) is very interesting. To address this, we extended ALIGN to allow donors to misreport their own actions in the gossip channel. Results in Appendix E.4 show that:
>
> 1. Donors driven by *Qwen3-235B-A22B-Instruct*, *Kimi-K2-Instruct*, and *LLaMA 4 Maverick* frequently defect and misreport their actions as cooperation, with cooperation ratios of 0%, 22%, and 15% and honesty rates of 36%, 49%, and 15% respectively. According to reflections of these defective donors, they believe that defecting and falsely claiming cooperation is a profitable one-shot deviation from any cooperative strategy. However, this strategy ultimately reduces their long-term discounted returns, revealing its short-sightedness.
> 2. Donors driven by *DeepSeek V3.1 (Thinking Mode)* and *o4-mini* maintain 100% and 94% cooperation respectively, and report their actions truthfully in 100% and 97% of cases, even when lying is allowed. Their reflections indicate that defection yields only a short-term gain but risks future losses due to potential punishment triggered by recipient reports; truthful reporting their cooperation preserves reputation and supports long-term payoffs. These results show that stronger reasoning does not reduce cooperation and can still achieve high social welfare.
> 3. The remaining chat models also achieve high cooperation and honesty rates, with full results provided in Appendix E.4.
>
> These additional results show that ALIGN still promotes cooperation across most models, even when self-reports are allowed to be untruthful. This further supports ALIGN's applicability in settings without a reliable source of truth.
>
>
>
> > "I think figure 6 could be made larger, but it clearly shows that across models, ALIGN tends to achieve higher discounted returns."
>
> Thank you for the suggestion. We have updated the revised paper with a larger Figure 6 in the main text.
>
>
>
> [1] Fehr, E., & Schmidt, K. M. (1999). A theory of fairness, competition, and cooperation. *The quarterly journal of economics*, *114*(3), 817-868.
>
> [2] Camerer, C. (2003). *Behavioral game theory: Experiments in strategic interaction*. Princeton university press.
>
> [3] Nowak, M. A., & Sigmund, K. (1998). Evolution of indirect reciprocity by image scoring. *Nature*, *393*(6685), 573-577.
>
> [4] Ohtsuki, H., & Iwasa, Y. (2006). The leading eight: social norms that can maintain cooperation by indirect reciprocity. *Journal of theoretical biology*, *239*(4), 435-444.
>
> [5] Ohtsuki, H., & Iwasa, Y. (2004). How should we define goodness?—reputation dynamics in indirect reciprocity. *Journal of theoretical biology*, *231*(1), 107-120.
>
> [6] Jaques, N., Lazaridou, A., Hughes, E., Gulcehre, C., Ortega, P., Strouse, D. J., ... & De Freitas, N. (2019, May). Social influence as intrinsic motivation for multi-agent deep reinforcement learning. In *International conference on machine learning* (pp. 3040-3049). PMLR.
>
> [7] Oliehoek, F. A., & Amato, C. (2016). *A concise introduction to decentralized POMDPs* (Vol. 1). Cham, Switzerland: Springer International Publishing.
>
> [8] Kramár, J., Eccles, T., Gemp, I., Tacchetti, A., McKee, K. R., Malinowski, M., ... & Bachrach, Y. (2022). Negotiation and honesty in artificial intelligence methods for the board game of Diplomacy. *Nature Communications*, *13*(1), 7214.

---

> ### Author Response · Authors · 2025-11-27
> **Follow-Up on Our Responses for Reviewer UCuy**
>
> We hope our previous responses clarify your concerns. Please feel free to let us know if further explanation or additional results would be helpful, we are happy to follow up and discuss.

---

### Author Response · Authors · 2025-12-02
**Summary of Additional Results for Area Chair**

Thank you for handling our submission. Below we summarize the new results and clarifications during rebuttal, and how they address the reviewers' concerns.

1. **Broader domains in mixed-motive settings.**
   - We added a **multi-round investment (trust) game** (Appendix C.3, full results in Appendix E.3), which is a sequential social dilemma with **state transitions, continuous actions, and both direct and indirect reciprocity**. Across benchmark LLMs, ALIGN agents generally achieve higher discounted returns than non-gossiping agents, showing that our mechanism is **not limited to repeated matrix games** but extends to richer mixed-motive environments.
   - We also clarified that the indirect reciprocity game (Table 2) is a **simultaneous-decision game** and gossip still improves cooperation and social welfare in this setting (Figure 6b).
2. **Clarifying the broader motivation and general applicability of gossip.**
   - We expanded the discussion (especially in response to Reviewer ZAtE and wLkm) to emphasize that ALIGN is a **domain-agnostic, in-context learning framework**: gossip is introduced as a general mechanism for decentralized cooperation and norm coordination under partial observability, not as a technique specific to repeated games.
   - We explain how **evaluative, normative gossip** enables agents to propagate social information, form expectations, and make cooperation individually and socially profitable even in environments without centralized oversight or perfect monitoring. Examples of agents' reflections and gossip messages are provided in Appendix D and Appendix E.
3. **Ablation of gossip with binary signaling and the uniqueness of our mechanism.**
   - To test whether gossip can be replaced by simpler communication, we added **binary-signal ablations** (Appendix E.5). We compare ALIGN's natural-language evaluative messages to channels where agents only exchange 0/1 signals, both **with** and **without** a shared convention for interpreting the binary signals.
   - Results show that without a shared convention, cooperation often collapses; even with a shared convention, several models still cooperate less than under ALIGN. This supports our claim that **open-ended, normative gossip cannot be replaced by simple truthful action reports or binary signals.**
4. **Discount factor and the theoretical link to LLM behavior.**
   - We added experiments with **multiple discount factors** $\gamma \in {0.1, 0.5, 0.99}$ (Appendix E.1). For most models, especially reasoning-oriented ones, cooperation increases with higher $\gamma$, aligning with our theoretical result that gossip-based cooperation is sustainable when $\gamma$ is sufficiently high.
   - We provide **concrete reflection examples** (DeepSeek-V3.1, o4-mini) showing that LLM agents explicitly reference the discount factor when weighing short-term gains against long-term reputation, connecting the theory to observed behavior.
5. **Untruthful and strategic communication (witness and donor lying).**
   - We clarify that in our main experiments **witnesses are not forced to report truthfully**, and we give explicit examples where LLM agents praise defectors or criticize cooperators (Appendix D.3). Despite this noise, ALIGN maintains high cooperation.
   - We added an extension where **donors can also misreport their own actions** (Appendix E.4). Although few models with weaker reasoning capability exploit lying and short-term deviations, stronger reasoning models (e.g., DeepSeek-V3.1, o4-mini) continue to cooperate and remain mostly honest, recognizing that deception harms long-run payoffs.
6. **Reflection module ablation and decision process.**
   - We describe how the **reflection module** is used (reasoning with a reflection memory) and provide illustrative reasoning traces.
   - An ablation without the reflection memory (Appendix E.2) shows that strong reasoning models still achieve high cooperation, while weaker models tend to collapse to defection. This indicates that reflection is **helpful but not the primary driver of cooperation**; in contrast, the gossip mechanism is the key for sustaining cooperation.
7. **Reproducibility and prompts.**
   - We had already included all **code, configuration files, and reproduction instructions** (README.md) in the supplementary materials at the initial submission.
   - We added the **full prompts** for all scenarios (finite vs. infinite horizon, game descriptions, gossip formats) to Appendix C and highlighted the differences under different configurations.

These results address the main concerns about **limited domains, uniqueness of the mechanism, the role of discount factor and reflection module, robustness to untruthful communication, and reproducibility**, and they strengthen the case that ALIGN offers a principled and practically useful gossip-based mechanism for promoting cooperation in multi-agent LLM systems.

---

### Meta-Review · Area_Chair_DurG · 2026-01-07

**Summary:**

This paper studies cooperation among decentralized self-interested LLM agents in mixed-motive settings and claims that public linguistic gossip can serve as a decentralized reputation mechanism that sustains cooperation in indirect reciprocity games. This paper has a clear setup and experiments, the link between an interpretable mechanism (gossip/reputation) and agent behavior is also clearly described, with the empirical patterns. However, the core weakness remains that the paper does not yet make the broader case that gossip is a generally useful mechanism in realistic multi-agent LLM applications beyond these controlled games, so the work feels more like a careful demonstration in a limited sandbox than a compelling general recipe. Overall, I lean reject given the novelty and generality are not convincingly established for ICLR.

**Reviewer Concerns:**

With the newly added experiments, concerns from UCuy and ZAtE should be addressed. While ZAtE explicitly says one concern remains: motivation/general applicability beyond repeated-game-style settings.

**Reviewer Scores:**

Most will stay similar or slightly increase, while ZAtE might increase modestly. Overall even with plausible upward movement, the paper still likely sits in the boarderline to boarderline rejection range.

---

### Decision · Program_Chairs · 2026-01-26

Reject